# When Demonstrations Meet Generative World Models: A Maximum Likelihood Framework for Offline Inverse Reinforcement Learning

**Siliang Zeng**[*]
University of Minnesota
Minneapolis, MN, USA
zeng0176@umn.edu

**Chenliang Li**[*]
Texas A&M University
College Station, TX, USA
chenliangli@tamu.edu

**Alfredo Garcia**
Texas A&M University
College Station, TX, USA
alfredo.garcia@tamu.edu

**Mingyi Hong**
University of Minnesota
Minneapolis, MN, USA
mhong@umn.edu

## Abstract

Offline inverse reinforcement learning (Offline IRL) aims to recover the structure of rewards and environment dynamics that underlie observed actions in a fixed, finite set of demonstrations from an expert agent. Accurate models of expertise in executing a task has applications in safety-sensitive applications such as clinical decision making and autonomous driving. However, the structure of an expert's preferences implicit in observed actions is closely linked to the expert's model of the environment dynamics (i.e. the "world" model). Thus, inaccurate models of the world obtained from finite data with limited coverage could compound inaccuracy in estimated rewards. To address this issue, we propose a bi-level optimization formulation of the estimation task wherein the upper level is likelihood maximization based upon a *conservative* model of the expert's policy (lower level). The policy model is conservative in that it maximizes reward subject to a penalty that is increasing in the uncertainty of the estimated model of the world. We propose a new algorithmic framework to solve the bi-level optimization problem formulation and provide statistical and computational guarantees of performance for the associated optimal reward estimator. Finally, we demonstrate that the proposed algorithm outperforms the state-of-the-art offline IRL and imitation learning benchmarks by a large margin, over the continuous control tasks in MuJoCo and different datasets in the D4RL benchmark[2].

## 1 Introduction

Reinforcement learning (RL) is a powerful and promising approach for solving large-scale sequential decision-making problems [1, 2, 3]. However, RL struggles to scale to the real-world applications due to two major limitations: 1) it heavily relies on the manually defined reward function [4], 2) it requires the online interactions with the environment [5]. In many application scenarios such as dialogue system [6] and robotics [7], it is difficult to manually design an appropriate reward for constructing the practical reinforcement learning system. Moreover, for some safety-sensitive applications like clinical decision making [8, 9] and autonomous driving [10, 11], online trials and errors are prohibited

---

[*]Equal Contribution.
[2]Our implementation is available at https://github.com/Cloud0723/Offline-MLIRL

37th Conference on Neural Information Processing Systems (NeurIPS 2023).

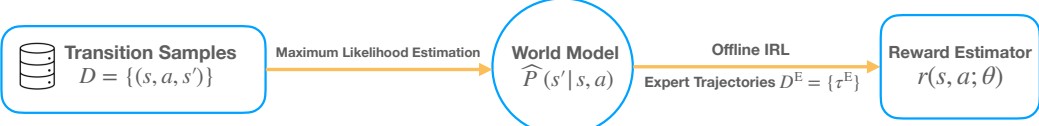

Figure 1: Illustration of the modular structure in our algorithmic framework, Offline ML-IRL. In Offline ML-IRL, it first estimates a world model from the dataset of transition samples, and then implements an ML based offline IRL algorithm on the estimated world model to recover the ground-truth reward function from the collected expert trajectories.

due to the safety concern. Due to these limitations in the practical applications, a new paradigm – learning from demonstrations, which relies on historical datasets of demonstrations to model the agent for solving sequential decision-making problems – becomes increasingly popular. In such a paradigm, it is important to understand the demonstrator and imitate the demonstrator's behavior by only utilizing the collected demonstration dataset itself, without further interactions with either the demonstrator or the environment.

In this context, offline inverse reinforcement learning (offline IRL) has become a promising candidate to enable learning from demonstrations [12, 13, 14, 9, 15]. Different from the setting of standard IRL [16, 17, 18, 19, 20, 21] which recovers the reward function and imitates expert behavior at the expense of extensive interactions with the environment, offline IRL is designed to get rid of the requirement in online environment interactions by only leveraging a finite dataset of demonstrations. While offline IRL holds great promises in practical applications, its study is still in an early stage, and many key challenges and open research questions remain to be addressed. For example, one central challenge in offline IRL arises from the so-called *distribution shift* [22, 15] – that is, the situation where the recovered reward function and recovered policy cannot generalize well to new unseen states and actions in the real environment. Moreover, in the training process of the offline IRL, any inaccuracies in the sequential decision-making process induced by distribution shift will compound, leading to poor performance of the estimated reward function / policy in the real-world environment. This is due to the fact that offline IRL is trained upon *fixed* datasets, which only provide *limited* coverage to the dynamics model of the real environment. Although there are some recent progress in offline IRL [14, 9, 15], how to alleviate distribution shift in offline IRL is still rarely studied and not clearly understood. Witnessing recent advances in a closely related area, the offline reinforcement learning, which incorporates conservative policy training to avoid overestimation of values in unseen states induced by the distribution shift [23, 24, 25, 26, 27], in this work we aim to propose effective offline IRL method to alleviate distribution shift and recover high-quality reward function from collected demonstration datasets. Due to the space limitations, we refer readers to Appendix B for more related work.

**Our Contributions.** To alleviate distribution shift and recover high-quality reward function from fixed demonstration datasets, we propose to incorporate conservatism into a model-based setting and consider offline IRL as a maximum likelihood estimation (MLE) problem. Overall, the goal is to recover a reward that generates an optimal policy to maximize the likelihood over observed expert demonstrations. Towards this end, we propose a two-stage procedure (see Fig. 1 for an overview). In the first stage, we estimate the dynamics model (the world model) from collected transition samples; by leveraging uncertainty estimation techniques to quantify the model uncertainty, we are able to construct a conservative Markov decision process (conservative MDP) where the state-action pairs with high model uncertainty and low data coverage receive a high penalty value to avoid risky exploration in the unfamiliar region. In the second stage, we propose an IRL algorithm to recover the reward function, whose corresponding optimal policy under the conservative MDP constructed in the first stage maximizes the likelihood of observed expert demonstrations. To the best of our knowledge, it is the first time that ML-based formulation, as well as the associated statistical and computational guarantees for reward recovery, has been developed for offline IRL.

To summarize, our main contributions are listed as follows:

• We consider a formulation of offline IRL based on MLE over observed transition samples and expert trajectories. In the proposed formulation, we respectively model the transition dynamics and

the reward function as the maximum likelihood estimators to generate all observed transition samples and all collected expert demonstrations. We provide a statistical guarantee to ensure that the optimal reward function of the proposed formulation could be recovered as long as the collected dataset of transition samples has sufficient coverage on the expert-visited state-action space.

• We develop a computationally efficient algorithm to solve the proposed formulation of offline IRL. To avoid repeatedly solving the policy optimization problem under each reward estimate, we propose an algorithm which alternates between one reward update step and one conservative policy improvement step. Under nonlinear parameterization for the reward function, we provide the theoretical analysis to show that the proposed algorithm converges to an approximate stationary point in finite time. Moreover, when the reward is linearly parameterized and there is sufficient coverage on the expert-visited state-action space to construct the estimated world model, we further show that the proposed algorithm approximately finds the optimal reward estimator of the MLE formulation.

• We conduct extensive experiments by using robotic control tasks in MuJoCo and collected datasts in D4RL benchmark. We show that the proposed algorithm outperforms the state-of-the-art offline IRL such as [14, 15] and imitation learning methods such as [28], especially when the number of observed expert demonstrations is limited. Moreover, we transfer the recovered reward across different datasets to show that the proposed method can recover high-quality reward function from the expert demonstrations.

## 2 Preliminaries and problem formulation

**Markov decision process (MDP)** is defined by the tuple $(\mathcal{S}, \mathcal{A}, P, \eta, r, \gamma)$, which consists of the state space $\mathcal{S}$, the action space $\mathcal{A}$, the transition dynamics $P : \mathcal{S} \times \mathcal{A} \times \mathcal{S} \to [0, 1]$, the initial state distribution $\eta(\cdot)$, the reward function $r : \mathcal{S} \times \mathcal{A} \to \mathbb{R}$ and the discounted factor $\gamma \in (0, 1)$. Under a transition dynamics model $P$ and a policy $\pi$, we are able to further define the state-action visitation measure as $d_P^\pi(s, a) := (1 - \gamma)\pi(a|s) \sum_{t=0}^{\infty} \gamma^t P^\pi(s_t = s|s_0 \sim \eta)$ for any state-action pair $(s, a)$.

**Maximum entropy inverse reinforcement learning (MaxEnt-IRL)** is a specific IRL formulation which aims to recover the ground-truth reward function and imitate the expert's policy from expert's demonstrations [18, 29, 20]. Let $\tau^{\mathrm{E}} := \{(s_t, a_t)\}_{t=0}^{\infty}$ denotes the expert trajectory sampled from the expert policy $\pi^{\mathrm{E}}$; let $\tau^{\mathrm{A}}$ denote the trajectory generated by the RL agent with policy $\pi$. Then the MaxEnt-IRL is formulated as:

$$\max_r \min_\pi \left\{ \mathbb{E}_{\tau^{\mathrm{E}} \sim \pi^{\mathrm{E}}} \left[ \sum_{t=0}^{\infty} \gamma^t \cdot r(s_t, a_t) \right] - \mathbb{E}_{\tau^{\mathrm{A}} \sim \pi} \left[ \sum_{t=0}^{\infty} \gamma^t \cdot r(s_t, a_t) \right] - H(\pi) \right\} \tag{1}$$

where $H(\pi) := \mathbb{E}_{\tau \sim \pi} \left[ \sum_{t=0}^{\infty} -\gamma^t \log \pi(a_t|s_t) \right]$ denotes the causal entropy of the policy $\pi$. The MaxEnt-IRL formulation aims to recover the ground-truth reward function which assigns high rewards to the expert policy while assigning low rewards to any other policies. Although MaxEnt-IRL has been well-studied theoretically [29, 30, 21, 31] and has been applied to several practical applications [32, 33, 34], it needs to repeatedly solve policy optimization problems under each reward function and online interactions with the environment is inevitable. Such repeated policy optimization subroutine requires extensive online trials and errors in the environment and thus makes MaxEnt-IRL quite limited in practical applications.

**Problem formulation** Let us now consider an ML formulation of offline IRL. Given the transition dataset $\mathcal{D} := \{(s, a, s')\}$, we train an estimated world model $\widehat{P}(s'|s, a)$ for any $s', s \in \mathcal{S}$ and $a \in \mathcal{A}$ (to be discussed in detail in Sec. 3). The constructed world model $\widehat{P}$ will be utilized as an estimate of the ground-truth dynamics model $P$. Based on the estimated world model $\widehat{P}$, we propose a model-based offline approach for IRL from the ML perspective, given below:

$$\max_\theta \quad L(\theta) := \mathbb{E}_{\tau^{\mathrm{E}} \sim (\eta, \pi^{\mathrm{E}}, P)} \left[ \sum_{t=0}^{\infty} \gamma^t \log \pi_\theta(a_t|s_t) \right] \tag{2a}$$

$$s.t. \quad \pi_\theta := \arg \max_\pi \mathbb{E}_{\tau^{\mathrm{A}} \sim (\eta, \pi, \widehat{P})} \left[ \sum_{t=0}^{\infty} \gamma^t \left( r(s_t, a_t; \theta) + U(s_t, a_t) + \mathcal{H}\big(\pi(\cdot|s_t)\big) \right) \right], \tag{2b}$$

where $\mathcal{H}\big(\pi(\cdot|s)\big) := \sum_{a \in \mathcal{A}} -\pi(a|s) \log \pi(a|s)$ denotes the entropy of the distribution $\pi(\cdot|s)$; $U(\cdot, \cdot)$ is a penalty function to quantify the uncertainty of the estimated world model $\widehat{P}(\cdot|s, a)$ under any

state-action pair $(s, a)$. In practice, the penalty function is constructed based on uncertainty heuristics over an ensemble of estimated dynamics models [35, 25, 24, 36]. A comprehensive study of the choices of the penalty function can be found in [37]. Next, let us make a few remarks about the above formulation, which we name *Offline ML-IRL*.

First, the problem takes a *bi-level* optimization form, where the lower-level problem (2b) assumes that the parameterized reward function $r(\cdot, \cdot; \theta)$ is fixed, and it describes the optimal policy $\pi_\theta$ as a unique solution to solve the conservative MDP; On the other hand, the upper-level problem (2a) optimizes the reward function $r(\cdot, \cdot; \theta)$ so that its corresponding optimal policy $\pi_\theta$ maximizes the log-likelihood $L(\theta)$ over observed expert trajectories.

Second, formulating the objective as a likelihood function is reasonable since it searches for an *optimal* reward function to explain the observed expert behavior within limited knowledge about the world (the world model $\widehat{P}$ is constructed based on a finite and diverse dataset $\mathcal{D} := \{(s, a, s')\}$).

Third, the lower-level problem (2b) corresponds to a model-based offline RL problem under the current reward estimate. The policy obtained is conservative in that state-action pairs that are not well covered by the dataset are penalized with a measure of uncertainty in the estimated world model. The penalty function $U(s, a)$ is used to quantify the model uncertainty and regularize the reward estimator. Therefore, the optimal policy $\pi_\theta$ under the conservative MDP will not take risky exploration on those *uncertain* region of the state-action space where the transition dataset does not have sufficient coverage, and the constructed world model has high prediction uncertainty.

## 3   The world model and statistical guarantee

In this section, we construct the world model $\widehat{P}$ from the transition dataset $\mathcal{D} := \{(s, a, s')\}$ and solve a certain approximated version of the formulation (2). Then we further show a high-quality reward estimator can be obtained with statistical guarantee.

Before proceeding, let us emphasize that one major challenge in solving (2) comes from the dynamics model mismatch between (2a) and (2b), which arises because the expert trajectory $\tau^{\mathrm{E}}$ is generated from the ground-truth transition dynamics $P$, while the agent samples its trajectory $\tau^{\mathrm{A}}$ through interacting with the estimated world model $\widehat{P}$. To better understand the above challenge, next we will explicitly analyze the likelihood objective in (2) and understand the mismatch error. Towards this end, let us introduce below the notions of the *soft Q-function* and the *soft value function* of the conservative MDP in (2) (defined for any reward parameter $\theta$ and the optimal policy $\pi_\theta$):

$$V_\theta(s) := \mathbb{E}_{\tau \sim (\eta, \pi_\theta, \widehat{P})} \Big[ \sum_{t=0}^{\infty} \gamma^t \Big( r(s_t, a_t; \theta) + U(s_t, a_t) + \mathcal{H}(\pi_\theta(\cdot|s_t)) \Big) \Big| s_0 = s \Big], \quad (3a)$$

$$Q_\theta(s, a) := r(s, a; \theta) + U(s, a) + \gamma \mathbb{E}_{s' \sim \widehat{P}(\cdot|s, a)} \big[ V_\theta(s') \big]. \quad (3b)$$

According to [38, 39, 40], the optimal policy $\pi_\theta$ and the optimal soft value function $V_\theta$ have the following closed-form expressions under any state-action pair $(s, a)$:

$$\pi_\theta(a|s) = \frac{\exp Q_\theta(s, a)}{\sum_{\tilde{a} \in \mathcal{A}} \exp Q_\theta(s, \tilde{a})}, \quad V_\theta(s) = \log \Big( \sum_{a \in \mathcal{A}} \exp Q_\theta(s, a) \Big). \quad (4)$$

Under the ground-truth dynamics model $P$ and the initial distribution $\eta(\cdot)$, we further define the visitation measure $d^{\mathrm{E}}(s, a)$ under the expert policy $\pi^{\mathrm{E}}$ as below:

$$d^{\mathrm{E}}(s, a) := (1 - \gamma) \pi^{\mathrm{E}}(a|s) \sum_{t=0}^{\infty} \gamma^t P^{\pi^{\mathrm{E}}}(s_t = s | s_0 \sim \eta). \quad (5)$$

By plugging the closed-form solution of the optimal policy $\pi_\theta$ into the objective function (2a), we can decompose the dynamics-model mismatch error from the likelihood function $L(\theta)$ in (2).

**Lemma 1.** *Under any reward parameter $\theta$, the objective $L(\theta)$ in (2a) can be decomposed as below:*

$$L(\theta) = \widehat{L}(\theta) + \frac{\gamma}{1 - \gamma} \cdot \mathbb{E}_{(s_t, a_t) \sim d^{\mathrm{E}}(\cdot, \cdot)} \Big[ \sum_{s' \in \mathcal{S}} V_\theta(s') \Big( \widehat{P}(s'|s, a) - P(s'|s, a) \Big) \Big] \quad (6)$$

*where $\widehat{L}(\theta)$ is a surrogate objective defined as:*

$$\widehat{L}(\theta) := \mathbb{E}_{\tau^{\mathrm{E}} \sim (\eta, \pi^{\mathrm{E}}, P)}\Big[\sum_{t=0}^{\infty} \gamma^t \Big(r(s_t, a_t; \theta) + U(s_t, a_t)\Big)\Big] - \mathbb{E}_{s_0 \sim \eta(\cdot)}\Big[V_\theta(s_0)\Big]. \qquad (7)$$

The detailed proof is included in Appendix D. In Lemma 1, we have shown that the likelihood function in (2) decomposes into two parts: a surrogate objective $\widehat{L}(\theta)$ and an error term dependent on the dynamics model mismatch between $\widehat{P}$ and $P$. As a remark, in the surrogate objective $\widehat{L}(\cdot)$, we separate the two dynamics models ($\widehat{P}$ and $P$) into two relatively independent components. Therefore, optimizing the surrogate objective is computationally tractable and we will propose an efficient algorithm to recover the reward parameter from it in the next section.

To further elaborate the connection between the likelihood objective $L(\theta)$ and the surrogate objective $\widehat{L}(\theta)$, we first introduce the following assumption:

**Assumption 1.** *For any reward parameter $\theta$ and state-action pair $(s, a)$, following conditions hold:*

$$|r(s, a; \theta)| \leq C_r, \quad |U(s, a)| \leq C_u \qquad (8)$$

*where $C_r$ and $C_u$ are positive constants.*

As a remark, the assumption of the bounded reward is common in the literature of inverse reinforcement learning and imitation learning [41, 22, 42, 43]. Moreover, the assumption of the bounded penalty function holds true for common choices of the uncertainty heuristics [37], such as the max aleatoric penalty and the ensemble variance penalty. Then we can show the following results.

**Lemma 2.** *Suppose Assumption 1 holds, then we obtain (where $C_v$ is a positive constant):*

$$|L(\theta) - \widehat{L}(\theta)| \leq \frac{\gamma C_v}{1 - \gamma} \cdot \mathbb{E}_{(s,a) \sim d^{\mathrm{E}}(\cdot, \cdot)}\big[\|P(\cdot|s, a) - \widehat{P}(\cdot|s, a)\|_1\big]. \qquad (9)$$

Please see Appendix E for the detailed proof. The above lemma suggests that the gap between the likelihood function and its surrogate version is bounded by the model mismatch error $\mathbb{E}_{(s,a) \sim d^{\mathrm{E}}(\cdot, \cdot)}\big[\|P(\cdot|s, a) - \widehat{P}(\cdot|s, a)\|_1\big]$. The fact that the objective approximation error $|L(\theta) - \widehat{L}(\theta)|$ depends on the model mismatch error evaluated in the expert-visited state-action distribution $d^{\mathrm{E}}(\cdot, \cdot)$ is crucial to the construction of the world model $\widehat{P}$. Based on Lemma 2, we understand that full data coverage on the joint state-action space $\mathcal{S} \times \mathcal{A}$ is *not* necessary. Instead, as long as the collected transition dataset $\mathcal{D} := \{(s, a, s')\}$ provides sufficient coverage on the expert-visited state-action space $\Omega := \{(s, a)|d^{\mathrm{E}}(s, a) > 0\}$, then the surrogate objective $\widehat{L}(\theta)$ will be an accurate approximation to the likelihood objective $L(\theta)$.

Intuitively, considering the goal is to recover a reward function to model expert behaviors which only lie in a quite limited region of the whole state-action space, data collection with full coverage can be redundant. This result is very useful in practice, since it serves to greatly reduce the efforts on data collection for constructing the world model. Moreover, it also matches recent theoretical understanding on offline reinforcement learning [44, 45, 46] and offline imitation learning [22], which show that it is enough to learn a good policy from offline data with partial coverage.

To analyze the sample complexity in the construction of the world model $\widehat{P}$, we quantitatively analyze the approximation error between $L(\theta)$ and $\widehat{L}(\theta)$. In discrete MDPs, the cardinalities of both state space and action space are finite ($|\mathcal{S}| < \infty$ and $|\mathcal{A}| < \infty$). Therefore, based on a collected transition dataset $\mathcal{D} = \{(s, a, s')\}$, we will use the empirical estimate to construct the world model $\widehat{P}$. Also recall that $\Omega := \{(s, a)|d^{\mathrm{E}}(s, a) > 0\}$ denotes the set of expert-visited state-action pairs. Define $\mathcal{S}^{\mathrm{E}} := \{s| \sum_{a \in \mathcal{A}} d^{\mathrm{E}}(s, a) > 0\} \subseteq \mathcal{S}$ as the set of expert-visited states. Using these definitions, we have the following result. The detailed proof is in Appendix H.

**Proposition 1.** *For any $\varepsilon \in (0, 2)$, suppose there are more than $N$ data points on each state-action pair $(s, a) \in \Omega$, and the total number of the collected transition samples satisfies:*

$$\#transition\ samples \geq |\Omega| \cdot N \geq \frac{c^2 \cdot |\Omega| \cdot |\mathcal{S}^{\mathrm{E}}|}{\varepsilon^2} \ln\Big(\frac{|\Omega|}{\delta}\Big)$$

*where $c$ is a constant dependent on $\delta$. With probability greater than $1 - \delta$, the following results hold:*

$$\mathbb{E}_{(s,a) \sim d^{\mathrm{E}}(\cdot, \cdot)}\big[\|P(\cdot|s, a) - \widehat{P}(\cdot|s, a)\|_1\big] \leq \varepsilon, \quad |L(\theta) - \widehat{L}(\theta)| \leq \frac{\gamma C_v}{1 - \gamma} \varepsilon. \qquad (10)$$

The above result estimates the total number of samples needed to construct the estimated world model $\widehat{P}$, so that the surrogate objective $\widehat{L}(\theta)$ can accurately approximate $L(\theta)$. A direct implication is that, the reward parameter obtained by solving $\widehat{L}(\cdot)$ also guarantees strong performance. To be more specific, define the optimal reward parameters associated with $L(\cdot)$ and $\widehat{L}(\cdot)$ as below, respectively:

$$\theta^* \in \arg\max_\theta \; L(\theta), \quad \hat{\theta} \in \arg\max_\theta \; \widehat{L}(\theta).$$

The next result characterizes the performance gap between the reward parameters $\hat{\theta}$ and $\theta^*$. The detailed proof is included in Appendix I.

**Theorem 1.** *For any $\varepsilon \in (0, \frac{4\gamma C_v}{1-\gamma})$, suppose there are more than $N$ data points on each state-action pair $(s,a) \in \Omega$ and the number of transition dataset $\mathcal{D}$ satisfies:*

$$\#\textit{transition samples} \geq |\Omega| \cdot N \geq \frac{4\gamma^2 \cdot C_v^2 \cdot c^2 \cdot |\Omega| \cdot |\mathcal{S}^{\mathrm{E}}|}{(1-\gamma)^2 \varepsilon^2} \ln\left(\frac{|\Omega|}{\delta}\right)$$

*where $c$ is a constant dependent on $\delta$. With probability greater than $1 - \delta$, the following result holds:*

$$L(\theta^*) - L(\hat{\theta}) \leq \varepsilon. \tag{11}$$

## 4 Algorithm design

In the following sections, we will design a computationally efficient algorithm to optimize $\widehat{L}(\cdot)$, and obtain its corresponding optimal reward parameter $\hat{\theta}$.

From the definition (7), it is clear that $\widehat{L}(\cdot)$ depends on the optimal soft value function $V_\theta(\cdot)$ in (3a), which in turn depends on the optimal policy $\pi_\theta$ as defined in (2b). Therefore, the surrogate objective maximization problem can be formulated as a bi-level optimization problem, expressed below, where the upper-level problem optimizes $\widehat{L}(\cdot)$ to search for a good reward estimate $\theta$, while the lower-level problem solves the optimal policy $\pi_\theta$ in a conservative MDP under the current reward estimate:

$$\max_\theta \; \widehat{L}(\theta), \quad \text{s.t.} \quad \pi_\theta := \arg\max_\pi \; \mathbb{E}_{\tau^{\mathrm{A}} \sim (\eta, \pi, \widehat{P})}\Big[\sum_{t=0}^\infty \gamma^t \Big(r(s_t, a_t; \theta) + U(s_t, a_t) + \mathcal{H}\big(\pi(\cdot|s_t)\big)\Big)\Big]. \tag{12}$$

In order to avoid the computational burden from repeatedly solving the optimal policy $\pi_\theta$ under each reward estimate $\theta$, we aim to design an algorithm which alternates between a policy optimization step and a reward update step. That is, at each iteration $k$, based on the current policy estimate $\pi_k$ and the reward parameter $\theta_k$, two steps will be performed consecutively: *(1)* the algorithm generates an updated policy $\pi_{k+1}$ through performing a conservative policy improvement step under the estimated world model $\widehat{P}$, and *(2)* it obtains an updated reward parameter $\theta_{k+1}$ through taking a reward update step. Next, we describe the proposed algorithm in detail.

**Policy Improvement Step.** Under the reward parameter $\theta_k$, we consider generating a new policy $\pi_{k+1}$ towards approaching the optimal policy $\pi_{\theta_k}$ as defined in (12). Similar to the definitions of $V_\theta$ and $Q_\theta$ in (3a) - (3b), under the current policy estimate $\pi_k$, the reward estimate $r(\cdot, \cdot; \theta_k)$ and the estimated world model $\widehat{P}$, we define the corresponding soft value function as $V_k(\cdot)$ and the soft Q-function as $Q_k(\cdot, \cdot)$. Please see (28a) - (28b) in Appendix for the precise definitions.

In order to perform a policy improvement step, we first approximate the soft Q-function by using an estimate $\widehat{Q}_k(s,a)$, which satisfies the following: (where $\epsilon_{\mathrm{app}} > 0$ is an approximation error)

$$\|\widehat{Q}_k - Q_k\|_\infty := \max_{s \in \mathcal{S}, a \in \mathcal{A}} |\widehat{Q}_k(s,a) - Q_k(s,a)| \leq \epsilon_{\mathrm{app}}. \tag{13}$$

With the approximator $\widehat{Q}_k$, an updated policy $\pi_{k+1}$ can be generated by a *soft policy iteration*:

$$\pi_{k+1}(a|s) \propto \exp\big(\widehat{Q}_k(s,a)\big), \quad \forall s \in \mathcal{S}, a \in \mathcal{A}. \tag{14}$$

As a remark, in practice, one can follow the popular reinforcement learning algorithms such as soft Q-learning [38] and soft Actor-Critic (SAC) [39] to obtain accurately approximated soft Q-function with low approximation error $\epsilon_{\mathrm{app}}$ (as outlined in (13)), so to achieve stable updates for the soft

**Algorithm 1** *A Model-based Approach for Offline Maximum Likelihood IRL (Offline ML-IRL)*

---

**Input:** Initialize reward parameter $\theta_0$ and policy $\pi_0$. Set the reward parameter's stepsize as $\alpha$.

Train the world model $\widehat{P}$ on the transition dataset $\mathcal{D}$.

Specify the penalty function $U(\cdot, \cdot)$ based on $\widehat{P}$.

**for** $k = 0, 1, \ldots, K - 1$ **do**

    **Policy Evaluation:** Approximate the soft Q-function $Q_k(\cdot, \cdot)$ by $\widehat{Q}_k(\cdot, \cdot)$

    **Policy Improvement:** $\pi_{k+1}(\cdot|s) \propto \exp\big(\widehat{Q}_k(s, \cdot)\big), \forall s \in \mathcal{S}$

    **Data Sampling I:** Sample an expert trajectory $\tau_k^{\mathrm{E}} := \{s_t, a_t\}_{t \geq 0}$

    **Data Sampling II:** Sample $\tau_k^{\mathrm{A}} := \{s_t, a_t\}_{t \geq 0}$ from $\pi_{k+1}$ and $\widehat{P}$

    **Estimating Gradient:** $g_k := h(\theta_k; \tau_k^{\mathrm{E}}) - h(\theta_k; \tau_k^{\mathrm{A}})$ where $h(\theta; \tau) := \sum_{t \geq 0} \gamma^t \nabla_\theta r(s_t, a_t; \theta)$

    **Reward Parameter Update:** $\theta_{k+1} := \theta_k + \alpha g_k$

**end for**

---

policy iteration in (14). In the literature of the model-based offline reinforcement learning, the state-of-the-art methods [25, 37, 47] also build their framework upon the implementation of SAC.

**Reward Optimization Step.** At each iteration $k$, given the current reward parameter $\theta_k$ and the updated policy $\pi_{k+1}$, we can update the reward parameter to $\theta_{k+1}$. First, let us compute the gradient of the surrogate objective $\nabla \widehat{L}(\theta_k)$. Please see Appendix F for the detailed proof.

**Lemma 3.** *The gradient of the surrogate objective $\widehat{L}(\theta)$ defined in (7), can be expressed as:*

$$\nabla \widehat{L}(\theta) = \mathbb{E}_{\tau^{\mathrm{E}} \sim (\eta, \pi^{\mathrm{E}}, P)}\Big[ \sum_{t=0}^{\infty} \gamma^t \nabla_\theta r(s_t, a_t; \theta) \Big] - \mathbb{E}_{\tau^{\mathrm{A}} \sim (\eta, \pi_\theta, \widehat{P})}\Big[ \sum_{t=0}^{\infty} \gamma^t \nabla_\theta r(s_t, a_t; \theta) \Big]. \quad (15)$$

In practice, we do not have access to the optimal policy $\pi_\theta$. This is due to the fact that repeatedly solving the underlying offline policy optimization problem under each reward parameter is computationally intractable. Therefore, at each iteration $k$, we construct an estimator of the exact gradient based on the current policy estimate $\pi_{k+1}$.

To be more specific, we take two approximation steps to develop a stochastic gradient estimator of $\nabla \widehat{L}(\theta)$: 1) choose one observed expert trajectory $\tau_k^{\mathrm{E}}$; 2) sample a trajectory $\tau_k^{\mathrm{A}}$ from the current policy estimate $\pi_{k+1}$ in the estimated world model $\widehat{P}$. Following these two approximation steps, the stochastic estimator $g_k$ which approximates the exact gradient $\nabla \widehat{L}(\theta_k)$ in (15) is defined as follows:

$$g_k := h(\theta_k; \tau_k^{\mathrm{E}}) - h(\theta_k; \tau_k^{\mathrm{A}}), \quad (16)$$

where $h(\theta; \tau) := \sum_{t=0}^{\infty} \gamma^t \nabla_\theta r(s_t, a_t; \theta)$ denotes the cumulative reward gradient under a trajectory.

Then we can update reward parameter according to the following update rule:

$$\theta_{k+1} = \theta_k + \alpha g_k \quad (17)$$

In Alg. 1, we summarize the proposed algorithm (named the Offline ML-IRL).

## 5 Convergence analysis

In this section, we present a theoretical analysis to show the finite-time convergence of Alg.1.

Before starting the analysis, let us point out the key challenges in analyzing the Alg. 1. Note that the algorithm relies on the updated policy $\pi_{k+1}$ to approximate the optimal policy $\pi_{\theta_k}$ at each iteration $k$. This coarse approximation can potentially lead to the distribution mismatch between the gradient estimator $g_k$ in (16) and the exact gradient $\nabla \widehat{L}(\theta_k)$ in (15). To maintain the stability of the proposed algorithm, we can use a relatively small stepsize $\alpha$ to ensure the policy estimates are updated in a faster time-scale compared with the reward parameter $\theta$. This allows the policy estimates $\{\pi_{k+1}\}_{k \geq 0}$ to closely track the optimal solutions $\{\pi_{\theta_k}\}_{k \geq 0}$ in the long run.

To proceed, let us introduce a few assumptions.

**Assumption 2** (Ergodic Dynamics). *Given any policy $\pi$, the Markov chain under the estimated world model $\widehat{P}$ is irreducible and aperiodic. There exist constants $\kappa > 0$ and $\rho \in (0,1)$ to ensure:*

$$\max_{s \in \mathcal{S}} \|\widehat{P}(s_t \in \cdot|s_0 = s, \pi) - \mu_{\widehat{P}}^\pi(\cdot)\|_{\text{TV}} \leq \kappa \rho^t, \quad \forall\, t \geq 0$$

*where $\|\cdot\|_{\text{TV}}$ denotes the total variation (TV) norm; $\mu_{\widehat{P}}^\pi$ is the stationary distribution of visited states under the policy $\pi$ and the world model $\widehat{P}$.*

The assumption about the ergodic dynamics is common in the literature of reinforcement learning [48, 49, 21, 31, 50], which ensures the Markov chain mixes at a geometric rate.

**Assumption 3** (Lipschitz Reward). *Under any reward parameter $\theta$, the following conditions hold for any $s \in \mathcal{S}$ and $a \in \mathcal{A}$:*

$$\big\|\nabla_\theta r(s, a; \theta)\big\| \leq L_r, \quad \big\|\nabla_\theta r(s, a; \theta_1) - \nabla_\theta r(s, a; \theta_2)\big\| \leq L_g \|\theta_1 - \theta_2\|, \tag{18}$$

*where $L_r$ and $L_g$ are positive constants.*

According to Assumption 3, the parameterized reward has bounded gradient and is Lipschitz smooth. This assumption is common for min-max / bi-level optimization problems [41, 51, 52, 31, 50].

Based on Assumptions 2 - 3, we show that certain Lipschitz properties of the optimal soft Q-function and the surrogate objective hold. Please see the detailed proof in Appendix G.

**Lemma 4.** *Suppose Assumptions 2 - 3 hold. Under any reward parameter $\theta_1$ and $\theta_2$, the optimal soft Q-function and the surrogate objective satisfy the following Lipschitz properties:*

$$|Q_{\theta_1}(s, a) - Q_{\theta_2}(s, a)| \leq L_q \|\theta_1 - \theta_2\|, \; \forall s \in \mathcal{S}, a \in \mathcal{A} \tag{19a}$$

$$\|\nabla\widehat{L}(\theta_1) - \nabla\widehat{L}(\theta_2)\| \leq L_c \|\theta_1 - \theta_2\| \tag{19b}$$

*where $L_q$ and $L_c$ are positive constants.*

Our main convergence result is summarized in the following theorem, which characterizes the convergence speed of the estimates $\{\pi_{k+1}\}_{k \geq 0}$ and $\{\theta_k\}_{k \geq 0}$. The detailed proof is in Appendix J.

**Theorem 2** (Convergence Analysis). *Suppose Assumptions 2 - 3 hold. Let $K$ denote the total number of iterations to be run in Alg. 1. Setting the stepsize as $\alpha = \alpha_0 \cdot K^{-\frac{1}{2}}$ where $\alpha_0 > 0$, we obtain the following convergence results:*

$$\frac{1}{K} \sum_{k=0}^{K-1} \mathbb{E}\Big[\big\| \log \pi_{k+1} - \log \pi_{\theta_k} \big\|_\infty\Big] = \mathcal{O}(K^{-\frac{1}{2}}) + \mathcal{O}(\epsilon_{\text{app}}) \tag{20a}$$

$$\frac{1}{K} \sum_{k=0}^{K-1} \mathbb{E}\Big[\|\nabla\widehat{L}(\theta_k)\|^2\Big] = \mathcal{O}(K^{-\frac{1}{2}}) + \mathcal{O}(\epsilon_{\text{app}}) \tag{20b}$$

*where we have defined $\| \log \pi_{k+1} - \log \pi_{\theta_k} \|_\infty := \max_{s \in \mathcal{S}, a \in \mathcal{A}} \big| \log \pi_{k+1}(a|s) - \log \pi_{\theta_k}(a|s)\big|$ and $\epsilon_{\text{app}}$ is the approximation error defined in (13).*

In Theorem 2, we demonstrated that Alg.1 identifies the approximate stationary solution of the surrogate problem (12) in finite time. Next, we show that when the reward is parameterized *linearly*, an improved result can be obtained, where the stationary solutions of problem (12) can be connected to the optimal solutions of the offline-IRL problem (12). See Appendix K for detailed proof.

**Theorem 3** (Optimality Guarantee). *Assume that the reward function is linearly parameterized, i.e., $r(s, a; \theta) := \phi(s, a)^\top \theta$ where $\phi(s, a)$ is the feature vector of the state-action pair $(s, a)$. Then any stationary point of the surrogate problem (12) is a global optimum. Furthermore, for any $\varepsilon \in (0, \frac{4\gamma C_v}{1-\gamma})$, suppose there are more than $N$ data points on each state-action pair $(s, a) \in \Omega$ and the number of transition dataset $\mathcal{D}$ satisfies:*

$$\#\textit{transition samples} \geq |\Omega| \cdot N \geq \frac{4\gamma^2 \cdot C_v^2 \cdot c^2 \cdot |\Omega| \cdot |\mathcal{S}^{\text{E}}|}{(1-\gamma)^2 \varepsilon^2} \ln\Big(\frac{|\Omega|}{\delta}\Big)$$

*where $c$ is a constant dependent on $\delta$. With probability greater than $1 - \delta$, any stationary point $\tilde{\theta}$ of the surrogate objective $\widehat{L}(\cdot)$ is an epsilon-optimal solution to the maximum likelihood problem (2):*

$$L(\theta^*) - L(\tilde{\theta}) \leq \varepsilon \tag{21}$$

*where $\theta^*$ is defined as the optimal reward parameter of the log-likelihood objective $L(\cdot)$.*

As a remark, the epsilon-optimal solution on the MLE problem implies:

$$L(\theta^*) - L(\tilde{\theta}) = \frac{1}{1-\gamma}\mathbb{E}_{s\sim d^{\mathrm{E}}(\cdot),a\sim\pi^{\mathrm{E}}(\cdot|s)}\big[\log\big(\frac{\pi_{\theta^*}(a|s)}{\pi_{\tilde{\theta}}(a|s)}\big)\big] \leq \varepsilon.$$

When the expert trajectories are consistent with the optimal policy under a ground truth reward parameter $\theta^*$, we have $\pi^{\mathrm{E}} = \pi_{\theta^*}$. Due to this property, we can obtain that

$$L(\theta^*) - L(\tilde{\theta}) = \frac{1}{1-\gamma}\mathbb{E}_{s\sim d^{\mathrm{E}}(\cdot),a\sim\pi^{\mathrm{E}}(\cdot|s)}\big[\log\big(\frac{\pi^{\mathrm{E}}(a|s)}{\pi_{\tilde{\theta}}(a|s)}\big)\big] = \frac{1}{1-\gamma}\mathbb{E}_{s\sim d^{\mathrm{E}}(\cdot)}\big[D_{KL}\big(\pi^{\mathrm{E}}(\cdot|s)||\pi_{\tilde{\theta}}(\cdot|s)\big)\big] \leq \varepsilon.$$

Hence, Theorem 3 also provides a formal guarantee that the recovered policy $\pi_{\tilde{\theta}}$ is $\epsilon$-close to the expert policy $\pi^{\mathrm{E}}$ measured by the KL divergence.

We remark that even under the linear parameterization assumption, showing the optimality of the stationary solution for the surrogate problem (12) is still non-trivial, since the problem is still not a concave problem with respect to $\theta$. In our analysis, we translate this problem to a certain saddle point problem. Under linear reward parameterization, we show that any stationary solution $\tilde{\theta}$ of the surrogate problem (12) together with the corresponding optimal policy $\pi_{\tilde{\theta}}$ consist of a saddle point to the saddle point problem. By further leveraging the property of the saddle point, we show the optimality of the stationary solution for the surrogate problem. Finally, by utilizing the statistical guarantee in Theorem 1, we obtain the performance guarantee for any stationary point $\tilde{\theta}$ in (21).

## 6 Numerical results

In this section, we present numerical results for the proposed algorithm. More specifically, we intend to address the following questions: 1) How does the proposed algorithm compare with other state-of-the-art methods? 2) Whether offline ML-IRL can recover a high-quality reward estimator of the expert, in the sense that the reward can be used across different datasets or environment?

We compare the proposed method with several benchmarks. Two classes of the existing algorithms are considered as baselines: 1) state-of-the-art offline IRL algorithms which are designed to recover the ground-truth reward and expert policy from demonstrations, including a model-based approach CLARE [15] and a model-free approach IQ-Learn [14]; 2) imitation learning algorithms which only learn a policy to mimic the expert beahviors, including behavior cloning (BC) and ValueDICE [28].

We test the performance of the proposed Offline ML-IRL on a diverse collection of robotics training tasks in MuJoCo simulator [53], as well as datasets in D4RL benchmark [54], which include three environments (halfcheetah, hopper and walker2d) and three dataset types (medium-replay, medium, and medium-expert). In each experiment set, both environment interactions and the ground-truth reward are not accessible. Moreover, we train the algorithm until convergence and record the average reward of the episodes over 6 random seeds.

In Offline ML-IRL, we estimate the dynamics model by neural networks which model the location of the next state by Gaussian distributions. Here, we independently train an ensemble of $N$ estimated world model $\{\widehat{P}^i_{\phi,\varphi}(s_{t+1}|s_t, a_t) = \mathcal{N}(\mu^i_\phi(s_t, a_t), \Sigma^i_\varphi(s_t, a_t))\}^N_{i=1}$ via likelihood maximization over transition samples. Then we can quantify the model uncertainty and construct the penalty function. For example, in [25], the aleatoric uncertainty is considered and the penalty function is constructed as $U(s, a) = -\max_{i=1,\cdots N} \|\Sigma^i_\varphi(s, a))\|_{\mathrm{F}}$. For the offline RL subrounte in (13) - (14), we follow the implementation of MOPO [25]. We follow the same setup in [37] to select the key hyperparameters for MOPO. We parameterize the reward function by a three-layer neural network. The reward parameter and the policy are updated alternatingly, where the former is updated accoding to (17), while the latter is optimized according to MOPO. More experiment details are in Appendix A.

In Figure 2 and Table 1, we show the performance comparison between the proposed algorithm and benchmarks. The results of IQ-Learn is not presented since it suffers from unstable performance. To obtain better performance in BC and ValueDICE, we only use the expert demonstrations to train their agents. From the results, it is clear that the proposed Offline ML-IRL outperforms the benchmark algorithms by a large margin in most cases. In the experiment sets where Offline ML-IRL does not obtain the best performance, we notice the performance gap is small compared with the leading benchmark algorithm. Moreover, among the three dataset types (medium-replay, medium and medium-expert), medium-expert dataset has the most complete coverage on the expert-visited

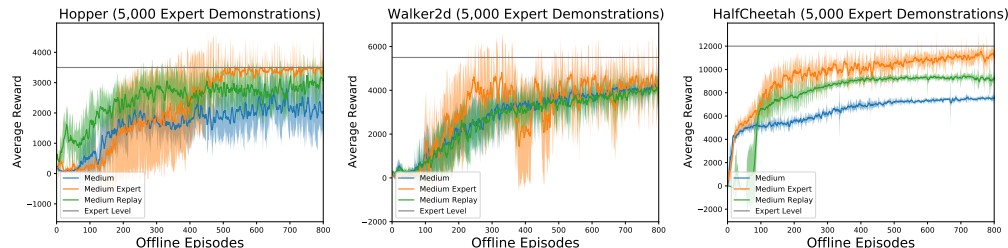

Figure 2: The performance of Offline ML-IRL given 5,000 expert demonstrations.

| Dataset type | Environment | Offline ML-IRL | BC | ValueDICE | CLARE | Expert Performance |
|---|---|---|---|---|---|---|
| medium | hopper | $2453.25 \pm 717.30$ | $2801.19 \pm 330.88$ | $\mathbf{3073.16 \pm 538.67}$ | $3015.37 \pm 474.38$ | $3512.09 \pm 21.65$ |
| medium | halfcheetah | $\mathbf{7640.73 \pm 195.00}$ | $4471.72 \pm 2835.55$ | $1125.17 \pm 959.45$ | $841.46 \pm 344.06$ | $12174.61 \pm 91.45$ |
| medium | walker2d | $\mathbf{3989.20 \pm 487.82}$ | $2328.75 \pm 906.96$ | $3191.47 \pm 1887.90$ | $237.49 \pm 160.62$ | $5383.98 \pm 52.15$ |
| medium-replay | hopper | $3046.36 \pm 429.25$ | $2801.19 \pm 330.88$ | $\mathbf{3073.16 \pm 538.67}$ | $2888.04 \pm 844.48$ | $3512.09 \pm 21.65$ |
| medium-replay | halfcheetah | $\mathbf{9236.84 \pm 309.10}$ | $4471.72 \pm 2835.55$ | $1125.17 \pm 959.45$ | $437.18 \pm 182.99$ | $12174.61 \pm 91.45$ |
| medium-replay | walker2d | $\mathbf{3995.32 \pm 487.82}$ | $2328.75 \pm 906.96$ | $3191.47 \pm 1887.90$ | $291.71 \pm 75.66$ | $5383.98 \pm 52.15$ |
| medium-exp | hopper | $3347.11 \pm 238.18$ | $2801.19 \pm 330.88$ | $3073.16 \pm 538.67$ | $\mathbf{3350.47 \pm 245.78}$ | $3512.09 \pm 21.65$ |
| medium-exp | halfcheetah | $\mathbf{11231.40 \pm 585.21}$ | $4471.72 \pm 2835.55$ | $1125.17 \pm 959.45$ | $622.79 \pm 56.46$ | $12174.61 \pm 91.45$ |
| medium-exp | walker2d | $\mathbf{4201.40 \pm 637.99}$ | $2328.75 \pm 906.96$ | $3191.47 \pm 1887.90$ | $959.50 \pm 470.64$ | $5383.98 \pm 52.15$ |

Table 1: **MuJoCo Results.** The performance versus different datasets and $5,000$ expert demonstrations. The bolded numbers are the best ones for each data set, among Offline ML-IRL, BC, ValueDICE, and CLARE.

state-action space. As a result, Offline ML-IRL can nearly match the expert performance when the world model is trained from the medium-expert transition dataset. This numerical result matches our theoretical understanding in Theorem 1. As a remark, in most cases, we notice that Offline ML-IRL outperforms CLARE by a large margin which is also a model-based offline IRL algorithm. As we discussed in the related work (Appendix B), this is due to the fact that when the expert demonstrations are limited and diverse transition samples take a large portion of all collected data, CLARE learns a policy to match the joint visitation measure of all collected data, thus fails to imitate expert behaviors.

In Appendix A, we further present an experiment showcasing the quality of the reward recovered by the Offline ML-IRL algorithm. Towards this end, we use the recovered reward from the medium-expert dataset to estimate the reward value for all transition samples $(s, a, s')$ in the medium-replay dataset. With the recovered reward function $r(\cdot, \cdot; \theta)$ and given those transition samples with estimated reward labels $\{(s, a, r(s, a; \theta), s')\}$, we run MOPO to solve Offline RL tasks given these transition samples with estiamted rewards. In Fig. 5, we can infer that the Offline ML-IRL can recover high-quality reward functions, since the recovered reward can label transition samples for solving offline RL tasks. Comparing with the numerical results of directly doing offline-IRL on the respective datasets in Table 1, we show that solving Offline RL by using transferred reward function and unlabelled transition datasts can achieve similar performance in Fig. 5.

# 7 Conclusion

In this paper, we model the offline Inverse Reinforcement Learning (IRL) problem from a maximum likelihood estimation perspective. Through constructing a generative world model to estimate the environment dynamics, we solve the offline IRL problem through a model-based approach. We develop a computationally-efficient algorithm that effectively recovers the underlying reward function and its associated optimal policy. We have also established statistical and computational guarantees for the performance of the recovered reward estimator. Through extensive experiments, we demonstrate that our algorithm outperforms existing benchmarks for offline IRL and Imitation Learning, especially on high-dimensional robotics control tasks. One limitation of our method is that we focus solely on aligning with expert demonstrations during the reward learning process. In an ideal scenario, reward learning should incorporate diverse metrics and data sources, such as expert demonstrations and preferences gathered through human feedback. One direction for future work is to broaden our algorithm framework and theoretical analysis for a wider scope in reward learning.

## Acknowledgments

M. Hong and S. Zeng are supported by NSF grant CIF-1910385.

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

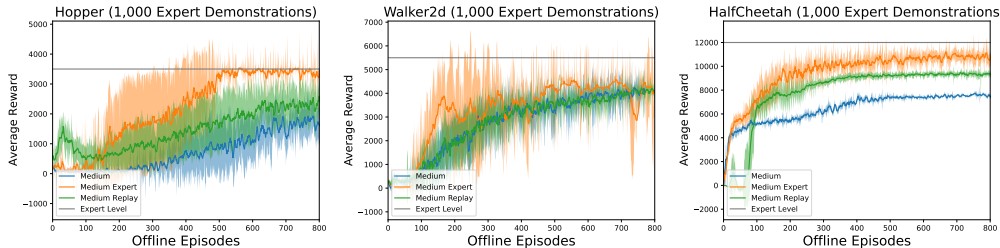

Figure 3: The performance of Offline ML-IRL in different environments given $1,000$ expert demonstrations

| Dataset type | Environment | Offline ML-IRL | BC | ValueDICE | CLARE | Expert Performance |
|---|---|---|---|---|---|---|
| medium | hopper | $1750.59 \pm 507.06$ | $843.59 \pm 503.83$ | $\mathbf{2417.83 \pm 1258.30}$ | $1559.00 \pm 324.10$ | $3533.90$ |
| medium | halfcheetah | $\mathbf{7690.30 \pm 119.17}$ | $2799.40 \pm 1046.00$ | $-175.87 \pm 343.57$ | $240.50 \pm 143.25$ | $12107.51$ |
| medium | walker2d | $\mathbf{4121.68 \pm 291.86}$ | $1248.13 \pm 72.89$ | $1794.97 \pm 1400.70$ | $320.51 \pm 93.07$ | $5284.33$ |
| medium-replay | hopper | $2395.03 \pm 593.16$ | $843.59 \pm 503.83$ | $\mathbf{2417.83 \pm 1258.30}$ | $2369.79 \pm 204.23$ | $3533.90$ |
| medium-replay | halfcheetah | $\mathbf{9313.29 \pm 261.94}$ | $2799.40 \pm 1046.00$ | $-175.87 \pm 343.57$ | $343.37 \pm 108.00$ | $12107.51$ |
| medium-replay | walker2d | $\mathbf{4100.99 \pm 293.88}$ | $1248.13 \pm 72.89$ | $1794.97 \pm 1400.70$ | $440.78 \pm 88.05$ | $5284.33$ |
| medium-expert | hopper | $\mathbf{3366.23 \pm 229.56}$ | $843.59 \pm 503.83$ | $2417.83 \pm 1258.30$ | $3071.31 \pm 186.37$ | $3533.90$ |
| medium-expert | halfcheetah | $\mathbf{10812.15 \pm 551.38}$ | $2799.40 \pm 1046.00$ | $-175.87 \pm 343.57$ | $292.95 \pm 84.86$ | $12107.51$ |
| medium-expert | walker2d | $\mathbf{4049.43 \pm 1046.61}$ | $1248.13 \pm 72.89$ | $1794.97 \pm 1400.70$ | $548.59 \pm 146.09$ | $5284.33$ |

Table 2: **MuJoCo Results.** The performance versus different datasets and $1,000$ expert demonstrations. Here, the expert demonstration dataset only includes a single expert trajectory ($1,000$ expert transition samples) and the value of expert performance corresponds to the cumulative reward value of the provided expert trajectory.

# Appendix

# Limitations and broader impacts

Offline inverse reinforcement learning is a method designed to recover the reward function and the associated optimal policy from an observed expert dataset. However, potential negative societal impacts may arise if the demonstration dataset incorporates low-quality data. For safety-critical applications, such as autonomous driving and clinical decision support, we need to exercise particular caution to avoid the introduction of detrimental biases from the demonstration dataset. Ensuring safe adaptation is vital for real-world applications.

One limitation of our method is that we exclusively use expert demonstration in the process of reward learning. Ideally, reward learning should incorporate varied data sources, like expert demonstrations and preferences (pairwise comparisions). A possible direction for future work is to broaden our algorithm and theoretical analysis for a wider scope in reward learning. This would allow us to develop a more robust reward model through the integration of diverse data inputs.

# A   Experiment details

In this section, we provide implementation details and additional numerical results.

## A.1   Detailed experiment setting

In our experiment, we use two kinds of datasets: 1) transition dataset $\mathcal{D} = \{(s, a, s')\}$ which includes diverse transition samples and is downloaded from D4RL V2; 2) expert demonstration dataset $\mathcal{D}^{\mathrm{E}} = \{\tau^{\mathrm{E}}\}$ which consists of several expert trajectories and the expert trajectories are collected from an expert-level policy. All evaluations are based on a single NVIDIA GeForce RTX 2080 Ti.

**The transition dataset.** The transition datasets are downloaded from D4RL V2 and the transition datasets for each specific task have three different type: medium, medium-replay and medium-expert. In D4RL V2, the datasets are generated as follows: **medium**: use SAC to train a policy with medium-level performance, then use it to collected 1 million transition samples; **medium-replay**: use SAC to train a policy until an environment-specific performance is obtained, then save all transition samples in the replay buffer; **medium-expert**: combine 1 million transition samples collected from a

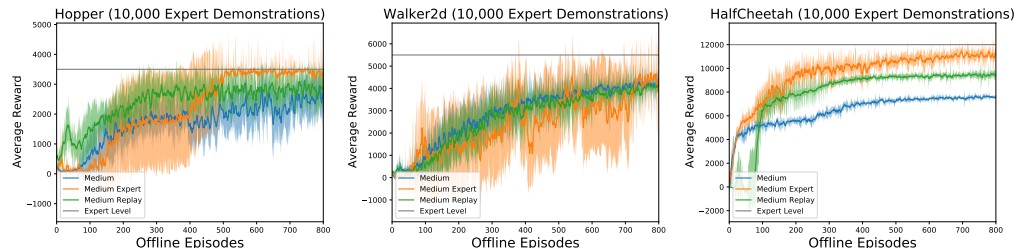

Figure 4: The performance of Offline ML-IRL in different environments given $10,000$ expert demonstrations

| Dataset type | Environment | Offline ML-IRL | BC | ValueDICE | CLARE | Expert Performance |
|---|---|---|---|---|---|---|
| medium | hopper | $2461.45 \pm 705.70$ | $3236.70 \pm 45.97$ | $\mathbf{3442.82 \pm 199.21}$ | $3357.52 \pm 270.45$ | $3512.64 \pm 17.10$ |
| medium | halfcheetah | $\mathbf{7706.43 \pm 159.39}$ | $4935.60 \pm 2835.55$ | $3248.60 \pm 2001.05$ | $841.46 \pm 344.06$ | $12156.16 \pm 88.01$ |
| medium | walker2d | $\mathbf{4195.36 \pm 352.86}$ | $2822.56 \pm 978.97$ | $3046.76 \pm 2001.28$ | $825.15 \pm 738.33$ | $5365.62 \pm 55.79$ |
| medium-replay | hopper | $2889.73 \pm 542.65$ | $3236.70 \pm 45.97$ | $\mathbf{3442.82 \pm 199.21}$ | $3139.98 \pm 478.75$ | $3512.64 \pm 17.10$ |
| medium-replay | halfcheetah | $\mathbf{9383.34 \pm 358.67}$ | $4935.60 \pm 2835.55$ | $3248.60 \pm 2001.05$ | $437.18 \pm 182.99$ | $12156.16 \pm 88.01$ |
| medium-replay | walker2d | $\mathbf{4092.58 \pm 308.71}$ | $2822.56 \pm 978.97$ | $3046.76 \pm 2001.28$ | $869.07 \pm 612.56$ | $5365.62 \pm 55.79$ |
| medium-expert | hopper | $3350.79 \pm 264.96$ | $3236.70 \pm 45.97$ | $\mathbf{3442.82 \pm 199.21}$ | $3166.69 \pm 512.77$ | $3512.64 \pm 17.10$ |
| medium-expert | halfcheetah | $\mathbf{11276.09 \pm 551.94}$ | $4935.60 \pm 2835.55$ | $3248.60 \pm 2001.05$ | $2020.51 \pm 520.46$ | $12156.16 \pm 88.01$ |
| medium-expert | walker2d | $\mathbf{4363.54 \pm 729.60}$ | $2822.56 \pm 978.97$ | $3046.76 \pm 2001.28$ | $3245.79 \pm 1911.56$ | $5365.62 \pm 55.79$ |

Table 3: **MuJoCo Results.** The performance versus different datasets and $10,000$ expert demonstrations.

medium-level policy with another 1 million transition samples collected from an expert-level policy. As a remark, since we are considering the setting of offline IRL where the ground-truth reward is not accessible, we hide the reward information of those downloaded transition datasets from D4RL V2.

**The expert demonstration dataset.** The expert demonstration dataset includes the transition samples in several collected expert trajectories. Here, we first train a reinforcement learning agent by SAC under the ground-truth reward function to achieve expert-level performance. Then we save the well-train expert-level policy to collect expert trajectories. For each trajectory, it includes $1,000$ consecutive transition samples $(s, a, s')$ in one episode.

In the model-based algorithms like Offline ML-IRL and CLARE, the estimated dynamic models are trained using the transition dataset. After the estimated dynamics model is constructed, the corresponding algorithms (Offline ML-IRL and CLARE) will further utilize the expert trajectories in the expert demonstration dataset $\mathcal{D}^{\mathrm{E}}$ to recover the ground-truth reward function and imitate the expert behaviors.

For model-free offline imitation learning algorithms like BC and ValueDICE, they directly learn a policy to imitate the expert behaviors. Hence, those model-free offline imitation learning algorithms (BC and ValueDICE) will only utilize the expert demonstration dataset $D^{\mathrm{E}}$. Due to the fact that BC and ValueDICE do not use the transition dataset $\mathcal{D}$, their recorded performance in Table 1 - 3 is not related to the type of the transition dataset $\mathcal{D}$.

In our implementation of Offline ML-IRL, we parameterize the reward network by a $(256, 256)$ MLP with ReLU activation function. The input of the reward network is the state-action pair $(s, a)$ and the output is the estimated reward value $r(s, a; \theta)$. Moreover, we use Adam as the optimizer and the stepsize to update the reward network is set to be $1 \times 10^{-4}$. For the policy optimization subroutine (13) - (14), we consider it as a model-based offline RL subtask. Since it is under an entropy-regularized framework, SAC-based algorithm is used as the corresponding RL solver. More specifically, we use model-based offline policy optimization (MOPO) [25] in the RL subroutine (13) - (14). For the implementation of MOPO and the corresponding hyperparameters, we follow the setup provided in [37] which guarantees strong performance through fine-tuning the key hyperparameters of MOPO, including the number of estimated world models, the choice of the penalty function $U(\cdot, \cdot)$, the penalty coefficient and the imaginary rollout length in the estimated world model. The official code base of [37] is available in `https://openreview.net/forum?id=zz9hXVhf40`. During the training of Offline ML-IRL, each time we implement a policy improvement subroutine under the current reward estimator, we update the agent by running MOPO for 20 offline episodes in the estimated world model and under the current reward estimator. After that, we sample expert trajectory

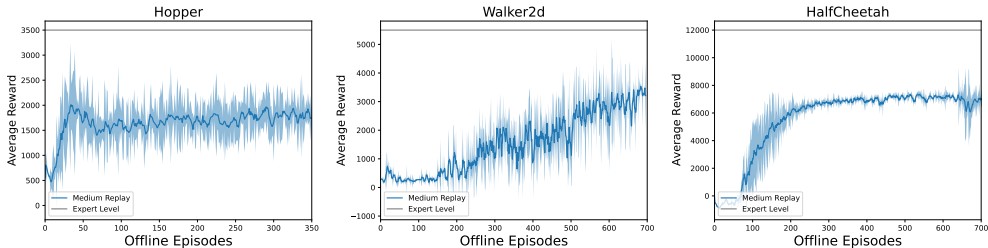

Figure 5: **Reward Transfer.** The recovered reward by Offline ML-IRL in the medium-expert dataset is transferred to the medium-replay datasets for solving offline RL tasks.

from the expert demonstration dataset $\mathcal{D}^{\mathrm{E}} = \{\tau^{\mathrm{E}}\}$, and sample agent trajectory from the estimated world model $\widehat{P}$ to construct the stochastic reward gradient estimator following the expressing given in (16). Then we are able to update the reward network by a stochastic gradient step.

For the benchmark algorithms, the official code base of CLARE is provided in `https://openreview.net/forum?id=5aT4ganOd98`. Moreover, the official implementation of ValueDICE is provided in `https://github.com/google-research/google-research/tree/master/value_dice`.

### A.2 Additional numerical results

In Fig. 3 - 4, we show the convergence curves of Offline ML-IRL when expert demonstration datasets $D^{\mathrm{E}}$ include $1,000$ and $10,000$ expert demonstrations (1 and 10 expert trajectories) respectively. Moreover, we compare the performance with benchmark algorithms in Table 2 - 3. According to the numerical results, we show that our proposed method Offline ML-IRL is insensitive to the number of expert demonstrations. Even when only $1,000$ expert demonstrations are provided, Offline ML-IRL can achieve strong performance (close to the expert-level performance with the medium-expert dataset).

In Fig. 5, we show the numerical results of reward transfer experiment. Here, we first use the medium-expert dataset and expert demonstration dataset to generate a reward estimator by running Offline ML-IRL. Then we transfer the recovered reward function $r(\cdot, \cdot; \theta)$ to the medium-replay dataset for labelling all transition samples with the estimated reward value. With the transferred reward function $r(\cdot, \cdot; \theta)$ to label the transition dataset as $\{(s, a, r(s, a; \theta), s')\}$, we can treat the problem as an Offline RL task and run MOPO to solve it. The numerical result is shown in Fig. 5. Comparing Fig. 5 with the results in Fig 4 which are obtained by directly running Offline ML-IRL with expert demonstrations / trajectories, similar performance can be obtained by run Offline RL on the transition datset $\mathcal{D}$ with the transferred reward estimator. These results suggest that Offline ML-IRL can recover high-quality reward estimator, which can be transferred across different datasets to label those unlabeled data with the estimated reward value.

## B Related work

Inverse reinforcement learning (IRL) consists of estimating the reward function and the optimal policy that best fits the expert demonstrations [16, 17, 18, 20, 31]. In the seminal work [18], a formulation for IRL is proposed based on the principle of maximum entropy which enjoys the theoretical guarantees to recover the reward function [29, 30, 55, 56]. Furthermore, in [57, 20, 21], a series of sample-efficient algorithms are proposed to solve the the maximum entropy IRL formulation. Moreover, there still exists one major limitation in IRL, due to the fact that most existing IRL methods require extensive online trials and errors which could be impractical in real-life applications.

To side-step the online interactions with the environment, offline IRL is considered to recover the reward function from fixed datasets of expert demonstrations. In [13], by taking into account the bias of expert demonstrations, the authors propose a gradient-based IRL methods to jointly estimate the reward function and the transition dynamics. In [14], a model-free algorithm called IQ-Learn is proposed by implicitly representing the reward function and the policy from a soft Q-function.

Although avoiding the online interactions with the environment, IQ-Learn sacrifices the accuracy of the estimated reward function, since its recovered reward is highly dependent on the environment dynamics. In [9], the authors propose a variational Bayesian framework to estimate the approximate posterior distribution of the reward function from a collected demonstration dataset. Recently, in [15], a model-based offline IRL approach (called CLARE) is proposed, which implements IRL algorithm in an estimated dynamics model to learn the reward. To avoid distribution shift, CLARE incorporates conservatism into its estimated reward to ensure the corresponding policy generates a state-action visitation measure to match the joint data distribution of collected transition samples and expert demonstrations. However, when the number of expert demonstrations is limited and/or most of the transition samples about the state-action-next state observations are collected from a low-quality behavior policy, matching the empirical state-action visitation measure of all collected data will force the recovered reward / policy to mimic the low-quality behavior policy, which is not enough to guarantee an accurate model of the expert. Moreover, matching the visitation measure is sensitive to the quality of the estimated dynamics model. Matching the visitation measure in an estimated dynamics model with poor prediction quality cannot guarantee high-quality recovered reward.

Here, we discuss the literature in several closely related areas as below.

**Online IRL.** [18] is a seminal work in which the expert policy is formulated as the model that maximizes entropy subject to a constraint requiring that the expected features under such policy match the empirical averages in the expert's observation dataset. [29, 19] propose algorithms with nested loop structure to solve such maximum entropy estimation problem. Those algorithms with a nested loop structure can suffer from computational burden, because they need to alternate between an outer loop with a reward update step and an inner loop that calculates the explicit policy estimates. In [58], the authors propose a method to estimate both reward functions and constraints from a group of experts. In [21], the authors propose an online IRL formulation based on maximum likelihood estimation perspective, which needs extensive interaction with the environment in order to learn the reward function. When the reward function is linearly parameterized, it is shown that the maximum likelihood IRL formulation is the dual problem to the classic maximum entropy IRL problem. Moreover, a theoretical analysis is provided to ensure the reward parameter can converge to a stationary solution in finite time. Despite the computational efficiency obtained in [21], it is not clear whether an optimal reward estimator can be recovered from the maximum likelihood formulation. As a remark, compared with the online IRL work [21], our paper extends the maximum likelihood formulation to the setting of offline IRL. We further provide a statistical guarantee to show that when the transition dataset has sufficient coverage over the expert-visited state-action space, an estimated world model can be constructed so that the optimal reward estimator can be recovered. Moreover, we provide a computational guarantee to show that the proposed algorithm can identify a stationary point in finite time. When the reward function is linearly parameterized, we show the optimal reward estimator of the maximum likelihood estimation formulation can be recovered by the proposed algorithm.

**Offline IRL.** In [59], the authors propose a method to perform IRL through constructing a linearly parameterized score function-based multi-class classification algorithm. With an estimated of the feature expectation of the expert, the proposed algorithm is able to avoid direct RL subroutine. Given appropriate heuristic and expert trajectories, the proposed algorithm could recover the underlying reward function without any online environment interactions. In [60], the authors propose a model-free method to construct the Deep Successor Feature Networks (DSFN) to estimate the feature expectations in an off-policy setting. By parameterizing the reward as a linear function and estimating the feature expectation, the offline IRL can be solved computation-friendly with limited expert demonstrations. In [61], a modular algorithm called Offline Reinforcement Imitation Learning (ORIL) is proposed. In ORIL, a reward function is first constructed by constrasting expert demonstrations with other transition samples $\{(s, a, s')\}$ which are sampled from a behavior policy with unknown quality. With the constructed reward function, the quality of all unlabeled data can be evaluated and then an agent is trained via offline reinforcement learning. In [62], the authors proposed an IRL method which leverages multi-task RL pre-training and successor features to train IRL from past experience. When a set of related tasks can be provided for RL pre-training and the feature expectation is estimated accurately, the proposed method could benefit from the multi-task pre-training. However, in practice, it is difficult to obtain full knowledge of the feature structure in the ground-truth reward function.

**Offline Imitation Learning.** Different from offline IRL, offline imitation learning aims to directly imitate the expert behavior by learning a policy in the offline setting. In [28], an algorithm called

ValueDICE is proposed to leverage off-policy data to learn an imitation policy with the use of any explicit reward functions. Moreover, the numerical results of ValueDICE show that it could be easily implemented in the offline regime, where additional interactions with the environment is not allowed. In [63], the authors propose method to learn both policies and environment and analyze the corresponding error bounds. In [64], a model-free offline imitation learning algorithm is proposed by energy-based distribution matching (EDM). EDM provides an effective way to minimize the divergence between the state-action visitation measure of the demonstrator and the imitation policy. In [65], the authors develop a variational model-based adversarial imitation learning (V-MAIL) algorithm for learning from visual demonstrations. By constructing a variational latent-space dynamics model, V-MAIL is able to solve high-dimensional visual tasks without any additional environment interactions. In [22], the authors introduce Model-based Imitation Learning from Offline data (MILO), which extends model-based offline reinforcement learning to imitation learning. The theoretical analysis of MILO shows that full coverage of the offline data is not necessary. When the offline dataset provides sufficient coverage to cover the expert-visited state-actions, MILO can provably avoid distribution shift in offline imitation learning by leveraging a constructed dynamics model.

**Offline RL.** Offline RL considers the problem of learning a policy from a fixed datasets where the reward value is provided for each collected transition samples. For model-free offline RL algorithms, a world model is not estimated and the algorithms directly learn a policy from the collected dataset. In [66, 67], model-free offline RL algorithms are proposed to solve the importance sampling problem. In [23, 68], conservatism is incorporated into the value function to avoid overestimation in the offline RL setting. For the model-based offline RL algorithms, [24] constructs the estimated world model and sets hard threshold on the model uncertainty for constructing terminating states to avoid dangerous explorations. In [25], the authors proposes a model-based offline policy optimization algorithm (MOPO) which utilizes uncertainty estimation techniques to construct a penalty function to regularize the reward function. Therefore, MOPO can learn a conservative policy which stays in the low-uncertainty region to aviod the distribution shift issue. As a follow-up work, [37] revisits the design choices of several key hyperparameters in MOPO and fine-tune the corresponding hyperparameters in MOPO to guanrantee strong performance. In [26], the authors propose a model-based offline RL algorithm called COMBO which does not rely on explicit uncertainty estimation. By regularizing the value function on out-of-distribution state-action pairs generated in the estimated world model, COMBO can benefit from the conservatism without requiring explicit uncertainty estimation techiques. As a remark, the algorithms proposed in [25, 24, 37, 26] all perform conservative policy optimization in a well-constructed dynamics model and the estimated dynamics model keeps fixed during the training of the RL agent. Different from those algorithms mentioned above, [45, 47] incorporate conservatism into the constructed dynamics model. By adversarially modifying the estimated dynamics model to minimize the value function under the current policy, the proposed methods can learn a robust policy with respect to the environment dynamics and can obtain probably approximately correct (PAC) performance guarantee.

## C  Auxiliary lemmas

Before we introduce the auxiliary lemmas, we re-write Assumptions 1 - 3 here for convenience.

**Assumption 1** For any reward parameter $\theta$ and any state-action pair $(s, a)$, the following conditions hold:

$$|r(s, a; \theta)| \leq C_r, \quad |U(s, a)| \leq C_u$$

where $C_r$ and $C_u$ are positive constants.

**Assumption 2** Given any policy $\pi$, the Markov chain under the estimated world model $\widehat{P}$ is irreducible and aperiodic. There exist constants $\kappa > 0$ and $\rho \in (0, 1)$ to ensure the following condition holds:

$$\max_{s \in \mathcal{S}} \|\widehat{P}(s_t \in \cdot | s_0 = s, \pi) - \mu_{\widehat{P}}^{\pi}(\cdot)\|_{\mathrm{TV}} \leq \kappa \rho^t, \quad \forall\, t \geq 0$$

where $\|\cdot\|_{\mathrm{TV}}$ denotes the total variation (TV) norm; $\mu_{\widehat{P}}^{\pi}$ is the stationary distribution of visited states under the policy $\pi$ and the world model $\widehat{P}$.

**Assumption 3** Under any reward parameter $\theta$, the following conditions hold for any $s \in \mathcal{S}$ and $a \in \mathcal{A}$:

$$\left\|\nabla_\theta r(s, a; \theta)\right\| \le L_r,$$
$$\left\|\nabla_\theta r(s, a; \theta_1) - \nabla_\theta r(s, a; \theta_2)\right\| \le L_g \|\theta_1 - \theta_2\|,$$

where $L_r$ and $L_g$ are positive constants.

Then we introduce the auxiliary lemmas as below.

**Lemma 5.** *(Proposition A.8. in [69]) Let $z$ be a discrete random variable that takes values in $\{1, ..., d\}$, distributed according to $q$. We write $q$ as a vector where $\vec{q} = [Pr(z = j)]_{j=1}^d$. Assume there are $N$ i.i.d. samples, and that the empirical estimate of $\vec{q}$ is $[\hat{q}]_j = \frac{1}{N}\sum_{i=1}^N \mathbf{1}[z_i = j]$, where $\hat{q}$ is a $d$-dimensional vector.*

*Then for any $\epsilon > 0$, the following result holds:*

$$Pr\left(\|\hat{q} - \vec{q}\|_2 \ge \frac{1}{\sqrt{N}} + \epsilon\right) \le e^{-N\epsilon^2}, \tag{23}$$

*which implies that:*

$$Pr\left(\|\hat{q} - \vec{q}\|_1 \ge \sqrt{d}\left(\frac{1}{\sqrt{N}} + \epsilon\right)\right) \le e^{-N\epsilon^2}. \tag{24}$$

**Lemma 6.** *(Lemma 3 in [70]) Under any initial distribution $\eta(\cdot)$ and any transition dynamics $P(\cdot|s, a)$, we denote $d_w(\cdot, \cdot)$ as the visitation measure of the visited state-action pair $(s, a)$ with a softmax policy parameterized by parameter $w$. Suppose Assumption 2 holds, then for all policy parameter $w$ and $w'$, we have*

$$\|d_w(\cdot, \cdot) - d_{w'}(\cdot, \cdot)\|_{\mathrm{TV}} \le C_d \|w - w'\| \tag{25}$$

*where $C_d$ is a positive constant.*

**Lemma 7.** *(Lemma 5 in [31]) Suppose Assumption 3 holds. Under the soft policy iteration defined in (13) - (14), we denote the soft Q-function under reward parameter $\theta_k$ and policy $\pi_{k+1}$ as $Q_{k+\frac{1}{2}}$. Moreover, recall that $Q_{k+1}$ has been defined as the soft Q-function under the reward parameter $\theta_{k+1}$ and policy $\pi_{k+1}$. Then for any $s \in \mathcal{S}$, $a \in \mathcal{A}$ and $k \ge 0$, the following inequality holds:*

$$|Q_{k+\frac{1}{2}}(s, a) - Q_{k+1}(s, a)| \le L_q \|\theta_k - \theta_{k+1}\|, \tag{26}$$

*where $L_q := \frac{L_r}{1-\gamma} > 0$ and $L_r$ is the positive constant defined in Assumption 3.*

**Lemma 8.** *(Lemma 6 in [31]) Following the soft policy iteration defined in (13) - (14), the following holds for any iteration $k \ge 0$:*

$$Q_k(s, a) \le Q_{k+\frac{1}{2}}(s, a) + \frac{2\gamma\epsilon_{\mathrm{app}}}{1-\gamma}, \quad \forall s \in \mathcal{S}, a \in \mathcal{A}, \tag{27a}$$

$$\|Q_{\theta_k} - Q_{k+\frac{1}{2}}\|_\infty \le \gamma\|Q_{\theta_k} - Q_k\|_\infty + \frac{2\gamma\epsilon_{\mathrm{app}}}{1-\gamma} \tag{27b}$$

*where $Q_{k+\frac{1}{2}}(\cdot, \cdot)$ denotes the soft Q-function under reward parameter $\theta_k$ and updated policy $\pi_{k+1}$, and $Q_{\theta_k}(\cdot, \cdot)$ denotes the soft Q-function under reward parameter $\theta_k$ and corresponding optimal policy $\pi_{\theta_k}$. Moreover, we denote $\|Q_{\theta_k} - Q_{k+\frac{1}{2}}\|_\infty = \max_{s\in\mathcal{S}}\max_{a\in\mathcal{A}}|Q_{\theta_k}(s, a) - Q_{k+\frac{1}{2}}(s, a)|$.*

**Remark.** In Lemma 7 and Lemma 8, the definitions of the soft Q-function $Q_k$ and the soft value function $V_k$ under a conservative MDP are given below:

$$Q_k(s, a) := r(s, a; \theta_k) + U(s, a) + \gamma\mathbb{E}_{s'\sim\widehat{P}(\cdot|s,a)}\big[V_k(s')\big] \tag{28a}$$

$$V_k(s) := \mathbb{E}_{\tau\sim(\eta,\pi_k,\widehat{P})}\Big[\sum_{t=0}^\infty \gamma^t\big(r(s_t, a_t; \theta_k) + U(s_t, a_t) + \mathcal{H}(\pi_k(\cdot|s_t))\big)\Big|s_0 = s\Big] \tag{28b}$$

where $U(s, a)$ is the penalty function for the state-action pair $(s, a)$, which is used to quantify the uncertainty in the estimated world model $\widehat{P}$. If we rewrite the above defined soft Q-function $Q_k$ as:

$$Q_k(s, a) = \tilde{r}(s, a; \theta_k) + \gamma\mathbb{E}_{s'\sim\widehat{P}(\cdot|s,a)}\big[V_k(s')\big]$$

where $\tilde{r}(s, a; \theta_k) := r(s, a; \theta_k) + U(s, a)$, then we can directly follow the proof steps in [31] to prove Lemma 7 and Lemma 8.

# D  Proof of Lemma 1

*Proof.* According to the closed-form expressions of the optimal policy $\pi_\theta$ and the optimal soft value function $V_\theta$ in (4), let us decompose the objective $L(\theta)$ defined in (2a) as below:

$$
\begin{aligned}
L(\theta) &= \mathbb{E}_{\tau^{\mathrm{E}} \sim (\eta, \pi^{\mathrm{E}}, P)} \left[ \sum_{t=0}^{\infty} \gamma^t \log \pi_\theta(a_t|s_t) \right] \\
&\overset{(i)}{=} \mathbb{E}_{\tau^{\mathrm{E}} \sim (\eta, \pi^{\mathrm{E}}, P)} \left[ \sum_{t=0}^{\infty} \gamma^t \log \left( \frac{\exp Q_\theta(s_t, a_t)}{\sum_{a \in \mathcal{A}} \exp Q_\theta(s_t, a)} \right) \right] \\
&\overset{(ii)}{=} \mathbb{E}_{\tau^{\mathrm{E}} \sim (\eta, \pi^{\mathrm{E}}, P)} \left[ \sum_{t=0}^{\infty} \gamma^t \Big( Q_\theta(s_t, a_t) - V_\theta(s_t) \Big) \right] \\
&= \sum_{t=0}^{\infty} \gamma^t \mathbb{E}_{(s_t, a_t) \sim (\eta, \pi^{\mathrm{E}}, P)} \Big[ Q_\theta(s_t, a_t) \Big] - \sum_{t=0}^{\infty} \gamma^t \mathbb{E}_{s_t \sim (\eta, \pi^{\mathrm{E}}, P)} \Big[ V_\theta(s_t) \Big] \\
&\overset{(iii)}{=} \sum_{t=0}^{\infty} \gamma^t \mathbb{E}_{(s_t, a_t) \sim (\eta, \pi^{\mathrm{E}}, P)} \Big[ r(s_t, a_t; \theta) + U(s_t, a_t) + \gamma \mathbb{E}_{s_{t+1} \sim \widehat{P}(\cdot|s_t, a_t)}[V_\theta(s_{t+1})] \Big] - \sum_{t=0}^{\infty} \gamma^t \mathbb{E}_{s_t \sim (\eta, \pi^{\mathrm{E}}, P)} \Big[ V_\theta(s_t) \Big] \\
&= \sum_{t=0}^{\infty} \gamma^t \mathbb{E}_{(s_t, a_t) \sim (\eta, \pi^{\mathrm{E}}, P)} \Big[ r(s_t, a_t; \theta) + U(s_t, a_t) \Big] - \mathbb{E}_{s_0 \sim \eta(\cdot)} \Big[ V_\theta(s_0) \Big] \\
&\quad + \left( \sum_{t=0}^{\infty} \gamma^{t+1} \mathbb{E}_{(s_t, a_t) \sim (\eta, \pi^{\mathrm{E}}, P), s_{t+1} \sim \widehat{P}(\cdot|s_t, a_t)} \big[ V_\theta(s_{t+1}) \big] - \sum_{t=0}^{\infty} \gamma^{t+1} \mathbb{E}_{s_{t+1} \sim (\eta, \pi^{\mathrm{E}}, P)} \big[ V_\theta(s_{t+1}) \big] \right) \\
&= \underbrace{\left( \sum_{t=0}^{\infty} \gamma^t \mathbb{E}_{(s_t, a_t) \sim (\eta, \pi^{\mathrm{E}}, P)} \Big[ r(s_t, a_t; \theta) + U(s_t, a_t) \Big] - \mathbb{E}_{s_0 \sim \eta(\cdot)} \Big[ V_\theta(s_0) \Big] \right)}_{\text{T1: surrogate objective}} \\
&\quad + \underbrace{\left( \sum_{t=0}^{\infty} \gamma^{t+1} \mathbb{E}_{(s_t, a_t) \sim (\eta, \pi^{\mathrm{E}}, P), s_{t+1} \sim \widehat{P}(\cdot|s_t, a_t)} \big[ V_\theta(s_{t+1}) \big] - \sum_{t=0}^{\infty} \gamma^{t+1} \mathbb{E}_{(s_t, a_t) \sim (\eta, \pi^{\mathrm{E}}, P), s_{t+1} \sim P(\cdot|s_t, a_t)} \big[ V_\theta(s_{t+1}) \big] \right)}_{\text{T2: error term due to transition probability mismatch}}
\end{aligned}
\tag{29}
$$

where (i) and (ii) follows the closed-form expression of the optimal policy $\pi_\theta$ and the optimal soft value function in (4). Moreover, (iii) follows the definition of the soft Q-function in (3b).

From the $T_2$ term in (29), we observe that the error term comes from the transition dynamics mismatch between the estimated world model $\widehat{P}$ and the ground-truth dynamics model $P$. Then we analyze the transition dynamics mismatch:

$$
\begin{aligned}
T_2 &= \sum_{t=0}^{\infty} \gamma^{t+1} \mathbb{E}_{(s_t, a_t) \sim (\eta, \pi^{\mathrm{E}}, P), s_{t+1} \sim \widehat{P}(\cdot|s_t, a_t)} \big[ V_\theta(s_{t+1}) \big] - \sum_{t=0}^{\infty} \gamma^{t+1} \mathbb{E}_{(s_t, a_t) \sim (\eta, \pi^{\mathrm{E}}, P), s_{t+1} \sim P(\cdot|s_t, a_t)} \big[ V_\theta(s_{t+1}) \big] \\
&= \sum_{t=0}^{\infty} \gamma^{t+1} \mathbb{E}_{(s_t, a_t) \sim (\eta, \pi^{\mathrm{E}}, P)} \left[ \mathbb{E}_{s_{t+1} \sim \widehat{P}(\cdot|s_t, a_t)} \big[ V_\theta(s_{t+1}) \big] - \mathbb{E}_{s_{t+1} \sim P(\cdot|s_t, a_t)} \big[ V_\theta(s_{t+1}) \big] \right] \\
&= \sum_{t=0}^{\infty} \gamma^{t+1} \mathbb{E}_{(s_t, a_t) \sim (\eta, \pi^{\mathrm{E}}, P)} \left[ \sum_{s_{t+1} \in \mathcal{S}} V_\theta(s_{t+1}) \big( \widehat{P}(s_{t+1}|s_t, a_t) - P(s_{t+1}|s_t, a_t) \big) \right]
\end{aligned}
$$

Recall that $d^{\mathrm{E}}(\cdot, \cdot)$ is defined in (5) where $d^{\mathrm{E}}(s, a) := (1-\gamma)\pi^{\mathrm{E}}(a|s) \sum_{t=0}^{\infty} \gamma^t P^{\pi^{\mathrm{E}}}(s_t = s|s_0 \sim \eta)$, we obtain the following result:

$$
\begin{aligned}
T_2 &= \sum_{t=0}^{\infty} \gamma^{t+1} \mathbb{E}_{(s_t, a_t) \sim (\eta, \pi^{\mathrm{E}}, P)} \left[ \sum_{s_{t+1} \in \mathcal{S}} V_\theta(s_{t+1}) \big( \widehat{P}(s_{t+1}|s_t, a_t) - P(s_{t+1}|s_t, a_t) \big) \right] \\
&= \gamma \sum_{t=0}^{\infty} \sum_{s \in \mathcal{S}, a \in \mathcal{A}} \gamma^t P(s_t = s|s_0 \sim \eta) \pi^{\mathrm{E}}(a_t = a|s_t = s) \left( \sum_{s_{t+1} \in \mathcal{S}} V_\theta(s_{t+1}) \big( \widehat{P}(s_{t+1}|s_t = s, a_t = a) - P(s_{t+1}|s_t = s, a_t = a) \big) \right) \\
&= \frac{\gamma}{1-\gamma} \cdot \mathbb{E}_{(s, a) \sim d^{\mathrm{E}}(\cdot, \cdot)} \left[ \sum_{s' \in \mathcal{S}} V_\theta(s') \big( \widehat{P}(s'|s, a) - P(s'|s, a) \big) \right]
\end{aligned}
\tag{30}
$$

Then we further denote the surrogate objective $\widehat{L}(\cdot)$ as below:

$$\widehat{L}(\theta) := \sum_{t=0}^{\infty} \gamma^t \mathbb{E}_{(s_t, a_t) \sim (\eta, \pi^{\mathrm{E}}, P)} \left[ r(s_t, a_t; \theta) + U(s_t, a_t) \right] - \mathbb{E}_{s_0 \sim \eta(\cdot)} \left[ V_\theta(s_0) \right]. \qquad (31)$$

By plugging (30) and (31) into (29), we obtain the decomposition of the likelihood objective $L(\theta)$ as below:

$$L(\theta) = \widehat{L}(\theta) + \frac{\gamma}{1-\gamma} \cdot \mathbb{E}_{(s,a) \sim d^{\mathrm{E}}(\cdot, \cdot)} \left[ \sum_{s' \in \mathcal{S}} V_\theta(s') \left( \widehat{P}(s'|s,a) - P(s'|s,a) \right) \right]. \qquad (32)$$

The lemma is proved. $\qquad\square$

## E  Proof of Lemma 2

*Proof.* According to (32), we have the following series of relations:

$$\begin{aligned}
|L(\theta) - \widehat{L}(\theta)| &= \frac{\gamma}{1-\gamma} \cdot \left| \mathbb{E}_{(s,a) \sim d^{\mathrm{E}}(\cdot, \cdot)} \left[ \sum_{s' \in \mathcal{S}} V_\theta(s') \left( \widehat{P}(s'|s,a) - P(s'|s,a) \right) \right] \right| \\
&\leq \frac{\gamma}{1-\gamma} \cdot \mathbb{E}_{(s,a) \sim d^{\mathrm{E}}(\cdot, \cdot)} \left[ \sum_{s' \in \mathcal{S}} |V_\theta(s')| \cdot |\widehat{P}(s'|s,a) - P(s'|s,a)| \right] \\
&\leq \frac{\gamma}{1-\gamma} \cdot \max_{\tilde{s} \in \mathcal{S}} |V_\theta(\tilde{s})| \cdot \mathbb{E}_{(s,a) \sim d^{\mathrm{E}}(\cdot, \cdot)} \left[ \sum_{s' \in \mathcal{S}} |\widehat{P}(s'|s,a) - P(s'|s,a)| \right] \\
&= \frac{\gamma}{1-\gamma} \cdot R(\theta) \cdot \mathbb{E}_{(s,a) \sim d^{\mathrm{E}}(\cdot, \cdot)} \left[ \left\| \widehat{P}(\cdot|s,a) - P(\cdot|s,a) \right\|_1 \right] \qquad (33)
\end{aligned}$$

where (33) follows the definition $R(\theta) := \max_{s \in \mathcal{S}} |V_\theta(s)|$ and the definition of the visitation measure $d^{\mathrm{E}}(\cdot, \cdot)$ in (5). According to the definition of $V_\theta(\cdot)$ in (3a), we have

$$\begin{aligned}
R(\theta) &= \max_{s \in \mathcal{S}} |V_\theta(s)| \\
&\overset{(i)}{=} \max_{s \in \mathcal{S}} \left| \mathbb{E}_{\tau \sim (\eta, \pi_\theta, \widehat{P})} \left[ \sum_{t=0}^{\infty} \gamma^t \left( r(s_t, a_t; \theta) + U(s_t, a_t) + \mathcal{H}(\pi_\theta(\cdot|s_t)) \right) \Big| s_0 = s \right] \right| \\
&\leq \max_{s \in \mathcal{S}} \mathbb{E}_{\tau \sim (\eta, \pi_\theta, \widehat{P})} \left[ \sum_{t=0}^{\infty} \gamma^t \left( |r(s_t, a_t; \theta)| + |U(s_t, a_t)| + |\mathcal{H}(\pi_\theta(\cdot|s_t))| \right) \Big| s_0 = s \right] \\
&\overset{(ii)}{\leq} \max_{s \in \mathcal{S}} \mathbb{E}_{\tau \sim (\eta, \pi_\theta, \widehat{P})} \left[ \sum_{t=0}^{\infty} \gamma^t \left( C_r + C_u + |\mathcal{H}(\pi_\theta(\cdot|s_t))| \right) \Big| s_0 = s \right] \\
&\overset{(iii)}{\leq} \max_{s \in \mathcal{S}} \mathbb{E}_{\tau \sim (\eta, \pi_\theta, \widehat{P})} \left[ \sum_{t=0}^{\infty} \gamma^t \left( C_r + C_u + \log |\mathcal{A}| \right) \Big| s_0 = s \right] \\
&= \frac{C_r + C_u + \log |\mathcal{A}|}{1 - \gamma}.
\end{aligned}$$

where (i) follows the definition of $V_\theta(\cdot)$ in (3a) and (ii) follows (8). Moreover, (iii) follows the fact that information entropy is non-negative and the maximum entropy is obtained under uniform distribution where we have $|\mathcal{H}(\pi_\theta(\cdot|s))| = \mathcal{H}(\pi_\theta(\cdot|s)) \leq -\sum_{a \in \mathcal{A}} \frac{1}{|\mathcal{A}|} \log |\frac{1}{|\mathcal{A}|}| = \log |\mathcal{A}|$.

Denote $C_v := \frac{C_r + C_u + \log |\mathcal{A}|}{1 - \gamma}$, we obtain the property: $R(\theta) = \max_{s \in \mathcal{S}} |V_\theta(s)| \leq C_v$. Plugging this result into (33), we obtain the following result to finish the proof:

$$|L(\theta) - \widehat{L}(\theta)| \leq \frac{\gamma C_v}{1 - \gamma} \cdot \mathbb{E}_{(s,a) \sim d^{\mathrm{E}}(\cdot, \cdot)} \left[ \left\| \widehat{P}(\cdot|s,a) - P(\cdot|s,a) \right\|_1 \right].$$

The lemma is proved. $\qquad\square$

## F Proof of Lemma 3

*Proof.* To start the analysis, we first take gradient of the surrogate objective $\widehat{L}(\theta)$ (as defined in (7)) w.r.t. $\theta$:

$$\nabla\widehat{L}(\theta) \stackrel{(i)}{=} \mathbb{E}_{\tau^{\mathrm{E}}\sim(\eta,\pi^{\mathrm{E}},P)}\left[\sum_{t=0}^{\infty}\gamma^t\left(\nabla_\theta r(s_t,a_t;\theta) + \nabla_\theta U(s_t,a_t)\right)\right] - \mathbb{E}_{s_0\sim\eta(\cdot)}\left[\nabla_\theta V_\theta(s_0)\right]$$

$$\stackrel{(ii)}{=} \mathbb{E}_{\tau^{\mathrm{E}}\sim(\eta,\pi^{\mathrm{E}},P)}\left[\sum_{t=0}^{\infty}\gamma^t\nabla_\theta r(s_t,a_t;\theta)\right] - \mathbb{E}_{s_0\sim\eta(\cdot)}\left[\nabla_\theta \log\left(\sum_{a\in\mathcal{A}}\exp Q_\theta(s_0,a)\right)\right]$$

$$= \mathbb{E}_{\tau^{\mathrm{E}}\sim(\eta,\pi^{\mathrm{E}},P)}\left[\sum_{t=0}^{\infty}\gamma^t\nabla_\theta r(s_t,a_t;\theta)\right] - \mathbb{E}_{s_0\sim\eta(\cdot)}\left[\sum_{a\in\mathcal{A}}\left(\frac{\exp Q_\theta(s_0,a)}{\sum_{\tilde{a}\in\mathcal{A}}\exp Q_\theta(s_0,\tilde{a})}\nabla_\theta Q_\theta(s_0,a)\right)\right]$$

$$\stackrel{(iii)}{=} \mathbb{E}_{\tau^{\mathrm{E}}\sim(\eta,\pi^{\mathrm{E}},P)}\left[\sum_{t=0}^{\infty}\gamma^t\nabla_\theta r(s_t,a_t;\theta)\right] - \mathbb{E}_{s_0\sim\eta(\cdot)}\left[\sum_{a\in\mathcal{A}}\pi_\theta(a|s_0)\nabla_\theta Q_\theta(s_0,a)\right]$$

$$= \mathbb{E}_{\tau^{\mathrm{E}}\sim(\eta,\pi^{\mathrm{E}},P)}\left[\sum_{t=0}^{\infty}\gamma^t\nabla_\theta r(s_t,a_t;\theta)\right] - \mathbb{E}_{s_0\sim\eta(\cdot),a_0\sim\pi_\theta(\cdot|s_0)}\left[\nabla_\theta Q_\theta(s_0,a_0)\right] \qquad (34)$$

where (i) follows the definition of the surrogate objective $\widehat{L}(\theta)$ in (7), and (ii) follows the closed-form expression of the optimal soft value function $V_\theta$ in (4) and the penalty function $U(s,a)$ is independent of the reward parameter $\theta$. Moreover, (iii) follows the closed-form expression of the optimal policy $\pi_\theta$ in (4). To further analyze the gradient expression in (34), we must derive the gradient of the optimal soft Q-function $Q_\theta$. Recall that $Q_\theta$ is the soft Q-function under the optimal policy $\pi_\theta$, the penalty function $U$ and the estimated world model $\widehat{P}$. Therefore, we have the following derivations:

$$\nabla_\theta Q_\theta(s_0,a_0)$$

$$\stackrel{(i)}{=} \nabla_\theta\left(r(s_0,a_0;\theta) + U(s_0,a_0) + \gamma\mathbb{E}_{s_1\sim\widehat{P}(\cdot|s_0,a_0)}\left[V_\theta(s_1)\right]\right)$$

$$\stackrel{(ii)}{=} \nabla_\theta r(s_0,a_0;\theta) + \nabla_\theta U(s_0,a_0) + \gamma\mathbb{E}_{s_1\sim\widehat{P}(\cdot|s_0,a_0)}\left[\nabla_\theta\log\left(\sum_{\tilde{a}\in\mathcal{A}}\exp Q_\theta(s_0,\tilde{a})\right)\right]$$

$$= \nabla_\theta r(s_0,a_0;\theta) + \gamma\mathbb{E}_{s_1\sim\widehat{P}(\cdot|s_0,a_0)}\left[\sum_{a\in\mathcal{A}}\frac{\exp Q_\theta(s_1,a)}{\sum_{\tilde{a}\in\mathcal{A}}\exp Q_\theta(s_1,\tilde{a})}\nabla_\theta Q_\theta(s_1,a)\right]$$

$$\stackrel{(iii)}{=} \nabla_\theta r(s_0,a_0;\theta) + \gamma\mathbb{E}_{s_1\sim\widehat{P}(\cdot|s_0,a_0)}\left[\sum_{a\in\mathcal{A}}\pi_\theta(a|s_1)\nabla_\theta Q_\theta(s_1,a)\right]$$

$$\stackrel{(iv)}{=} \nabla_\theta r(s_0,a_0;\theta) + \gamma\mathbb{E}_{s_1\sim\widehat{P}(\cdot|s_0,a_0),a_1\sim\pi_\theta(\cdot|s_1)}\left[\nabla_\theta\left(r(s_1,a_1;\theta) + U(s_1,a_1) + \gamma\mathbb{E}_{s_2\sim\widehat{P}(\cdot|s_1,a_1)}\left[V_\theta(s_2)\right]\right)\right]$$

where (i) and (iv) follows the definition of the optimal soft Q-function in (3b); (ii) follows the closed-form expression of $V_\theta$ in (4); (iii) is from (4). By recursively applying the equalities (i) and (iv), we obtain the following result:

$$\nabla_\theta Q_\theta(s_0,a_0)$$

$$= \nabla_\theta\left(r(s_0,a_0;\theta) + U(s_0,a_0) + \gamma\mathbb{E}_{s_1\sim\widehat{P}(\cdot|s_0,a_0)}\left[V_\theta(s_1)\right]\right)$$

$$= \nabla_\theta r(s_0,a_0;\theta) + \gamma\mathbb{E}_{s_1\sim\widehat{P}(\cdot|s_0,a_0),a_1\sim\pi_\theta(\cdot|s_1)}\left[\nabla_\theta\left(r(s_1,a_1;\theta) + U(s_1,a_1) + \gamma\mathbb{E}_{s_2\sim\widehat{P}(\cdot|s_1,a_1)}\left[V_\theta(s_2)\right]\right)\right]$$

$$= \mathbb{E}_{\tau^{\mathrm{A}}\sim(\pi_\theta,\widehat{P})}\left[\sum_{t=0}^{\infty}\gamma^t\nabla_\theta r(s_t,a_t;\theta) \mid s_0,a_0\right]. \qquad (35)$$

Finally, by plugging (35) into (34), we obtain the gradient expression of the surrogate objective $\widehat{L}(\theta)$ as below:

$$
\begin{aligned}
\nabla \widehat{L}(\theta) &= \mathbb{E}_{\tau^{\mathrm{E}} \sim (\eta, \pi^{\mathrm{E}}, P)} \left[ \sum_{t=0}^{\infty} \gamma^t \nabla_\theta r(s_t, a_t; \theta) \right] - \mathbb{E}_{s_0 \sim \eta(\cdot), a_0 \sim \pi_\theta(\cdot | s_0)} \left[ \nabla_\theta Q_\theta(s_0, a_0) \right] \\
&= \mathbb{E}_{\tau^{\mathrm{E}} \sim (\eta, \pi^{\mathrm{E}}, P)} \left[ \sum_{t=0}^{\infty} \gamma^t \nabla_\theta r(s_t, a_t; \theta) \right] - \mathbb{E}_{s_0 \sim \eta(\cdot), a_0 \sim \pi_\theta(\cdot | s_0)} \left[ \mathbb{E}_{\tau^{\mathrm{A}} \sim (\pi_\theta, \widehat{P})} \left[ \sum_{t=0}^{\infty} \gamma^t \nabla_\theta r(s_t, a_t; \theta) \mid s_0, a_0 \right] \right] \\
&= \mathbb{E}_{\tau^{\mathrm{E}} \sim (\eta, \pi^{\mathrm{E}}, P)} \left[ \sum_{t=0}^{\infty} \gamma^t \nabla_\theta r(s_t, a_t; \theta) \right] - \mathbb{E}_{\tau^{\mathrm{A}} \sim (\eta, \pi_\theta, \widehat{P})} \left[ \sum_{t=0}^{\infty} \gamma^t \nabla_\theta r(s_t, a_t; \theta) \right]. \quad (36)
\end{aligned}
$$

The lemma is proved. $\qquad\square$

## G  Proof of Lemma 4

Suppose Assumptions 2 - 3 hold, we prove the Lipschitz continuous property of the optimal soft Q-function $Q_\theta$ in (19a) and the Lipschitz smooth property of the surrogate objective $\widehat{L}(\theta)$ in (19b) respectively.

### G.1  Proof of Inequality (19a)

*Proof.* In order to show that the optimal soft Q-function $Q_\theta(s, a)$ is Lipschitz continuous w.r.t. the reward parameter $\theta$. for any state-action pair $(s, a)$, we take two steps to finish the proof. First, we show that $Q_\theta$ has bounded gradient under any reward parameter $\theta$. Then we could use the mean value theorem to complete the proof.

According to the gradient expression of the optimal soft Q-function $Q_\theta$ in (35), the following result holds:

$$
\begin{aligned}
\|\nabla_\theta Q_\theta(s, a)\| &= \left\| \mathbb{E}_{\tau^{\mathrm{A}} \sim (\pi_\theta, \widehat{P})} \left[ \sum_{t=0}^{\infty} \gamma^t \nabla_\theta r(s_t, a_t; \theta) \mid s_0 = s, a_0 = a \right] \right\| \\
&\overset{(i)}{\leq} \mathbb{E}_{\tau^{\mathrm{A}} \sim (\pi_\theta, \widehat{P})} \left[ \sum_{t=0}^{\infty} \gamma^t \|\nabla_\theta r(s_t, a_t; \theta)\| \mid s_0 = s, a_0 = a \right] \\
&\overset{(ii)}{\leq} \mathbb{E}_{\tau^{\mathrm{A}} \sim (\pi_\theta, \widehat{P})} \left[ \sum_{t=0}^{\infty} \gamma^t L_r \mid s_0 = s, a_0 = a \right] \\
&= \frac{L_r}{1 - \gamma} \quad (37)
\end{aligned}
$$

where (i) follows the Jensen's inequality and (ii) follows from the assumption that the reward gradient is bounded; see (18) in Assumption 3. Denote $L_q := \frac{L_r}{1-\gamma}$, we are able to show the Lipschitz continuous property of the optimal soft Q-function $Q_\theta$ as below:

$$
|Q_{\theta_1}(s, a) - Q_{\theta_2}(s, a)| \overset{(i)}{=} \left| \langle \theta_1 - \theta_2, \nabla_\theta Q_{\tilde{\theta}}(s, a) \rangle \right| \leq \|\theta_1 - \theta_2\| \cdot \|\nabla_\theta Q_{\tilde{\theta}}(s, a)\| \overset{(ii)}{\leq} L_q \|\theta_1 - \theta_2\|
$$

where $\tilde{\theta}$ is a convex combination between $\theta_1$ and $\theta_2$. Moreover, (i) is from the mean value theorem and (ii) follows (37). $\qquad\square$

## G.2 Proof of Inequality (19b)

*Proof.* In Lemma 3, we have shown the expression of the gradient of the surrogate objective $\widehat{L}(\theta)$. Then for any reward parameters $\theta_1$ and $\theta_2$, we are able to obtain the following result:

$$
\|\nabla\widehat{L}(\theta_1) - \nabla\widehat{L}(\theta_2)\|
$$
$$
\overset{(i)}{:=} \left\|\left(\mathbb{E}_{\tau^{\mathrm{E}}\sim(\eta,\pi^{\mathrm{E}},P)}\left[\sum_{t=0}^{\infty}\gamma^t\nabla_\theta r(s_t,a_t;\theta_1)\right] - \mathbb{E}_{\tau^{\mathrm{A}}\sim(\eta,\pi_{\theta_1},\widehat{P})}\left[\sum_{t=0}^{\infty}\gamma^t\nabla_\theta r(s_t,a_t;\theta_1)\right]\right)\right.
$$
$$
\left. - \left(\mathbb{E}_{\tau^{\mathrm{E}}\sim(\eta,\pi^{\mathrm{E}},P)}\left[\sum_{t=0}^{\infty}\gamma^t\nabla_\theta r(s_t,a_t;\theta_2)\right] - \mathbb{E}_{\tau^{\mathrm{A}}\sim(\eta,\pi_{\theta_2},\widehat{P})}\left[\sum_{t=0}^{\infty}\gamma^t\nabla_\theta r(s_t,a_t;\theta_2)\right]\right)\right\|
$$
$$
\leq \left\|\mathbb{E}_{\tau^{\mathrm{E}}\sim(\eta,\pi^{\mathrm{E}},P)}\left[\sum_{t=0}^{\infty}\gamma^t\nabla_\theta r(s_t,a_t;\theta_1)\right] - \mathbb{E}_{\tau^{\mathrm{E}}\sim(\eta,\pi^{\mathrm{E}},P)}\left[\sum_{t=0}^{\infty}\gamma^t\nabla_\theta r(s_t,a_t;\theta_2)\right]\right\|
$$
$$
+ \left\|\mathbb{E}_{\tau^{\mathrm{A}}\sim(\eta,\pi_{\theta_1},\widehat{P})}\left[\sum_{t=0}^{\infty}\gamma^t\nabla_\theta r(s_t,a_t;\theta_1)\right] - \mathbb{E}_{\tau^{\mathrm{A}}\sim(\eta,\pi_{\theta_2},\widehat{P})}\left[\sum_{t=0}^{\infty}\gamma^t\nabla_\theta r(s_t,a_t;\theta_2)\right]\right\| \quad (38)
$$

where (i) follows the gradient expression in (15). For the first term in (38), the following series of relations holds:

$$
\left\|\mathbb{E}_{\tau^{\mathrm{E}}\sim(\eta,\pi^{\mathrm{E}},P)}\left[\sum_{t=0}^{\infty}\gamma^t\nabla_\theta r(s_t,a_t;\theta_1)\right] - \mathbb{E}_{\tau^{\mathrm{E}}\sim(\eta,\pi^{\mathrm{E}},P)}\left[\sum_{t=0}^{\infty}\gamma^t\nabla_\theta r(s_t,a_t;\theta_2)\right]\right\|
$$
$$
= \left\|\mathbb{E}_{\tau^{\mathrm{E}}\sim(\eta,\pi^{\mathrm{E}},P)}\left[\sum_{t=0}^{\infty}\gamma^t\big(\nabla_\theta r(s_t,a_t;\theta_1) - \nabla_\theta r(s_t,a_t;\theta_2)\big)\right]\right\|
$$
$$
\overset{(i)}{\leq} \mathbb{E}_{\tau^{\mathrm{E}}\sim(\eta,\pi^{\mathrm{E}},P)}\left[\sum_{t=0}^{\infty}\gamma^t\big\|\nabla_\theta r(s_t,a_t;\theta_1) - \nabla_\theta r(s_t,a_t;\theta_2)\big\|\right]
$$
$$
\overset{(ii)}{\leq} \mathbb{E}_{\tau^{\mathrm{E}}\sim(\eta,\pi^{\mathrm{E}},P)}\left[\sum_{t=0}^{\infty}\gamma^t L_g\big\|\theta_1 - \theta_2\big\|\right]
$$
$$
= \frac{L_g}{1-\gamma}\big\|\theta_1 - \theta_2\big\| \quad (39)
$$

where (i) follows Jensen's inequality and (ii) follows (18) from the Assumption 3. For the second term in (38), we have

$$
\left\|\mathbb{E}_{\tau^{\mathrm{A}}\sim(\eta,\pi_{\theta_1},\widehat{P})}\left[\sum_{t=0}^{\infty}\gamma^t\nabla_\theta r(s_t,a_t;\theta_1)\right] - \mathbb{E}_{\tau^{\mathrm{A}}\sim(\eta,\pi_{\theta_2},\widehat{P})}\left[\sum_{t=0}^{\infty}\gamma^t\nabla_\theta r(s_t,a_t;\theta_2)\right]\right\|
$$
$$
\leq \underbrace{\left\|\mathbb{E}_{\tau^{\mathrm{A}}\sim(\eta,\pi_{\theta_1},\widehat{P})}\left[\sum_{t=0}^{\infty}\gamma^t\nabla_\theta r(s_t,a_t;\theta_1)\right] - \mathbb{E}_{\tau^{\mathrm{A}}\sim(\eta,\pi_{\theta_2},\widehat{P})}\left[\sum_{t=0}^{\infty}\gamma^t\nabla_\theta r(s_t,a_t;\theta_1)\right]\right\|}_{\text{T1: error term due to policy mismatch}}
$$
$$
+ \underbrace{\left\|\mathbb{E}_{\tau^{\mathrm{A}}\sim(\eta,\pi_{\theta_2},\widehat{P})}\left[\sum_{t=0}^{\infty}\gamma^t\nabla_\theta r(s_t,a_t;\theta_1)\right] - \mathbb{E}_{\tau^{\mathrm{A}}\sim(\eta,\pi_{\theta_2},\widehat{P})}\left[\sum_{t=0}^{\infty}\gamma^t\nabla_\theta r(s_t,a_t;\theta_2)\right]\right\|}_{\text{T2: error term due to reward parameter mismatch}}. \quad (40)
$$

In (40), we decompose the difference between reward gradient trajectories into two error terms. The first error term is due to the policy mismatch between $\pi_{\theta_1}$ and $\pi_{\theta_2}$. The second error term is due to

the reward parameter mismatch between $\theta_1$ and $\theta_2$. Here, we first bound the error term T1 in (40):

$$\left\| \mathbb{E}_{\tau^A \sim (\eta, \pi_{\theta_1}, \widehat{P})} \left[ \sum_{t=0}^{\infty} \gamma^t \nabla_\theta r(s_t, a_t; \theta_1) \right] - \mathbb{E}_{\tau^A \sim (\eta, \pi_{\theta_2}, \widehat{P})} \left[ \sum_{t=0}^{\infty} \gamma^t \nabla_\theta r(s_t, a_t; \theta_1) \right] \right\|$$

$$\overset{(i)}{=} \left\| \frac{1}{1-\gamma} \mathbb{E}_{(s,a) \sim d_{\widehat{P}}^{\pi_{\theta_1}}(\cdot, \cdot)} \left[ \nabla_\theta r(s, a; \theta_1) \right] - \frac{1}{1-\gamma} \mathbb{E}_{(s,a) \sim d_{\widehat{P}}^{\pi_{\theta_2}}(\cdot, \cdot)} \left[ \nabla_\theta r(s, a; \theta_1) \right] \right\|$$

$$= \frac{1}{1-\gamma} \left\| \sum_{s \in \mathcal{S}, a \in \mathcal{A}} \nabla_\theta r(s, a; \theta_1) \cdot \left( d_{\widehat{P}}^{\pi_{\theta_1}}(s, a) - d_{\widehat{P}}^{\pi_{\theta_2}}(s, a) \right) \right\|$$

$$\leq \frac{1}{1-\gamma} \sum_{s \in \mathcal{S}, a \in \mathcal{A}} \left\| \nabla_\theta r(s, a; \theta_1) \right\| \cdot \left| d_{\widehat{P}}^{\pi_{\theta_1}}(s, a) - d_{\widehat{P}}^{\pi_{\theta_2}}(s, a) \right|$$

$$\overset{(ii)}{\leq} \frac{2L_r}{1-\gamma} \left\| d_{\widehat{P}}^{\pi_{\theta_1}}(\cdot, \cdot) - d_{\widehat{P}}^{\pi_{\theta_2}}(\cdot, \cdot) \right\|_{\mathrm{TV}}$$

$$\overset{(iii)}{\leq} \frac{2L_r C_d}{1-\gamma} \left\| Q_{\theta_1} - Q_{\theta_2} \right\|$$

$$\overset{(iv)}{\leq} \frac{2L_r C_d \sqrt{|\mathcal{S}| \cdot |\mathcal{A}|}}{1-\gamma} \left\| Q_{\theta_1} - Q_{\theta_2} \right\|_\infty$$

$$\overset{(v)}{\leq} \frac{2L_q L_r C_d \sqrt{|\mathcal{S}| \cdot |\mathcal{A}|}}{1-\gamma} \left\| \theta_1 - \theta_2 \right\| \tag{41}$$

where (i) is from the definition of the visitation measure and (ii) follows the bounded gradient of reward function in (18) in Assumption 3. Moreover, (iii) is from (25) in Lemma 6 and (v) follows the Lipschitz property (19a) in Lemma 4. For the inequality (iv), it holds due to the fact $|Q_{\theta_1}(s, a) - Q_{\theta_2}(s, a)| \leq \|Q_{\theta_1} - Q_{\theta_2}\|_\infty$ for any state-action pair $(s, a)$ and thus $\|Q_{\theta_1} - Q_{\theta_2}\| \leq \sqrt{|\mathcal{S}| \cdot |\mathcal{A}|} \|Q_{\theta_1} - Q_{\theta_2}\|_\infty$.

Next, let us bound the second error term in (40):

$$\left\| \mathbb{E}_{\tau^A \sim (\eta, \pi_{\theta_2}, \widehat{P})} \left[ \sum_{t=0}^{\infty} \gamma^t \nabla_\theta r(s_t, a_t; \theta_1) \right] - \mathbb{E}_{\tau^A \sim (\eta, \pi_{\theta_2}, \widehat{P})} \left[ \sum_{t=0}^{\infty} \gamma^t \nabla_\theta r(s_t, a_t; \theta_2) \right] \right\|$$

$$= \left\| \mathbb{E}_{\tau^A \sim (\eta, \pi_{\theta_2}, \widehat{P})} \left[ \sum_{t=0}^{\infty} \gamma^t \left( \nabla_\theta r(s_t, a_t; \theta_1) - \nabla_\theta r(s_t, a_t; \theta_2) \right) \right] \right\|$$

$$\overset{(i)}{\leq} \mathbb{E}_{\tau^A \sim (\eta, \pi_{\theta_2}, \widehat{P})} \left[ \sum_{t=0}^{\infty} \gamma^t \left\| \nabla_\theta r(s_t, a_t; \theta_1) - \nabla_\theta r(s_t, a_t; \theta_2) \right\| \right]$$

$$\overset{(ii)}{\leq} \mathbb{E}_{\tau^A \sim (\eta, \pi_{\theta_2}, \widehat{P})} \left[ \sum_{t=0}^{\infty} \gamma^t L_g \|\theta_1 - \theta_2\| \right]$$

$$= \frac{L_g}{1-\gamma} \|\theta_1 - \theta_2\| \tag{42}$$

where (i) follows Jensen's inequality and (ii) follows the Lipschitz property in (18) from Assumption 3. Plugging (41) and (42) into (40), we could show the following result:

$$\left\| \mathbb{E}_{\tau^A \sim (\eta, \pi_{\theta_1}, \widehat{P})} \left[ \sum_{t=0}^{\infty} \gamma^t \nabla_\theta r(s_t, a_t; \theta_1) \right] - \mathbb{E}_{\tau^A \sim (\eta, \pi_{\theta_2}, \widehat{P})} \left[ \sum_{t=0}^{\infty} \gamma^t \nabla_\theta r(s_t, a_t; \theta_2) \right] \right\| \leq \frac{2L_q L_r C_d \sqrt{|\mathcal{S}| \cdot |\mathcal{A}|} + L_g}{1-\gamma} \cdot \|\theta_1 - \theta_2\|. \tag{43}$$

We denote the constant $L_c = \frac{2L_q L_r C_d \sqrt{|\mathcal{S}| \cdot |\mathcal{A}|} + 2L_g}{1-\gamma}$. By plugging (39) and (43) into (38), we arrive at the desired Lipschitz property of the surrogate objective $\widehat{L}(\theta)$, as shown below:

$$\|\nabla \widehat{L}(\theta_1) - \nabla \widehat{L}(\theta_2)\| \leq L_c \|\theta_1 - \theta_2\|.$$

We completed the proof of the Inequality (19b). $\qquad\square$

# H Proof of Proposition 1

*Proof.* Before starting the proof, first recall that we have defined the expert-visited state-action space $\Omega := \{(s,a) \mid d^{\mathrm{E}}(s,a) > 0\}$ and the expert-visited state space $\mathcal{S}^{\mathrm{E}} := \{s \mid \sum_{a \in \mathcal{A}} d^{\mathrm{E}}(s,a) > 0\}$. Then we have the following analysis.

For all state-action pairs $(s,a) \in \Omega$, there are at most $|\mathcal{S}^{\mathrm{E}}|$ *active* states in the distribution $P(\cdot|s,a)$ such that $P(s'|s,a) > 0$. This is due to the fact that the next state $s'$ of an expert-visited state-action pair $(s,a)$ must belong to the expert-visited state space $\mathcal{S}^{\mathrm{E}}$. Then suppose there are $N$ i.i.d. samples on each state-action pair $(s,a) \in \Omega$, we can obtain an empirical estimate of the transition probability as $\widehat{P}(s'|s,a) = \frac{N(s,a,s')}{N}$ for any $s' \in \mathcal{S}^{\mathrm{E}}$, where $N(s,a,s')$ is the number of observed transition samples $(s,a,s')$. For any $(s,a) \in \Omega$ and $s' \notin \mathcal{S}^{\mathrm{E}}$, we know $\widehat{P}(s'|s,a) = \frac{N(s,a,s')}{N} = 0$, since the expert only visit states in $\mathcal{S}^{\mathrm{E}}$.

Therefore, at any state-action pair $(s,a) \in \Omega$ and by following (24) in Lemma 5, the following result holds with probability greater than $1 - e^{-N\epsilon^2}$:

$$\|P(\cdot|s,a) - \widehat{P}(\cdot|s,a)\|_1 \leq \sqrt{|\mathcal{S}^{\mathrm{E}}|} \cdot \left(\frac{1}{\sqrt{N}} + \epsilon\right). \tag{44}$$

Let us denote $\tilde{\delta} := e^{-N\epsilon^2}$, then we have $\epsilon = \sqrt{\frac{1}{N}\ln\frac{1}{\tilde{\delta}}}$. By plugging $\epsilon$ into (44), we can show that the following result holds with probability greater than $1 - \tilde{\delta}$:

$$\|P(\cdot|s,a) - \widehat{P}(\cdot|s,a)\|_1 \leq \sqrt{|\mathcal{S}^{\mathrm{E}}|} \cdot \left(\frac{1}{\sqrt{N}} + \sqrt{\frac{1}{N}\ln\frac{1}{\tilde{\delta}}}\right) = c\sqrt{\frac{|\mathcal{S}^{\mathrm{E}}|}{N}\ln\frac{1}{\tilde{\delta}}}. \tag{45}$$

where we define $c := 1 + \frac{1}{\sqrt{\ln\frac{1}{\tilde{\delta}}}}$ as a positive constant dependent on the probability $\tilde{\delta}$. Then we can further show that

$$P\left(\max_{(s,a)\in\Omega} \|P(\cdot|s,a) - \widehat{P}(\cdot|s,a)\|_1 \geq c\sqrt{\frac{|\mathcal{S}^{\mathrm{E}}|}{N}\ln\frac{1}{\tilde{\delta}}}\right)$$

$$\overset{(i)}{\leq} \sum_{(s,a)\in\Omega} P\left(\|P(\cdot|s,a) - \widehat{P}(\cdot|s,a)\|_1 \geq c\sqrt{\frac{|\mathcal{S}^{\mathrm{E}}|}{N}\ln\frac{1}{\tilde{\delta}}}\right)$$

$$\overset{(ii)}{\leq} |\Omega| \cdot \tilde{\delta}$$

where (i) follows the union bound and (ii) follows (45). Therefore, with probability greater than $1 - |\Omega| \cdot \tilde{\delta}$, we have

$$\max_{(s,a)\in\Omega} \|P(\cdot|s,a) - \widehat{P}(\cdot|s,a)\|_1 \leq c\sqrt{\frac{|\mathcal{S}^{\mathrm{E}}|}{N}\ln\frac{1}{\tilde{\delta}}}. \tag{46}$$

Denoting $\delta := |\Omega| \cdot \tilde{\delta}$, then we have $\tilde{\delta} = \frac{\delta}{|\Omega|}$. Therefore, with probability greater than $1 - \delta$, the following result holds

$$\max_{(s,a)\in\Omega} \|P(\cdot|s,a) - \widehat{P}(\cdot|s,a)\|_1 \leq c\sqrt{\frac{|\mathcal{S}^{\mathrm{E}}|}{N}\ln\left(\frac{|\Omega|}{\delta}\right)}. \tag{47}$$

In order to control the estimation error $\max_{(s,a)\in\Omega} \|P(\cdot|s,a) - \widehat{P}(\cdot|s,a)\|_1 \leq \varepsilon$, the number of samples $N$ at each state-action pair $(s,a) \in \Omega$ should satisfy:

$$c\sqrt{\frac{|\mathcal{S}^{\mathrm{E}}|}{N}\ln\left(\frac{|\Omega|}{\delta}\right)} \leq \varepsilon \implies N \geq \frac{c^2 \cdot |\mathcal{S}^{\mathrm{E}}|}{\varepsilon^2}\ln\left(\frac{|\Omega|}{\delta}\right).$$

Suppose we uniformly sample each expert-visited state-action pair $(s,a) \in \Omega$ and the total number of samples in the transition dataset $\mathcal{D} = \{(s,a,s')\}$ satisfies:

$$\#\text{transition samples} \geq |\Omega| \cdot N \geq \frac{c^2 \cdot |\Omega| \cdot |\mathcal{S}^{\mathrm{E}}|}{\varepsilon^2}\ln\left(\frac{|\Omega|}{\delta}\right). \tag{48}$$

Then with probability greater than $1-\delta$, the generalization error between $P$ and $\widehat{P}$ could be controlled:

$$\mathbb{E}_{(s,a)\sim d^{\mathrm{E}}(\cdot,\cdot)}\big[\|P(\cdot|s,a)-\widehat{P}(\cdot|s,a)\|_1\big] \leq \max_{(s,a)\in\Omega}\|P(\cdot|s,a)-\widehat{P}(\cdot|s,a)\|_1 \leq \varepsilon. \qquad (49)$$

The theorem is proved. $\qquad\square$

# I  Proof of Theorem 1

*Proof.* First, we have the following decomposition of the error between the loss function evaluated at $\theta^*$ and $\hat{\theta}$, respectively:

$L(\theta^*) - L(\hat{\theta})$

$= \big(L(\theta^*) - \widehat{L}(\theta^*)\big) + \big(\widehat{L}(\theta^*) - \widehat{L}(\hat{\theta})\big) + \big(\widehat{L}(\hat{\theta}) - L(\hat{\theta})\big)$

$\overset{(i)}{\leq} \dfrac{\gamma C_v}{1-\gamma}\cdot\mathbb{E}_{(s,a)\sim d^{\mathrm{E}}(\cdot,\cdot)}\big[\|\widehat{P}(\cdot|s,a)-P(\cdot|s,a)\|_1\big] + \big(\widehat{L}(\theta^*)-\widehat{L}(\hat{\theta})\big) + \dfrac{\gamma C_v}{1-\gamma}\cdot\mathbb{E}_{(s,a)\sim d^{\mathrm{E}}(\cdot,\cdot)}\big[\|\widehat{P}(\cdot|s,a)-P(\cdot|s,a)\|_1\big]$

$= \dfrac{2\gamma C_v}{1-\gamma}\cdot\mathbb{E}_{(s,a)\sim d^{\mathrm{E}}(\cdot,\cdot)}\big[\|\widehat{P}(\cdot|s,a)-P(\cdot|s,a)\|_1\big] + \big(\widehat{L}(\theta^*)-\widehat{L}(\hat{\theta})\big) \qquad (50)$

where (i) follows (9). Since we have defined $\hat{\theta}$ as the optimal solution to $\widehat{L}(\cdot)$, we know that $\widehat{L}(\theta) - \widehat{L}(\hat{\theta}) \leq 0$ for any $\theta$. Plugging this result into (50), the following result holds:

$$L(\theta^*) - L(\hat{\theta}) \leq \dfrac{2\gamma C_v}{1-\gamma}\cdot\mathbb{E}_{(s,a)\sim d^{\mathrm{E}}(\cdot,\cdot)}\big[\|\widehat{P}(\cdot|s,a)-P(\cdot|s,a)\|_1\big] + \big(\widehat{L}(\theta^*)-\widehat{L}(\hat{\theta})\big)$$

$$\leq \dfrac{2\gamma C_v}{1-\gamma}\cdot\mathbb{E}_{(s,a)\sim d^{\mathrm{E}}(\cdot,\cdot)}\big[\|\widehat{P}(\cdot|s,a)-P(\cdot|s,a)\|_1\big].$$

Based on (48) and (49), we assume that each expert-visited state-action pair is uniformly sampled and the total number of transition samples satisfies:

$$\#\text{transition samples} \geq \dfrac{c^2\cdot|\Omega|\cdot|\mathcal{S}^{\mathrm{E}}|}{\big(\frac{(1-\gamma)\varepsilon}{2\gamma C_v}\big)^2}\ln\Big(\dfrac{|\Omega|}{\delta}\Big) = \dfrac{4\gamma^2\cdot C_v^2\cdot c^2\cdot|\Omega|\cdot|\mathcal{S}^{\mathrm{E}}|}{(1-\gamma)^2\varepsilon^2}\ln\Big(\dfrac{|\Omega|}{\delta}\Big).$$

Then with probability greater than $1-\delta$, the generalization error between transition dynamics and the optimality gap between reward estimates could be controlled:

$$\mathbb{E}_{(s,a)\sim d^{\mathrm{E}}(\cdot,\cdot)}\big[\|\widehat{P}(\cdot|s,a)-P(\cdot|s,a)\|_1\big] \leq \dfrac{(1-\gamma)\varepsilon}{2\gamma C_v},$$

$$L(\theta^*) - L(\hat{\theta}) \leq \dfrac{2\gamma C_v}{1-\gamma}\cdot\mathbb{E}_{(s,a)\sim d^{\mathrm{E}}(\cdot,\cdot)}\big[\|\widehat{P}(\cdot|s,a)-P(\cdot|s,a)\|_1\big] \leq \varepsilon.$$

The theorem is proved. $\qquad\square$

# J  Proof of Theorem 2

*Proof.* In this section, we prove the convergence results (20a) - (20b) respectively.

## J.1  Proof of convergence of the policy estimates in (20a)

In this section, we show the convergence result of the policy estimates $\{\pi_{k+1}\}_{k\geq 0}$, which track the optimal solutions $\{\pi_{\theta_k}\}_{k\geq 0}$. Recall that each policy $\pi_{k+1}$ is generated from the soft policy iteration in (14), then we track the approximation error between $\pi_{k+1}$ and $\pi_{\theta_k}$ as below:

$$\Big|\log\pi_{k+1}(a|s) - \log\pi_{\theta_k}(a|s)\Big|$$

$$\overset{(i)}{=} \Big|\log\Big(\dfrac{\exp\widehat{Q}_k(s,a)}{\sum_{\tilde{a}\in\mathcal{A}}\exp\widehat{Q}_k(s,\tilde{a})}\Big) - \log\Big(\dfrac{\exp Q_{\theta_k}(s,a)}{\sum_{\tilde{a}\in\mathcal{A}}\exp Q_{\theta_k}(s,\tilde{a})}\Big)\Big|$$

$$= \Big|\Big(\widehat{Q}_k(s,a) - \log\big(\sum_{\tilde{a}\in\mathcal{A}}\exp\widehat{Q}_k(s,\tilde{a})\big)\Big) - \Big(Q_{\theta_k}(s,a) - \log\big(\sum_{\tilde{a}\in\mathcal{A}}\exp Q_{\theta_k}(s,\tilde{a})\big)\Big)\Big|$$

$$\leq \Big|\widehat{Q}_k(s,a) - Q_{\theta_k}(s,a)\Big| + \Big|\log\big(\sum_{\tilde{a}\in\mathcal{A}}\exp\widehat{Q}_k(s,\tilde{a})\big) - \log\big(\sum_{\tilde{a}\in\mathcal{A}}\exp Q_{\theta_k}(s,\tilde{a})\big)\Big| \qquad (51)$$

where (i) follows (14) and (4). In order to analyze the second error term in (51), we first denote two $|\mathcal{A}|$-dimensional vectors $\overrightarrow{a} := [a_1, a_2, \cdots, a_{|\mathcal{A}|}]$ and $\overrightarrow{b} := [b_1, b_2, \cdots, b_{|\mathcal{A}|}]$. Then we could obtain the following result:

$$
\begin{aligned}
\left| \log\left( \|\exp(\overrightarrow{a})\|_1 \right) - \log\left( \|\exp(\overrightarrow{b})\|_1 \right) \right| &\overset{(i)}{=} \left| \langle \overrightarrow{a} - \overrightarrow{b}, \nabla_{\overrightarrow{v}} \log\left( \|\exp(\overrightarrow{v})\|_1 \right) \rangle \right| \\
&\leq \|\overrightarrow{a} - \overrightarrow{b}\|_\infty \cdot \|\nabla_{\overrightarrow{v}} \log\left( \|\exp(\overrightarrow{v})\|_1 \right)\|_1 \\
&\overset{(ii)}{=} \|\overrightarrow{a} - \overrightarrow{b}\|_\infty
\end{aligned}
\tag{52}
$$

where (i) follows the mean value theorem and $\overrightarrow{v}$ is a convex combination between vectors $\overrightarrow{a}$ and $\overrightarrow{b}$. Moreover, (ii) is due to the equality that

$$
[\nabla_{\overrightarrow{v}} \log\left( \|\exp(\overrightarrow{v})\|_1 \right)]_i = \frac{\exp(v_i)}{\|\exp(\overrightarrow{v})\|_1}, \quad \|\nabla_{\overrightarrow{v}} \log\left( \|\exp(\overrightarrow{v})\|_1 \right)\|_1 = 1, \quad \forall \overrightarrow{v} \in \mathbb{R}^{|\mathcal{A}|}.
$$

Based on the property we show in (52), we could further analyze (51) as below:

$$
\begin{aligned}
\left| \log \pi_{k+1}(a|s) - \log \pi_{\theta_k}(a|s) \right| &\leq \left| \widehat{Q}_k(s,a) - Q_{\theta_k}(s,a) \right| + \left| \log\left( \sum_{\tilde{a} \in \mathcal{A}} \exp \widehat{Q}_k(s,\tilde{a}) \right) - \log\left( \sum_{\tilde{a} \in \mathcal{A}} \exp Q_{\theta_k}(s,\tilde{a}) \right) \right| \\
&\leq \left| \widehat{Q}_k(s,a) - Q_{\theta_k}(s,a) \right| + \max_{\tilde{a} \in \mathcal{A}} \left| \widehat{Q}_k(s,\tilde{a}) - Q_{\theta_k}(s,\tilde{a}) \right|.
\end{aligned}
\tag{53}
$$

If we take the maximum over all state-action pairs on the both sides of (53) and denote $\|\log \pi\|_\infty := \max_{s \in \mathcal{S}, a \in \mathcal{A}} |\log \pi(a|s)|$, then we obtain the following property:

$$
\left\| \log \pi_{k+1} - \log \pi_{\theta_k} \right\|_\infty \leq 2 \left\| \widehat{Q}_k - Q_{\theta_k} \right\|_\infty.
\tag{54}
$$

Recall that $\widehat{Q}_k$ is an approximation to the soft Q-function $Q_k$ where $\epsilon_{\mathrm{app}}$ is the approximation error, then we have:

$$
\left\| \log \pi_{k+1} - \log \pi_{\theta_k} \right\|_\infty \leq 2 \left\| \widehat{Q}_k - Q_{\theta_k} \right\|_\infty = 2 \left\| \widehat{Q}_k - Q_k + Q_k - Q_{\theta_k} \right\|_\infty \leq 2\epsilon_{\mathrm{app}} + 2 \left\| Q_k - Q_{\theta_k} \right\|_\infty.
\tag{55}
$$

In order to track the approximation error $\|\log \pi_{k+1} - \log \pi_{\theta_k}\|_\infty$, we could analyze the convergence between the $Q_k$ and $Q_{\theta_k}$ according to (55). Here, we define an auxiliary sequence $\{Q_{k+\frac{1}{2}}\}_{k \geq 0}$ in the conservative MDP where $Q_{k+\frac{1}{2}}$ is the soft Q-function under the reward parameter $\theta_k$ and the policy $\pi_{k+1}$ defined in (14). Then we have the following analysis:

$$
\begin{aligned}
\left\| Q_k - Q_{\theta_k} \right\|_\infty &= \left\| Q_k - Q_{\theta_k} + Q_{\theta_{k-1}} - Q_{\theta_{k-1}} + Q_{k-\frac{1}{2}} - Q_{k-\frac{1}{2}} \right\|_\infty \\
&\leq \left\| Q_{\theta_k} - Q_{\theta_{k-1}} \right\|_\infty + \left\| Q_{\theta_{k-1}} - Q_{k-\frac{1}{2}} \right\|_\infty + \left\| Q_k - Q_{k-\frac{1}{2}} \right\|_\infty \\
&\overset{(i)}{\leq} \left\| Q_{\theta_{k-1}} - Q_{k-\frac{1}{2}} \right\|_\infty + 2L_q \|\theta_k - \theta_{k-1}\| \\
&\overset{(ii)}{\leq} \gamma \left\| Q_{\theta_{k-1}} - Q_{k-1} \right\|_\infty + 2L_q \|\theta_k - \theta_{k-1}\| + \frac{2\gamma \epsilon_{\mathrm{app}}}{1 - \gamma}
\end{aligned}
\tag{56}
$$

where (i) follows (19a) in Lemma 4 and (26) in Lemma 7. Moreover, the inequality (ii) follows (27b) in Lemma 8. Recall that the update rule of the reward parameter $\theta$ is defined in (17), then the following result holds:

$$
\|\theta_k - \theta_{k-1}\| = \alpha \|g_{k-1}\| \overset{(i)}{=} \alpha \|h(\theta_{k-1}; \tau_{k-1}^{\mathrm{E}}) - h(\theta_{k-1}; \tau_{k-1}^{\mathrm{A}})\| \leq \alpha \|h(\theta_{k-1}; \tau_{k-1}^{\mathrm{E}})\| + \alpha \|h(\theta_{k-1}; \tau_{k-1}^{\mathrm{A}})\| \overset{(ii)}{\leq} \frac{2\alpha L_r}{1 - \gamma}
$$

where (i) follows the definition of the reward gradient estimator defined in (16). The inequality (ii) follows the definition of $h(\theta; \tau)$ in (16) and the bound gradient property $\|\nabla_\theta r(s,a; \theta)\| \leq L_r$ in (18) from Assumption 3. Recall that we have defined the constant $L_q := \frac{L_r}{1 - \gamma}$, then we obtain the following result:

$$
\|\theta_k - \theta_{k-1}\| \leq 2\alpha L_q.
\tag{57}
$$

Plugging (57) into (56), we can show that

$$
\left\| Q_k - Q_{\theta_k} \right\|_\infty \leq \gamma \left\| Q_{k-1} - Q_{\theta_{k-1}} \right\|_\infty + 4\alpha L_q^2 + \frac{2\gamma \epsilon_{\mathrm{app}}}{1 - \gamma}.
\tag{58}
$$

Summing (58) from $k = 1$ to $k = K$ and dividing $K$ on both sides, then we have the following result:

$$\frac{1}{K} \sum_{k=1}^{K} \left\| Q_k - Q_{\theta_k} \right\|_\infty \leq \frac{\gamma}{K} \sum_{k=0}^{K-1} \left\| Q_k - Q_{\theta_k} \right\|_\infty + 4\alpha L_q^2 + \frac{2\gamma \epsilon_{\text{app}}}{1 - \gamma}. \tag{59}$$

By rearranging (59), we obtain the following inequality:

$$\frac{1 - \gamma}{K} \sum_{k=1}^{K} \left\| Q_k - Q_{\theta_k} \right\|_\infty \leq \frac{\gamma}{K} \left( \left\| Q_0 - Q_{\theta_0} \right\|_\infty - \left\| Q_K - Q_{\theta_K} \right\|_\infty \right) + 4\alpha L_q^2 + \frac{2\gamma \epsilon_{\text{app}}}{1 - \gamma}. \tag{60}$$

Assuming the initial error $\left\| Q_0 - Q_{\theta_0} \right\|_\infty$ is bounded by a positive constant $\Delta_0$ where $\left\| Q_0 - Q_{\theta_0} \right\|_\infty \leq \Delta_0$ and dividing $1 - \gamma$ on the both sides of (60), then the following inequality holds:

$$\frac{1}{K} \sum_{k=1}^{K} \left\| Q_k - Q_{\theta_k} \right\|_\infty \leq \frac{\gamma}{K(1 - \gamma)} \Delta_0 + \frac{4\alpha L_q^2}{1 - \gamma} + \frac{2\gamma \epsilon_{\text{app}}}{(1 - \gamma)^2}. \tag{61}$$

Then we subtract $\frac{1}{K} \left\| Q_K - Q_{\theta_K} \right\|_\infty$ and add $\frac{1}{K} \left\| Q_0 - Q_{\theta_0} \right\|_\infty$ on both sides on (61), we obtain the following result:

$$\frac{1}{K} \sum_{k=0}^{K-1} \left\| Q_k - Q_{\theta_k} \right\|_\infty \leq \frac{\gamma}{K(1 - \gamma)} \Delta_0 + \frac{1}{K} \Delta_0 - \frac{1}{K} \left\| Q_K - Q_{\theta_K} \right\|_\infty + \frac{4\alpha L_q^2}{1 - \gamma} + \frac{2\gamma \epsilon_{\text{app}}}{(1 - \gamma)^2}$$

$$\leq \frac{1}{K(1 - \gamma)} \Delta_0 + \frac{4\alpha L_q^2}{1 - \gamma} + \frac{2\gamma \epsilon_{\text{app}}}{(1 - \gamma)^2}. \tag{62}$$

Plugging (62) into (55), then we obtain the convergence result of the policy estimates:

$$\frac{1}{K} \sum_{k=0}^{K-1} \left\| \log \pi_{k+1} - \log \pi_{\theta_k} \right\|_\infty \overset{(i)}{\leq} 2\epsilon_{\text{app}} + \frac{2}{K} \sum_{k=0}^{K-1} \left\| Q_k - Q_{\theta_k} \right\|_\infty$$

$$\overset{(ii)}{\leq} \left( 2 + \frac{4\gamma}{(1 - \gamma)^2} \right) \epsilon_{\text{app}} + \frac{2\Delta_0}{K(1 - \gamma)} + \frac{8\alpha L_q^2}{1 - \gamma} \tag{63}$$

where (i) follows (55) and (ii) is from (62). Recall that the stepsize $\alpha$ is defined to be $\alpha = \alpha_0 \times K^{-\frac{1}{2}}$, then we could obtain the convergence rate of the policy estimates as below:

$$\frac{1}{K} \sum_{k=0}^{K-1} \left\| \log \pi_{k+1} - \log \pi_{\theta_k} \right\|_\infty = \mathcal{O}(\epsilon_{\text{app}}) + \mathcal{O}(K^{-\frac{1}{2}}). \tag{64}$$

The relation (20a) has been proven. $\qquad \square$

### J.2 Proof of the convergence of reward parameter (20b)

*Proof.* In the section, we prove the convergence of the reward parameters $\{\theta_k\}_{k \geq 0}$. Recall that we have shown the Lipschitz property of the surrogate objective $\widehat{L}(\theta)$ in (19b) of Lemma 4. Based on the Lipschitz smooth property, we are able to have following analysis:

$$\widehat{L}(\theta_{k+1}) \overset{(i)}{\geq} \widehat{L}(\theta_k) + \left\langle \nabla \widehat{L}(\theta_k), \theta_{k+1} - \theta_k \right\rangle - \frac{L_c}{2} \left\| \theta_{k+1} - \theta_k \right\|^2$$

$$\overset{(ii)}{=} \widehat{L}(\theta_k) + \alpha \left\langle \nabla \widehat{L}(\theta_k), g_k \right\rangle - \frac{L_c \alpha^2}{2} \left\| g_k \right\|^2$$

$$= \widehat{L}(\theta_k) + \alpha \left\langle \nabla \widehat{L}(\theta_k), g_k - \nabla \widehat{L}(\theta_k) \right\rangle + \alpha \left\| \nabla \widehat{L}(\theta_k) \right\|^2 - \frac{L_c \alpha^2}{2} \left\| g_k \right\|^2 \tag{65}$$

where (i) is due to the Lipschitz smooth property in (19b) and (ii) follows the update rule of the reward parameter in (17). Recall that the gradient estimator $g_k$ is defined in (16). Due to the bound gradient of any reward parameter, we could obtain the following bound:

$$\left\| g_k \right\| \overset{(i)}{=} \left\| h(\theta_k; \tau_k^{\text{E}}) - h(\theta_k; \tau_k^{\text{A}}) \right\| \leq \left\| h(\theta_k; \tau_k^{\text{E}}) \right\| + \left\| h(\theta_k; \tau_k^{\text{A}}) \right\| \overset{(ii)}{\leq} \frac{2L_r}{1 - \gamma} \tag{66}$$

where (i) is from (16) and (ii) follows the bounded gradient of reward parameter in (18) of Assumption 3. Recall that we have defined the constant $L_q := \frac{L_r}{1-\gamma}$, then the following result holds:

$$\widehat{L}(\theta_{k+1}) \geq \widehat{L}(\theta_k) + \alpha\langle\nabla\widehat{L}(\theta_k), g_k - \nabla\widehat{L}(\theta_k)\rangle + \alpha\|\nabla\widehat{L}(\theta_k)\|^2 - 2L_c L_q^2\alpha^2. \tag{67}$$

After taking expectation on both sides of (67), we have

$$\mathbb{E}\big[\widehat{L}(\theta_{k+1})\big]$$
$$\geq \mathbb{E}\big[\widehat{L}(\theta_k)\big] + \alpha\mathbb{E}\big[\langle\nabla\widehat{L}(\theta_k), g_k - \nabla\widehat{L}(\theta_k)\rangle\big] + \alpha\mathbb{E}\big[\|\nabla\widehat{L}(\theta_k)\|^2\big] - 2L_c L_q^2\alpha^2$$
$$= \mathbb{E}\big[\widehat{L}(\theta_k)\big] + \alpha\mathbb{E}\big[\langle\nabla\widehat{L}(\theta_k), \mathbb{E}[g_k - \nabla\widehat{L}(\theta_k)|\theta_k]\rangle\big] + \alpha\mathbb{E}\big[\|\nabla\widehat{L}(\theta_k)\|^2\big] - 2L_c L_q^2\alpha^2$$
$$\overset{(i)}{=} \mathbb{E}\big[\widehat{L}(\theta_k)\big] + \alpha\mathbb{E}\Big[\Big\langle\nabla\widehat{L}(\theta_k), \mathbb{E}_{\tau^A\sim(\eta,\pi_{\theta_k},\widehat{P})}\big[\sum_{t=0}^{\infty}\gamma^t\nabla_\theta r(s_t,a_t;\theta_k)\big] - \mathbb{E}_{\tau^A\sim(\eta,\pi_{k+1},\widehat{P})}\big[\sum_{t=0}^{\infty}\gamma^t\nabla_\theta r(s_t,a_t;\theta_k)\big]\Big\rangle\Big]$$
$$\quad + \alpha\mathbb{E}\big[\|\nabla\widehat{L}(\theta_k)\|^2\big] - 2L_c L_q^2\alpha^2$$
$$\overset{(ii)}{\geq} \mathbb{E}\big[\widehat{L}(\theta_k)\big] - \underbrace{2\alpha L_q\mathbb{E}\Big[\Big\|\mathbb{E}_{\tau^A\sim(\eta,\pi_{\theta_k},\widehat{P})}\big[\sum_{t=0}^{\infty}\gamma^t\nabla_\theta r(s_t,a_t;\theta_k)\big] - \mathbb{E}_{\tau^A\sim(\eta,\pi_{k+1},\widehat{P})}\big[\sum_{t=0}^{\infty}\gamma^t\nabla_\theta r(s_t,a_t;\theta_k)\big]\Big\|\Big]}_{\text{T1: error term due to policy mismatch}}$$
$$\quad + \alpha\mathbb{E}\big[\|\nabla\widehat{L}(\theta_k)\|^2\big] - 2L_c L_q^2\alpha^2 \tag{68}$$

where (i) is from the definitions of the reward gradient estimator in (16) and the reward gradient expression in (15) of Lemma 3; (ii) follows $\|\nabla\widehat{L}(\theta_k)\| \leq 2L_q$ which could be proved according to (66). Then we analyze the error term due to policy mismatch as below:

$$\mathbb{E}\Big[\Big\|\mathbb{E}_{\tau^A\sim(\eta,\pi_{\theta_k},\widehat{P})}\big[\sum_{t=0}^{\infty}\gamma^t\nabla_\theta r(s_t,a_t;\theta_k)\big] - \mathbb{E}_{\tau^A\sim(\eta,\pi_{k+1},\widehat{P})}\big[\sum_{t=0}^{\infty}\gamma^t\nabla_\theta r(s_t,a_t;\theta_k)\big]\Big\|\Big]$$
$$\overset{(i)}{=} \mathbb{E}\Big[\Big\|\frac{1}{1-\gamma}\mathbb{E}_{(s,a)\sim d_{\widehat{P}}^{\pi_{\theta_k}}(\cdot,\cdot)}\big[\nabla_\theta r(s,a;\theta_k)\big] - \frac{1}{1-\gamma}\mathbb{E}_{(s,a)\sim d_{\widehat{P}}^{\pi_{k+1}}(\cdot,\cdot)}\big[\nabla_\theta r(s,a;\theta_k)\big]\Big\|\Big]$$
$$= \frac{1}{1-\gamma}\mathbb{E}\Big[\Big\|\sum_{s\in\mathcal{S},a\in\mathcal{A}}\nabla_\theta r(s,a;\theta_k)\cdot\big(d_{\widehat{P}}^{\pi_{\theta_k}}(s,a) - d_{\widehat{P}}^{\pi_{k+1}}(s,a)\big)\Big\|\Big]$$
$$\leq \frac{1}{1-\gamma}\mathbb{E}\Big[\sum_{s\in\mathcal{S},a\in\mathcal{A}}\|\nabla_\theta r(s,a;\theta_k)\|\cdot\big|d_{\widehat{P}}^{\pi_{\theta_k}}(s,a) - d_{\widehat{P}}^{\pi_{k+1}}(s,a)\big|\Big]$$
$$\overset{(ii)}{\leq} \frac{L_r}{1-\gamma}\mathbb{E}\Big[\sum_{s\in\mathcal{S},a\in\mathcal{A}}\big|d_{\widehat{P}}^{\pi_{\theta_k}}(s,a) - d_{\widehat{P}}^{\pi_{k+1}}(s,a)\big|\Big]$$
$$\overset{(iii)}{=} \frac{2L_r}{1-\gamma}\mathbb{E}\Big[\big\|d_{\widehat{P}}^{\pi_{\theta_k}}(\cdot,\cdot) - d_{\widehat{P}}^{\pi_{k+1}}(\cdot,\cdot)\big\|_{\text{TV}}\Big] \tag{69}$$

where (i) follows the definition of the visitation measures $d_{\widehat{P}}^{\pi_{\theta_k}}(\cdot,\cdot)$ and $d_{\widehat{P}}^{\pi_{k+1}}$, (ii) follows the bound gradient of reward parameter in (18) of Assumption 3 and (iii) follows the definition of the total variation norm. Recall that we have defined the constant $L_q := \frac{L_r}{1-\gamma}$. Due to the fact that $\pi_{\theta_k}$ and $\pi_{k+1}$ are softmax policies parameterized by $Q_{\theta_k}$ and $\widehat{Q}_k$, then we obtain the following result based

on Lemma 6:

$$\mathbb{E}\Big[\Big\|\mathbb{E}_{\tau^{\mathrm{A}}\sim(\eta,\pi_{\theta_k},\widehat{P})}\big[\sum_{t=0}^{\infty}\gamma^t\nabla_\theta r(s_t,a_t;\theta_k)\big] - \mathbb{E}_{\tau^{\mathrm{A}}\sim(\eta,\pi_{k+1},\widehat{P})}\big[\sum_{t=0}^{\infty}\gamma^t\nabla_\theta r(s_t,a_t;\theta_k)\big]\Big\|\Big]$$

$$= 2L_q\mathbb{E}\Big[\big\|d_{\widehat{P}}^{\pi_{\theta_k}}(\cdot,\cdot) - d_{\widehat{P}}^{\pi_{k+1}}(\cdot,\cdot)\big\|_{\mathrm{TV}}\Big]$$

$$\overset{(i)}{\leq} 2L_qC_d\mathbb{E}\Big[\big\|Q_{\theta_k} - \widehat{Q}_k\big\|\Big]$$

$$\overset{(ii)}{\leq} 2L_qC_d\sqrt{|\mathcal{S}|\times|\mathcal{A}|}\,\mathbb{E}\Big[\big\|Q_{\theta_k} - \widehat{Q}_k\big\|_\infty\Big]$$

$$= 2L_qC_d\sqrt{|\mathcal{S}|\times|\mathcal{A}|}\,\mathbb{E}\Big[\big\|Q_{\theta_k} - Q_k + Q_k - \widehat{Q}_k\big\|_\infty\Big]$$

$$\leq 2L_qC_d\sqrt{|\mathcal{S}|\times|\mathcal{A}|}\,\mathbb{E}\Big[\big\|Q_{\theta_k} - Q_k\big\|_\infty + \epsilon_{\mathrm{app}}\Big] \tag{70}$$

where (i) follows (25) in Lemma 6 and (ii) follows the conversion between Frobenius norm and the infinity norm. Plugging (70) into (68), we have

$$\mathbb{E}\big[\widehat{L}(\theta_{k+1})\big] \geq \mathbb{E}\big[\widehat{L}(\theta_k)\big] - 4\alpha C_dL_q^2\sqrt{|\mathcal{S}|\times|\mathcal{A}|}\,\mathbb{E}\Big[\big\|Q_{\theta_k} - Q_k\big\|_\infty + \epsilon_{\mathrm{app}}\Big] + \alpha\mathbb{E}\big[\|\nabla\widehat{L}(\theta_k)\|^2\big] - 2L_cL_q^2\alpha^2 \tag{71}$$

Rearranging (71) and dividing its both sides by $\alpha$, we could obtain the following result:

$$\mathbb{E}\big[\|\nabla\widehat{L}(\theta_k)\|^2\big] \leq \frac{1}{\alpha}\mathbb{E}\big[\widehat{L}(\theta_{k+1}) - \widehat{L}(\theta_k)\big] + 4C_dL_q^2\sqrt{|\mathcal{S}|\times|\mathcal{A}|}\,\mathbb{E}\Big[\big\|Q_{\theta_k} - Q_k\big\|_\infty + \epsilon_{\mathrm{app}}\Big] + 2\alpha L_cL_q^2. \tag{72}$$

Let us denote the constant $C_0 := 4C_dL_q^2\sqrt{|\mathcal{S}|\times|\mathcal{A}|}$. Summing (72) from $k=0$ to $k=K-1$ and dividing $K$ on the both sides, then we obtain

$$\frac{1}{K}\sum_{k=0}^{K-1}\mathbb{E}\big[\|\nabla\widehat{L}(\theta_k)\|^2\big] \leq \frac{\mathbb{E}\big[\widehat{L}(\theta_K) - \widehat{L}(\theta_0)\big]}{\alpha K} + \frac{C_0}{K}\sum_{k=0}^{K-1}\Big[\big\|Q_{\theta_k} - Q_k\big\|_\infty + \epsilon_{\mathrm{app}}\Big] + 2\alpha L_cL_q^2 \tag{73}$$

Recall that we bound the gap between the likelihood objective and the surrogate objective in (9). Then we have

$$|L(\theta) - \widehat{L}(\theta)| \leq \frac{\gamma C_v}{1-\gamma}\cdot\mathbb{E}_{(s,a)\sim d^{\mathrm{E}}(\cdot,\cdot)}\big[\|P(\cdot|s,a) - \widehat{P}(\cdot|s,a)\|_1\big] \leq \frac{2\gamma C_v}{1-\gamma} \tag{74}$$

where the last inequality is due to the fact that $\|P(\cdot|s,a) - \widehat{P}(\cdot|s,a)\|_1 \leq 2$ holds for any state-action pair $(s,a)$. Since $L(\theta)$ is the log-likelihood function which is always negative, we can show that the surrogate objective $\widehat{L}(\theta)$ is upper bounded under any reward parameter $\theta$. Denote a positive constant $C_1 := \frac{2\gamma C_v}{1-\gamma}$, we have

$$\widehat{L}(\theta) \leq L(\theta) + \frac{2\gamma C_v}{1-\gamma} \leq \frac{2\gamma C_v}{1-\gamma} = C_1. \tag{75}$$

Furthermore, considering $\widehat{L}(\theta_0)$ is the initial value of the surrogate objective, we can simply denote $C_2 := \widehat{L}(\theta_0)$. After plugging (62) into (73), we obtain the following result:

$$\frac{1}{K}\sum_{k=0}^{K-1}\mathbb{E}\big[\|\nabla\widehat{L}(\theta_k)\|^2\big] \leq \frac{C_1 - C_2}{\alpha K} + C_0\epsilon_{\mathrm{app}} + \frac{C_0\Delta_0}{K(1-\gamma)} + \frac{4\alpha C_0L_q^2}{1-\gamma} + \frac{2\gamma\epsilon_{\mathrm{app}}C_0}{(1-\gamma)^2} + 2\alpha L_cL_q^2.$$

Recall that the stepsize $\alpha$ is defined as $\alpha = \alpha_0\cdot K^{-\frac{1}{2}}$, then we can show the order of the convergence error as

$$\frac{1}{K}\sum_{k=0}^{K-1}\mathbb{E}\big[\|\nabla\widehat{L}(\theta_k)\|^2\big] = \mathcal{O}(K^{-\frac{1}{2}}) + \mathcal{O}(\epsilon_{\mathrm{app}}).$$

This completes the entire proof. $\qquad\square$

# K    Proof of Theorem 3

*Proof.* In this section, we prove the optimality guarantee when the reward function is linearly parameterized as $r(s, a; \theta) := \phi(s, a)^\top \theta$ where $\phi(s, a)$ is the feature vector of the state-action pair $(s, a)$. Based on the definition of the surrogate objective in (7) and the definition of the soft value function in (3a), we can rewrite the formulation (12) as below:

$$\max_\theta \quad \widehat{L}(\theta) := \mathbb{E}_{\tau^E \sim (\eta, \pi^E, P)} \Big[ \sum_{t=0}^\infty \gamma^t \Big( r(s_t, a_t; \theta) + U(s_t, a_t) \Big) \Big]$$

$$- \mathbb{E}_{\tau^A \sim (\eta, \pi_\theta, \widehat{P})} \Big[ \sum_{t=0}^\infty \gamma^t \Big( r(s_t, a_t; \theta) + U(s_t, a_t) + \mathcal{H}(\pi_\theta(\cdot|s_t)) \Big) \Big] \tag{76a}$$

$$s.t. \quad \pi_\theta := \arg\max_\pi \mathbb{E}_{\tau^A \sim (\eta, \pi, \widehat{P})} \Big[ \sum_{t=0}^\infty \gamma^t \Big( r(s_t, a_t; \theta) + U(s_t, a_t) + \mathcal{H}(\pi(\cdot|s_t)) \Big) \Big]. \tag{76b}$$

As a remark, there is a common term $\mathbb{E}_{\tau^A \sim (\eta, \pi, \widehat{P})} \Big[ \sum_{t=0}^\infty \gamma^t \Big( r(s_t, a_t; \theta) + U(s_t, a_t) + \mathcal{H}(\pi(\cdot|s_t)) \Big) \Big]$ in both (76a) and (76b). Then we can obtain the following formulation which is equivalent to (76):

$$\max_\theta \min_\pi \mathcal{L}(\theta, \pi) := \mathbb{E}_{\tau^E \sim (\eta, \pi^E, P)} \Big[ \sum_{t=0}^\infty \gamma^t \Big( r(s_t, a_t; \theta) + U(s_t, a_t) \Big) \Big]$$

$$- \mathbb{E}_{\tau^A \sim (\eta, \pi, \widehat{P})} \Big[ \sum_{t=0}^\infty \gamma^t \Big( r(s_t, a_t; \theta) + U(s_t, a_t) + \mathcal{H}(\pi(\cdot|s_t)) \Big) \Big]. \tag{77}$$

Based on the equivalence between (76) and (77), we further show that any stationary point $\tilde{\theta}$ in (76) together with its corresponding optimal policy $\pi_{\tilde{\theta}}$ consist of a saddle point $(\tilde{\theta}, \pi_{\tilde{\theta}})$ to the problem (77) when the reward is linearly parameterized. We summarize this statement in the following claim:

**Claim 1:** Assume the reward function is linearly parameterized, i.e., $r(s, a; \theta) := \phi(s, a)^\top \theta$ where $\phi(s, a)$ is the feature vector of the state-action pair $(s, a)$. Any stationary point $\tilde{\theta}$ in (76) together with its optimal policy $\pi_{\tilde{\theta}}$ consist of a saddle point $(\tilde{\theta}, \pi_{\tilde{\theta}})$ to the problem (77).

Here, we show the proof to Claim 1 as below.

Under the linearly parameterized reward function $r(s, a; \theta) := \phi(s, a)^\top \theta$, we can further rewrite $\mathcal{L}(\theta, \pi)$ as below:

$$\mathcal{L}(\theta, \pi) := \underbrace{\left\langle \theta, \mathbb{E}_{\tau^E \sim (\eta, \pi^E, P)} \Big[ \sum_{t=0}^\infty \gamma^t \phi(s_t, a_t) \Big] - \mathbb{E}_{\tau^A \sim (\eta, \pi, \widehat{P})} \Big[ \sum_{t=0}^\infty \gamma^t \phi(s_t, a_t) \Big] \right\rangle}_{\text{Term A: a linear function of the reward parameter } \theta}$$

$$+ \underbrace{\mathbb{E}_{\tau^E \sim (\eta, \pi^E, P)} \Big[ \sum_{t=0}^\infty \gamma^t U(s_t, a_t) \Big] - \mathbb{E}_{\tau^A \sim (\eta, \pi, \widehat{P})} \Big[ \sum_{t=0}^\infty \gamma^t \Big( U(s_t, a_t) + \mathcal{H}(\pi(\cdot|s_t)) \Big) \Big]}_{\text{Term B: independent of the reward parameter } \theta}.$$

$$\tag{78}$$

Note that the max-min objective $\mathcal{L}(\cdot, \cdot)$ is linear (thus concave) in the reward parameter $\theta$ given any fixed policy $\pi$. We can utilize this property to prove the statement in Claim 1.

Recall that a tuple $(\tilde{\theta}, \pi_{\tilde{\theta}})$ is called a saddle point of $\mathcal{L}(\cdot, \cdot)$ if the following condition holds:

$$\mathcal{L}(\theta, \pi_{\tilde{\theta}}) \leq \mathcal{L}(\tilde{\theta}, \pi_{\tilde{\theta}}) \leq \mathcal{L}(\tilde{\theta}, \pi) \tag{79}$$

for any other reward parameter $\theta$ and policy $\pi$. To show that $(\tilde{\theta}, \pi_{\tilde{\theta}})$ satisfies the condition (79), we prove the following conditions respectively:

$$\tilde{\theta} \in \arg\max_\theta \mathcal{L}(\theta, \pi_{\tilde{\theta}}), \tag{80a}$$

$$\pi_{\tilde{\theta}} \in \arg\min_\pi \mathcal{L}(\tilde{\theta}, \pi). \tag{80b}$$

Here, we first show that any stationary point $\tilde{\theta}$ of the surrogate objective $\widehat{L}(\cdot)$ satisfies the optimality condition (80a). Recall the gradient expression of $\widehat{L}(\cdot)$ in (15), any stationary point $\tilde{\theta}$ of the surrogate objective $\widehat{L}(\cdot)$ satisfies the following first-order condition:

$$\nabla\widehat{L}(\tilde{\theta}) = \mathbb{E}_{\tau^{\mathrm{E}} \sim (\eta, \pi^{\mathrm{E}}, P)}\Big[\sum_{t=0}^{\infty}\gamma^t\nabla_\theta r(s_t, a_t; \tilde{\theta})\Big] - \mathbb{E}_{\tau^{\mathrm{A}} \sim (\eta, \pi_{\tilde{\theta}}, \widehat{P})}\Big[\sum_{t=0}^{\infty}\gamma^t\nabla_\theta r(s_t, a_t; \tilde{\theta})\Big] = 0$$

Moreover, when the reward function is linearly parameterized as $r(s, a; \theta) := \phi(s, a)^\top\theta$, we obtain the following condition for any stationary point $\tilde{\theta}$:

$$\nabla\widehat{L}(\tilde{\theta}) = \mathbb{E}_{\tau^{\mathrm{E}} \sim (\eta, \pi^{\mathrm{E}}, P)}\Big[\sum_{t=0}^{\infty}\gamma^t\phi(s_t, a_t)\Big] - \mathbb{E}_{\tau^{\mathrm{A}} \sim (\eta, \pi_{\tilde{\theta}}, \widehat{P})}\Big[\sum_{t=0}^{\infty}\gamma^t\phi(s_t, a_t)\Big] = 0 \qquad (81)$$

Then we can go back to the formulation $\mathcal{L}(\cdot, \cdot)$ in (78). Given any fixed policy $\pi$, we can show the gradient of $\mathcal{L}(\theta, \pi)$ w.r.t. the reward parameter $\theta$ as below:

$$\nabla_\theta\mathcal{L}(\theta, \pi) = \mathbb{E}_{\tau^{\mathrm{E}} \sim (\eta, \pi^{\mathrm{E}}, P)}\Big[\sum_{t=0}^{\infty}\gamma^t\phi(s_t, a_t)\Big] - \mathbb{E}_{\tau^{\mathrm{A}} \sim (\eta, \pi, \widehat{P})}\Big[\sum_{t=0}^{\infty}\gamma^t\phi(s_t, a_t)\Big]. \qquad (82)$$

Due to the first-order condition we show in (81), we obtain the following result:

$$\nabla_\theta\mathcal{L}(\theta = \tilde{\theta}, \pi = \pi_{\tilde{\theta}}) = 0. \qquad (83)$$

Recall that we have shown $\mathcal{L}(\cdot, \cdot)$ is concave in terms of the reward parameter $\theta$ given any fixed policy $\pi$. Due to the concavity as shown in (78) and the condition in (83), we have completed the proof of the optimality condition (80a).

Then we prove the optimality condition (80b). Recall that $\pi_{\tilde{\theta}}$ is the optimal policy defined in (76b) under the reward parameter $\theta$. By observing the objective $\mathcal{L}(\cdot, \cdot)$, we obtain the following result:

$$\mathcal{L}(\theta, \pi) := \underbrace{\mathbb{E}_{\tau^{\mathrm{E}} \sim (\eta, \pi^{\mathrm{E}}, P)}\Big[\sum_{t=0}^{\infty}\gamma^t\Big(r(s_t, a_t; \theta) + U(s_t, a_t)\Big)\Big]}_{\text{Term } \mathrm{I}_1: \text{ independent of } \pi}$$

$$- \underbrace{\mathbb{E}_{\tau^{\mathrm{A}} \sim (\eta, \pi, \widehat{P})}\Big[\sum_{t=0}^{\infty}\gamma^t\Big(r(s_t, a_t; \theta) + U(s_t, a_t) + \mathcal{H}(\pi(\cdot|s_t))\Big)\Big]}_{\text{Term } \mathrm{I}_2: \text{ a function of the policy } \pi}. \qquad (84)$$

Here, we can observe that the second term in (84) is exactly the objective in (76b). Moreover, since $\pi_{\tilde{\theta}}$ is the optimal policy defined in (76b) under reward parameter $\tilde{\theta}$, we obtain the result that

$$\pi_{\tilde{\theta}} \in \arg\min_\pi \mathcal{L}(\tilde{\theta}, \pi)$$

which completes the proof to show the opitmality condition (80b). Hence, we obtain that any stationary point $\tilde{\theta}$ of the surrogate objective $\widehat{L}(\cdot, \cdot)$ together with its optimal policy $\pi_{\tilde{\theta}}$ is a saddle point of $\mathcal{L}(\cdot, \cdot)$.

Therefore, we complete the proof of Claim 1.

Next, we show that for any saddle point of (77), its reward parameter is a globally optimal solution to (76). We summarize this statement in the following claim:

**Claim 2:** For any saddle point $(\tilde{\theta}, \pi_{\tilde{\theta}})$ of the saddle point problem (77), the reward parameter $\tilde{\theta}$ is a globally optimal solution to the surrogate objective $\widehat{L}(\cdot)$ in (76).

Here, we start to show the proof to Claim 2 as below.

Given a saddle point $(\tilde{\theta}, \pi_{\tilde{\theta}})$ of $\mathcal{L}(\cdot, \cdot)$ defined in (77), we have the following property that

$$\min_\pi\max_\theta\mathcal{L}(\theta, \pi) \leq \max_\theta\mathcal{L}(\theta, \pi_{\tilde{\theta}}) \stackrel{(i)}{=} \mathcal{L}(\tilde{\theta}, \pi_{\tilde{\theta}}) \stackrel{(ii)}{=} \min_\pi\mathcal{L}(\tilde{\theta}, \pi) \leq \max_\theta\min_\pi\mathcal{L}(\theta, \pi) \qquad (85)$$

where $(i)$ follows the optimality condition (80a) and $(ii)$ follows the optimality condition (80b). According to the minimax inequality, we always have the following condition that

$$\max_{\theta} \min_{\pi} \mathcal{L}(\theta, \pi) \le \min_{\pi} \max_{\theta} \mathcal{L}(\theta, \pi). \tag{86}$$

Through combining the saddle point inequality (85) and the minimax inequality (86), we obtain the following equality:

$$\min_{\pi} \max_{\theta} \mathcal{L}(\theta, \pi) = \max_{\theta} \mathcal{L}(\theta, \pi_{\tilde{\theta}}) = \mathcal{L}(\tilde{\theta}, \pi_{\tilde{\theta}}) = \min_{\pi} \mathcal{L}(\tilde{\theta}, \pi) = \max_{\theta} \min_{\pi} \mathcal{L}(\theta, \pi). \tag{87}$$

Therefore, for any saddle point $(\tilde{\theta}, \pi_{\tilde{\theta}})$, we obtain the following property of the reward parameter $\tilde{\theta}$ and the corresponding policy $\pi_{\tilde{\theta}}$:

$$\tilde{\theta} \in \arg\max_{\theta} \min_{\pi} \mathcal{L}(\theta, \pi), \tag{88a}$$

$$\pi_{\tilde{\theta}} \in \arg\min_{\pi} \max_{\theta} \mathcal{L}(\theta, \pi). \tag{88b}$$

Due to the expression of the surrogate objective $\widehat{L}(\cdot)$ in (76) and the objective $\mathcal{L}(\cdot, \cdot)$ in (77), we have the following equality holds for any reward parameter $\theta$:

$$\widehat{L}(\theta) = \min_{\pi} \mathcal{L}(\theta, \pi). \tag{89}$$

Combining (88a) and (89), we obtain the following result:

$$\tilde{\theta} \in \arg\max_{\theta} \min_{\pi} \mathcal{L}(\theta, \pi) = \arg\max_{\theta} \widehat{L}(\theta). \tag{90}$$

According to the property we shown in (90), we obtain that for any saddle point $(\tilde{\theta}, \pi_{\tilde{\theta}})$ of $\mathcal{L}(\cdot, \cdot)$, the reward parameter $\tilde{\theta}$ is a globally optimal solution of the surrogate objective $\widehat{L}(\cdot)$ in (76).

Therefore, we complete the entire proof of Claim 2.

Through combining Claim 1 - Claim 2, we obtain that when the reward function is linearly parameterized, any stationary point of the surrogate problem (76) is a global optimum.

Let's denote the optimal reward parameters associated with $L(\cdot)$ and $\widehat{L}(\cdot)$ as below, respectively:

$$\theta^* \in \arg\max_{\theta} L(\theta), \quad \hat{\theta} \in \arg\max_{\theta} \widehat{L}(\theta). \tag{91}$$

Recall that in Theorem 1, we have shown for any $\epsilon \in (0, 2)$, suppose there are more than $N$ data points on each state-action pair $(s, a) \in \Omega$ and the number of transition dataset $\mathcal{D}$ satisfies:

$$\#\textit{transition samples} \ge |\Omega| \cdot N \ge \frac{4\gamma^2 \cdot C_v^2 \cdot c^2 \cdot |\Omega| \cdot |\mathcal{S}^{\mathrm{E}}|}{(1-\gamma)^2 \varepsilon^2} \ln\left(\frac{|\Omega|}{\delta}\right) \tag{92}$$

where $c$ is a constant dependent on $\delta$. With probability greater than $1 - \delta$, the following result holds:

$$L(\theta^*) - L(\hat{\theta}) \le \varepsilon.$$

Through combining Claim 1 - Claim 2, we have already shown that any stationary point $\tilde{\theta}$ is a global optimum of the surrogate problem (76) when the reward function is linearly parameterized. Therefore, we obtain that when the number of transition samples satisfies (92), any stationary point $\tilde{\theta}$ of the surrogate problem (76) is an epsilon-optimal solution to the maximum likelihood estimation problem (2). For any stationary point $\tilde{\theta}$ of the surrogate problem (76), with probability greater than $1 - \delta$, it holds that

$$L(\theta^*) - L(\tilde{\theta}) \le \varepsilon.$$

This completes the entire proof of Theorem 3. $\qquad\square$

