# OpenReview forum: "When Demonstrations meet Generative World Models: A Maximum Likelihood Framework for Offline Inverse Reinforcement Learning"
_NeurIPS.cc/2023/Conference — NeurIPS 2023 oral_

### Official Review · Reviewer_VRGm · 2023-06-27

**Soundness:** 4 excellent
**Presentation:** 3 good
**Contribution:** 4 excellent
**Rating:** 8
**Confidence:** 3

**Summary:**

This paper focuses on learning from demonstration via offline inverse reinforcement learning (IRL). Offline IRL suffers a similar problem as offline reinforcement learning (RL) and imitation learning (IL), where the policy cannot generalize well on unseen states and actions---this problem is known as distribution shift. To address this problem, this paper proposes to first learn a dynamic model, and formulates a maximum likelihood (ML) objective to simultaneously recovers both the reward function and the policy. Notably, the policy is optimized using a maximum entropy objective along with pessimism based on the uncertainty of the learned dynamic model. This paper provides PAC-style bounds to quantify the amount of samples required to achieve $\varepsilon$-optimal solution to the MLE objective. The paper further provides an algorithm that obtains such a $\varepsilon$-optimal solution under specific assumptions. Finally, this paper provides empirical evaluation on the D4RL benchmarks.

## Contributions
- A new MLE objective for recovering a policy that is close to the expert policy.
- A theoretical analysis that describes the statistical guarantees of the objective
- An algorithm that obtains to near-optimal solution under linear parameterization of the reward function
- An empirical evaluation on standard benchmark, D4RL, that indicates statistically significant improvement over existing baselines in majority of the tasks.

**Strengths:**

- The paper is well written and easy to follow in general---I particularly appreciate the presentation on providing formal statements followed by the high-level intuitions.
- The paper proposes a novel formulation for offline inverse reinforcement learning.
- The paper provides numerous theoretical justifications and an algorithm that is inspired by said analyses.

**Weaknesses:**

- In practice, how do we ensure assumption 2 (ergodicity)? It seems like this assumption actually "hides" some part of the coverage requirement?
- I am curious as to how the MLE objective connects to the policy error---I completely understand that if $\pi_\theta = \pi^E$ then the policy error is zero. However, it does not seem to me that achieving $\varepsilon$-error on the MLE (i.e. $L(\theta^*) - L(\hat \theta) \leq \varepsilon$) does not directly tell us the policy error.
- I think the training description for BC is somewhat vague---on page 8, line 314: what exactly does it mean by "train the algorithm until convergence"? Do we have some form of validation checking for BC? [1, 2, 3] have results regarding how BC would perform based on specific validation. Secondly, was there any hyperparameter search on BC, ValueDICE, and CLARE?
- Regarding the experiment on recovered rewards, what is the performance if we were to fix the reward to 0? Isn't it better if we were to consider the correlation between the true reward function and the obtained reward function?

## References
[1]: Hussenot, L., Andrychowicz, M., Vincent, D., Dadashi, R., Raichuk, A., Ramos, S., ... & Pietquin, O. (2021, July). Hyperparameter selection for imitation learning. In International Conference on Machine Learning (pp. 4511-4522). PMLR.
[2]: Mandlekar, A., Xu, D., Wong, J., Nasiriany, S., Wang, C., Kulkarni, R., ... & Martín-Martín, R. (2021). What matters in learning from offline human demonstrations for robot manipulation. arXiv preprint arXiv:2108.03298.
[3]: Ablett, T., Chan, B., & Kelly, J. (2023). Learning from Guided Play: Improving Exploration for Adversarial Imitation Learning with Simple Auxiliary Tasks. IEEE Robotics and Automation Letters.

**Questions:**

- On page 3, equation 2a: $\theta$ parameterizes only the reward function, and $\pi_\theta$ corresponds to the policy obtained when $r$ is parameterized by $\theta$, correct?
- On page 5, proposition 1: Is there any result regarding larger $\varepsilon$? It seems like we may want to sacrifice this approximation error.

## Possible Typos
- Page 3, line 93: gamma \in (0, 1)?
- Page 3, line 95: The $P$ should not be the same as the transition dynamics right? I feel we should be clear that we are either overloading notation or use another symbol.
- Page 5, equation 11: What's $\epsilon$? Is it the same as $\varepsilon$?
- Page 8, line 303: Missing space after "2)"

**Limitations:**

- The paper's proposed method requires uncertainty estimation which is still an active research problem, especially in the context of neural networks---as far as I understand the paper leverages existing work that lacks theoretical guarantee.

---

> ### Author Rebuttal · Authors · 2023-08-09
>
> We thank the reviewer for the detailed review of the paper and the valuable feedback. Below, we address the reviewer's comments in a point-by-point manner.
>
> **Response to Weakness 1:** We thank the reviewer for this good question. As we discussed in Section. 6, we constructed an ensemble of estimated dynamics models as $ \\{ \hat{P}^{i}\_{\phi,\varphi}(\cdot|s,a) = \mathcal{N}( \mu^{i}\_{\phi}(s,a), \Sigma^{i}\_{\varphi}(s,a) ) \\}\_{i=1}^N$ where each one models the location of the next state by Gaussian distribution. Under Gaussian distributions, it is achievable to eventually get from every state to every other state with positive probability. Hence, the ergodicity holds in our estimated dynamics model in practice.
>
> **Response to Weakness 2:** We appreciate the reviewer for raising this insightful question. The general answer is that $\varepsilon$-error on the MLE implies that the recovered policy $\pi_{\hat{\theta}}$ is $\varepsilon$-close to the expert policy $\pi^E$ measured by the KL divergence.
>
> In our formulation eq (2a) - (2b), the MLE objective follows $L(\theta) := \mathbb{E}\_{\tau^E \sim (\eta,\pi^E,P)}[ \sum\_{t=0}^{\infty} \gamma^t \log \pi\_{\theta}(a_t|s_t) ]$. According to the definition of the state visitation measure $d^E(s)$ under the expert policy $\pi^E$ where $d^E(s) := (1-\gamma) \sum\_{t=0}^{\infty} \gamma^t P^{\pi^E}(s_t = s|s_0 \sim \eta)$, we can rewrite the MLE objective as below: $L(\theta) := \mathbb{E}\_{\tau^E \sim (\eta,\pi^E,P)}[ \sum\_{t=0}^{\infty} \gamma^t \log \pi\_{\theta}(a_t|s_t) ] = \frac{1}{1-\gamma}\mathbb{E}\_{s \sim d^E(\cdot), a \sim \pi^E(\cdot|s)}[ \log \pi\_{\theta}(a|s) ].$
>
> Therefore, the $\varepsilon$-error on the MLE implies that $L(\theta^*) - L(\hat{\theta}) = \frac{1}{1-\gamma}\mathbb{E}\_{s\sim d^E(\cdot), a \sim \pi^E(\cdot|s)}[\log \pi\_{\theta^*}(a|s) - \log \pi\_{\hat{\theta}}(a|s)] = \frac{1}{1-\gamma}\mathbb{E}\_{s\sim d^E(\cdot), a \sim \pi^E(\cdot|s)}[\log  \frac{\pi\_{\theta^*}(a|s)}{\log \pi\_{\hat{\theta}}(a|s)} ] \leq \varepsilon.$ When the expert trajectories are consistent with the optimal policy under a ground truth reward parameter $\theta^*$, we have $\pi^E = \pi\_{\theta^*}$. Due to this property, we can show $L(\theta^*) - L(\hat{\theta}) = \frac{1}{1-\gamma}\mathbb{E}\_{s\sim d^E(\cdot), a \sim \pi^E(\cdot|s)}[\log  \frac{\pi\_{\theta^*}(a|s)}{\log \pi\_{\hat{\theta}}(a|s)} ] = \frac{1}{1-\gamma}\mathbb{E}\_{s\sim d^E(\cdot), a \sim \pi^E(\cdot|s)}[\log  \frac{\pi^E(a|s)}{\log \pi\_{\hat{\theta}}(a|s)} ] = \frac{1}{1-\gamma}\mathbb{E}\_{s\sim d^E(\cdot)} [ D\_{KL}( \pi^E(\cdot|s) || \pi\_{\hat{\theta}}(\cdot|s) ) ] \leq \varepsilon.$ Hence, we can show the $\varepsilon$-error on the MLE implies that the recovered policy $\pi\_{\hat{\theta}}$ is $\varepsilon$-close to the expert policy $\pi^E$ measured by the KL divergence.
>
> **Response to Weakness 3:** We appreciate the reviewer’s comments. Regarding the training of BC, we assess the checkpoints generated throughout the training process and monitor the performance of the resulting policies at every few updates. Once the performance of BC stops to achieve further improvement (in terms of the average rewards in episodes) within 20 training epochs, we will use the updated policy to generate ten rollout episodes and then record the average reward of the rollout episodes as the final performance measure.
>
> Moreover, it is important to note that for benchmark algorithms BC, ValueDice, and CLARE, we have utilized their official implementations, incorporating their default hyper-parameters, which have been fine-tuned beforehand.
>
> Below, we provide the sources for the official implementations of the benchmark algorithms, which we have mentioned in Appendix A (the section of experiment details):
>
> BC, ValueDice: https://github.com/google-research/google-research/tree/master/value_dice
>
> CLARE: https://openreview.net/forum?id=5aT4ganOd98
>
> **Response to Weakness 4:** We appreciate this suggestion. The numerical results are included in the PDF in global response (see https://openreview.net/forum?id=oML3v2cFg2&noteId=oCZvqhc8PI).
>
> **Response to Question 1:** Yes. When the reward function is $r(\cdot,\cdot;\theta)$, we use $\pi_{\theta}$ to denote the optimal policy obtained in the estimated dynamics model.
>
> **Response to Question 2:** We appreciate the reviewer for this good question. The reason we define $\varepsilon \in (0,2)$ is because we use $\varepsilon$ to bound $\mathbb{E}\_{(s,a)\sim d^E(\cdot,\cdot)}[|| P(\cdot|s,a) -  \hat{P}(\cdot|s,a)||_1]$ and the maximum dynamics mismatch error under L1 norm is bounded by 2.
>
> Based on the definition of the L1 distance, we have
> $|| P(\cdot | s,a) - \hat{P}(\cdot | s,a) ||_1 = \sum\_{s^\prime \in \mathcal{S}} | P(s^\prime | s,a) - \hat{P}(s^\prime | s,a) | \in [0,2]$. When two distributions perfectly match, their L1 distance is 0, while a complete mismatch between distributions results in an L1 distance of 2.
>
> Hence, when analyzing sample complexity, we define $\varepsilon \in (0,2)$ to bound the error $\mathbb{E}\_{(s,a)\sim d^E(\cdot,\cdot)}[|| P(\cdot|s,a) -  \hat{P}(\cdot|s,a)||_1]$.
>
> **Response to Typo 1:** We will explicitly define $\gamma \in (0,1)$ in our paper.
>
> **Response to Typo 2:** We appreciate the comments. There is one typo in the term $P(s_t = s | s_0 \sim \eta)$, which should be corrected as $P^{\pi}(s_t = s | s_0 \sim \eta ).$ Under any fixed policy , the term $P^{\pi}(s_t = s | s_0 \sim \eta )$ denotes the probability that $s_t = s$ at time step t when $s_0$ is sampled from $\eta(\cdot)$ and the actions in the MDP are sampled from $\pi$.
>
> Hence, the state-action visitation measure $d^{\pi}\_{P}(s,a)$ in line 95 follows: $d^{\pi}_{P}(s,a):= (1 - \gamma) \pi(a|s) \sum\_{t=0}^{\infty} \gamma^t P^{\pi}(s_t = s | s_0 \sim \eta ).$
>
> **Response to Typo 3:** We appreciate the comments. $\epsilon$ is a typo and we should correct it as $\varepsilon$.
>
> **Response to Typo 4:** We will include the space after "2)".

---

> > ### Comment · Reviewer_VRGm · 2023-08-11
> >
> > Thank you for the detailed answers and the extra experiments.
> > I believe the authors have addressed all my questions or concerns (especially for weakness 2, my understanding is that this will translate to policy error due to [1]), I am happy to increase the score from 6 to 8.
> >
> > [1]: Xu, Tian, Ziniu Li, and Yang Yu. "Error bounds of imitating policies and environments." Advances in Neural Information Processing Systems 33 (2020): 15737-15749.

---

> > > ### Author Response · Authors · 2023-08-11
> > > **Many thanks for your comments and positive feedback**
> > >
> > > We truly appreciate your detailed review and insightful comments. We will discuss this paper and the translation to policy errors in our revised version.

---

### Official Review · Reviewer_1VXD · 2023-06-27

**Soundness:** 3 good
**Presentation:** 2 fair
**Contribution:** 2 fair
**Rating:** 6
**Confidence:** 4

**Summary:**

This paper addressed the issue of covariate shift in offline imitation learning. The authors extended the uncertainty-regularized model-based offline RL to the imitation learning setting. The key idea is to first learn transition dynamics from samples, and then solve an optimization problem that jointly seeks a policy such that it optimizes the learned transition dynamics accompanied by a reward model and maximizes the log-likelihood of actions in data. The authors provide theoretical guarantees for the maximization of action log-likelihood and empirical results for the learned policy.

**Strengths:**

1. The issue of covariate shift is indeed important for offline IRL.
2. The efficacy of the algorithm is partially supported by empirical results.
3. Analysis is provided for the model-based part of this algorithm.

**Weaknesses:**

1. The paper is somewhat hard to follow. See questions below.
2. The effect of overcoming the distributional shift is not emphasized in the experiment section. None of the experiments was carried out on small datasets where the coverage of state–action space is limited. In fact, the medium datasets in D4RL contain 1M transitions, and the medium-expert versions contain 2M transitions. The datasets for results in Figure 2 contain 5000 expert demonstrations, which might correspond to 5M transitions if each expert trajectory contains 1000 transitions.
3. The paper does not have an informative conclusion part.

**Questions:**

1. Why do you include discounting in the log-likelihood term?
2. Why do you include the entropy term in eq. 3a?
3. The theoretical analysis relies on the relation between optimal policy and optimal soft Q-function (eq.4 and eq. 14). But as mentioned in line 226--230, practical implementations utilize an actor-critic framework to approximate the optimal policy. How does this approximation affect the analysis?
4. Eq. 15 shows a clear resemblance with the reward maximization part in max-entropy IRL (eq. (1)). It suggests that the proposed algorithm seems to replace the online interaction of max-entropy IRL with sample generation using the learned dynamics. Then, what is the motivation to maximize the log-likelihood of expert actions?

A suggestion.
One concern of this approach is that to estimate the transition dynamics we need a sufficient amount of data, but the covariate shift issue occurs when we only have a small amount of data. I would suggest the authors include results on a few expert demonstrations (e.g. 10~100) to verify their claim.

**Limitations:**

No. there is no discussion on limitations.

---

> ### Author Rebuttal · Authors · 2023-08-09
>
> We thank the reviewer for the detailed review of the paper and the valuable feedback. Below, we address the reviewer's comments in a point-by-point manner.
>
> **Response to Weakness 1:** We will address your comments to improve the readability of the paper.
>
> **Response to Weakness 2:** We appreciate the reviewer’s comments. In our paper, in fact, we have utilized small datasets to evaluate the performance of our proposed algorithm when the coverage of the expert-visited state-action space is limited. To address some potential misunderstanding regarding our experiment, we would like to clarify that **in our work, one expert demonstration corresponds to one transition sample $(s, a, s^\prime)$ from the expert trajectories.** We have explicitly mentioned this in Appendix A and the caption of Table 2.  In Fig. 2 -4, we respectively report the performance of our proposed algorithm when five trajectories (5,000 transition samples), one trajectory (1,000 transition samples) and ten trajectories (10,000 transition samples) are available.
>
> **Response to Weakness 3:** We appreciate the reviewer’s suggestion. We provide a conclusion section in the global response and we kindly refer the reviewer to check it (see https://openreview.net/forum?id=oML3v2cFg2&noteId=oCZvqhc8PI).
>
> **Response to Question 1:** We appreciate the reviewer for raising this insightful question. The general answer that the discounted log-likelihood objective is derived from the dual problem of the classic formulation maximum entropy (MaxEnt) IRL.
>
> In the literature of MaxEnt IRL [1], the problem aims to learn a (linearly parameterized) reward that can induce a policy to achieve the same expected reward as the expert trajectories while maximizing its entropy. Under infinite horizon MDP, the existing results have shown that the dual problem of MaxEnt IRL is the maximization of the “discounted” log-likelihood over expert trajectories on an optimal policy which solves an underlying entropy-regularized MDP (see Theorem 1 in [2]). This fundamental result justifies the use of discounting in the log-likelihood term.
>
> **Response to Question 2:** The general answer is that the entropy term in our formulation eq. (2b) is translated from the maximum entropy objective of MaxEnt IRL.
>
> As we discussed in the response to Question 1, in online IRL with linear parameterized reward, it has been shown that the dual problem of MaxEnt IRL is a maximum likelihood formulation of IRL where the optimal policy is modeled as a solution to a entropy-regularized MDP. Here, in the derivation of maximum likelihood formulation of IRL, the entropy term in the objective of MaxEnt IRL has been translated to the model of agent’s behavior. Hence, in this paper, when we propose the maximum likelihood formulation of IRL with nonlinear reward parameterization in the offline setting, we include the entropy term in the definition of the soft value function and soft Q-function in eq. 3a - 3b.
>
> **Response to Question 3:** We would like to clarify that our theoretical analysis aligns with the algorithmic framework of our practical implementations, since we have taken the approximation error into account in our theoretical analysis.
>
> More specifically, we have considered the approximation error from the soft actor-critic (SAC) algorithm [3] into our policy optimization subroutine eq. 13 - 14. Our theoretical analysis does not require the policy or the soft Q-function to be optimal at each step, therefore, running a few steps of the soft actor-critic algorithm will be sufficient. At each step k, we first run policy evaluation steps (critic steps) to approximate the corresponding soft Q-function $Q_k$ of the current policy $\pi_k$ by $\hat{Q}\_k$, as shown in eq. 13. Then we run the soft policy iteration (action steps) to obtain the updated policy $\pi_{k+1}(a|s) \propto \exp \hat{Q}_k(s,a)$, as shown in eq. 14. Note that our policy optimization subroutine matches the practical implementation of SAC. Moreover, the approximation error $||\hat{Q}_k - Q_k ||\_{\infty}$ in estimating the soft Q-function has been explicitly considered in our theoretical analysis (see Theorem 2).
>
> **Response to Question 4:** The general answer is that the maximum likelihood formulation is a broader problem formulation compared with MaxEnt IRL. First of all, as we discussed in the response to Question 1, the MaxEnt IRL is based on the online IRL setting with linear reward. In this case, the maximum likelihood IRL is the dual problem to MaxEnt IRL. Hence, we can find resemblance between the maximum likelihood IRL and MaxEnt IRL. However, since MaxEnt IRL is limited to the online setting with linear reward, its formulation is incompatible with broader problems. Therefore, the maximum likelihood IRL formulation offers greater flexibility to model IRL problems in a broader scope. It can be applied effectively to scenarios involving either linear or nonlinear reward parameterization, as well as online or offline settings.
>
>
> **Response to Reviewer's Suggestion:** We appreciate the reviewer’s suggestion. As we clarified in our response to Weakness 2, one expert demonstration in our paper means one transition sample $(s, a, s^\prime)$ collected from an expert trajectory. In our numerical results, we have considered the setting that only 1 / 5 / 10 expert trajectories are available.
>
> **Response to Limitations:** We have include them in the section of Limitations and broader impacts in Appendix.
> ______
> [1] Ziebart, Brian D. Modeling purposeful adaptive behavior with the principle of maximum causal entropy. Carnegie Mellon University, 2010.
>
> [2] Zeng, Siliang, et al. "Maximum-likelihood inverse reinforcement learning with finite-time guarantees." Advances in Neural Information Processing Systems 35 (2022): 10122-10135.
>
> [3] Haarnoja, Tuomas, et al. "Soft actor-critic: Off-policy maximum entropy deep reinforcement learning with a stochastic actor." International conference on machine learning. PMLR, 2018.

---

> > ### Comment · Reviewer_1VXD · 2023-08-15
> >
> > Thanks for the replies. I have adjusted my evaluation based on them.

---

> > > ### Author Response · Authors · 2023-08-16
> > > **Many thanks for your positive feedback to our response**
> > >
> > > We truly thank you for taking time to review our paper and recognizing the contributions of this work!

---

### Official Review · Reviewer_2Qwv · 2023-07-02

**Soundness:** 3 good
**Presentation:** 3 good
**Contribution:** 3 good
**Rating:** 7
**Confidence:** 3

**Summary:**

Offline inverse reinforcement learning (IRL) is a method for finding an unknown reward function optimized by an agent from demonstrations using only a finite dataset. The most common framework is maximum entropy (MaxEnt) IRL, which attempts to find a reward function that induces a policy which achieves the same expected reward as the trajectories from the expert demonstrations while maximizing its entropy. Prior work has shown that this is equivalent to finding a policy which maximizes the likelihood of the demonstrations under the constraint that this policy comes from solving a MaxEnt RL problem. This formulation as a bi-level optimization problem reduces the computational burden that results from alternating between finding the policy and updating the reward. However, it requires access to the true dynamics of the environment, which is incompatible with the offline IRL setup. Instead, this work proposes to learn the dynamics model in an uncertainty-aware fashion and incorporate a measure of this uncertainty in the learned reward function. This results in a two-stage procedure: 1) fitting a dynamics model from transition samples in the dataset and 2) recovering the reward function using the maximum likelihood (ML) formulation of the IRL problem. To perform the second step, the authors propose a novel decomposition of the upper-level objective, which consists of a surrogate objective that is more computationally tractable to optimize. The authors provide statistical guarantees about the optimality of the recovered reward function in terms of dataset coverage, a concept common in offline RL. Importantly, their bounds depend on dataset coverage on expert-visited state-action pairs, not the full joint space.

**Strengths:**

- Offline inverse RL is an important area for tackling challenging sequential decision making problems in potentially safety-critical applications.
- The paper is well organized and clearly written. It does a good job explaining the novelty and results and provides enough information to support its claims.
- The paper presents an extensive experimental evaluation on several benchmarks, comparing to both model-based and model-free offline IRL algorithms and existing imitation learning approaches. Their algorithm outperforms these baselines in most cases.
- The authors provide a nice analysis of surrogate objective and its relation to the true upper-level objective. This provides a nice motivation for optimizing the surrogate instead, which is more computationally tractable.
- The authors present a nice optimality guarantee of the stationary point of their algorithm in terms of the surrogate objective in the case where the reward function is linear in a feature vector of states and actions. They also relate this stationary point to the true optimal solution of the original problem.
- The additional reward transfer experiments indicate that the learned reward function may transfer to dynamics models trained on different state-action distributions. This appears to hold even if the state-action coverage used to train the reward function is close to expert-visited states.

**Weaknesses:**

- The alternating optimization scheme discussed in this work appears to be identical to that presented in [21]. If true, that is fine, as the main contribution of the work lies in modeling the conservative MDP and providing novel bounds in the offline setting. However, it should be made explicit in the paper and mentioned in the contributions.
- Theorem 2 seems very similar to Theorem 5.4 in [21], except that the Q function approximation error is considered explicitly. If this is true, it should be discussed that this is the novelty in the text.
- Section 6 should discuss the differences in the three dataset types used, as the current text does not explain what they entail. This makes it difficult to understand the performance of the proposed algorithm in each setting without carefully looking at the Appendix. It should also talk about the purpose of using these different datasets. From the Appendix, it appears that they are only used to train the dynamics model. Thus, they are evaluating the effect of dataset coverage around the expert on performance. This should be explicitly discussed in the paper. I know space is limited, but these are important details that should be in the main text.
- A minor comment is that it would be nice for the main paper to end with a conclusion section rather than ending abruptly. And this conclusion should mention limitations of the current method.

**Questions:**

- Is there a difference in the alternating optimization scheme presented here and the one in [21]?
- How does Theorem 2 in this text relate to Theorem 5.4 in [21]?
- Am I correct in assuming that the three different datasets are used to evaluate how the coverage of the dataset around expert state-action pairs affects the algorithm's performance?

**Limitations:**

There is no discussion of limitations in the paper. The paper would be made a lot stronger if this was discussed in a conclusion section.

---

> ### Author Rebuttal · Authors · 2023-08-07
>
> We thank the reviewer for the detailed review of the paper and the valuable feedback. Below, we address the reviewer's comments in a point-by-point manner.
>
> **Response to Weakness 1&Question 1:** We appreciate the reviewer’s comments. However, we do not agree that the proposed alternating optimization algorithm in this work is identical to the one presented in [21]. In this response, we would like to highlight the key difference between our proposed algorithm and the one presented in [21].
>
> Compared with the online IRL algorithm presented in [21], one key difference is that our proposed offline IRL algorithm solves a **much harder** IRL problem (2a) - (2b) than the one in [21]. This is due to the fact that our outer objective $L(\theta)$ in eq (2a) is defined on the *ground-truth* dynamics model P while the inner problem in eq (2b) considers policy optimization in the *estimated* dynamics model $\hat{P}$.  Due to the existence of the dynamics model mismatch, we turn to optimize a novel surrogate objective by developing a *new* algorithm.
>
> To be more specific, compared with the existing algorithm in [21], there are a few major differences in our proposed algorithm. Indeed, as reviewer has mentioned, we adopt a generic “alternating optimization” scheme, which alternates between one conservative policy improvement step and one reward optimization step. However, the algorithm under the hood of “alternating optimization” is very different.
> * First, compared with the existing algorithm in [21], the proposed algorithm in paper aims to solve a completely different problem – offline IRL. Due to the difference in problem settings and formulations, we need to analyze the gradient expression of the surrogate objective function w.r.t. the reward parameter, which leads to a different reward update step compared to the one in [21].
>
> * Second, for the policy optimization under each reward estimator, we consider updating the policy under the estimated dynamics model with a regularization term (penalty function) based on the model uncertainty. In contrast, the existing algorithm in [21] only solves the online IRL problem where there is no estimated dynamics model and uncertainty estimation. This difference in algorithm design also leads to new algorithm implementations where we solve the underlying optimal policy in the estimated dynamics model through taking advantage of the recent advances in model-based offline policy optimization and uncertainty estimation.
>
> * Third, in the policy optimization subroutine (13) - (14), we consider a more realistic scheme compared with [21], where we first approximate the current soft Q function $Q_k$ by $\widehat{Q}_k$ in (13) and then update the policy by an approximate soft policy iteration in (14). Compared with the existing algorithm in [21] which simply assumes the soft Q-function can be accurately estimated without any approximation error, the proposed algorithm in our paper is more practical since the approximation error has been explicitly taken into account.
>
> **Response to Weakness 2&Question 2:** We appreciate the reviewer’s comments. In this response, we would like to highlight the key differences between Theorem 2 in this work and Theorem 5.4 in [21].
>
> In the convergence analysis of our proposed algorithm, the previous analysis in [21] does not hold anymore. This is due to the fact that we are optimizing a *surrogate objective function* which is different from [21]. Moreover, as opposed to the analysis considered in [21], the analysis of our proposed algorithm involves two dynamics models: the ground-truth dynamics model $P$ and the estimated dynamics model $\hat{P}$. Due to the existence of the dynamic model mismatch and regularized penalty function in the setting of offline IRL, it is not clear whether previous properties, such as the Lipschitz continuity proved in [21] can still hold here. Furthermore, as shown in eq (13) - (14) which is different from [21], we do not assume that the soft Q function $Q_k$ can be accurately approximated at each step. In the convergence analysis of [21], the accurate approximation of the soft Q function at each step can guarantee a monotonic improvement for each soft policy iteration step. However, in our work, the approximation error between $\widehat{Q}_k$ and $Q_k$ in (13) - (14) also makes it more challenging to analyze the stability of our proposed alternating algorithm. To tackle this problem, we develop new proof techniques in the proof of Theorem 2 to guarantee the finite-time convergence analysis of our proposed algorithm.
>
> We appreciate the reviewer’s comments, which encourages us to discuss the key difference in proof techniques compared with [21]. We will include the discussion above into our paper.
>
> **Response to Weakness 3:** We appreciate the reviewer’s comments. These transition datasets are used to train the dynamics model and further evaluate the effect of the dataset’s coverage over the expert-visited state-action space on the performance of benchmark algorithms. We will explicitly include these details and move the discussion in Appendix into the main paper.
>
> **Response to Weakness 4:** We appreciate the reviewer’s suggestion. We present a conclusion and discuss the limitations in the global response and we kindly refer the reviewer to check it (see https://openreview.net/forum?id=oML3v2cFg2&noteId=oCZvqhc8PI).
>
> **Response to Question 3:** Yes. Indeed, the three types of datasets are gathered from policies with varying performance levels and exhibit different coverage across the expert-visited state-action space. We leverage these three dataset types to assess how their coverage of the expert-visited state-action space influences the algorithm's performance.
>
> **Response to Limitations:** We would like to kindly remind the reviewer that the limitations of this work have been discussed in the section of Limitations and broader impacts in Appendix. We will discuss it in the conclusion section.

---

> > ### Comment · Reviewer_2Qwv · 2023-08-14
> >
> > Thank you for your clarifications! I have adjusted my score accordingly, assuming you make the proposed modifications to the text.

---

> > > ### Author Response · Authors · 2023-08-16
> > > **Many thanks for your comments and positive feedback**
> > >
> > > We truly appreciate your detailed review and insightful comments! We will include the discussions and the proposed modifications in our revised version.

---

### Official Review · Reviewer_VBZK · 2023-07-06

**Soundness:** 2 fair
**Presentation:** 3 good
**Contribution:** 3 good
**Rating:** 7
**Confidence:** 3

**Summary:**

In this paper the authors present a two level maximum likelihood based framework for offline inverse reinforcement learning, where both a world model and a reward model are learnt from expert demonstrations. In this two level algorithm the outer level or loop involves estimating the reward function, while the inner loop estimates the optimal policy for the chosen reward function in a conservative MDP setting, where a penalty is added which is loosely proportional to the uncertainty in the learnt model. The authors provide theoretical guarantees for the performance of their algorithm under fairly standard technical assumptions. They also show the numerical performance of their algorithm on 3 MuJoCo environments, comparing them with other state of the art offline RL algorithms.

**Strengths:**

1. The paper is novel, clearly written and is easy to comprehend.
2. The authors have stated their results formally in the form of Lemmas and Theorems and have proved them in the supplementary material. This analysis proves the validity and utility of their proposed approach.
3. While model based offline inverse RL has been studied, I think the theoretical guarantees from this paper are novel and important.

**Weaknesses:**

1. The authors have demonstrated performance on only 3 environments, in which in one of the cases, their proposed algorithm is not the best.

**Questions:**

1. How is the state action visitation measure defined in Eq (5)? Specifically, I did not understand the term $P(S_t = s|s_0 \sim \eta)$. Does this imply that the transitions are action independent?
2. In the proof of Lemmma 1, how is the equation below the one labelled (iii) obtained?
3. How is the last step in Eq(30) in Proof of Lemma 1 obtained?
4. [Typo] Line 806 "bonded" -> "bounded".
5. I have understood most of the proofs as they are well written with adequate justification, but since I am not sure about the definition of the state-action visitation measure, I was unable to verify some proofs in the supplementary material.

**Limitations:**

The authors address the limitations of this work and also some suggestions to overcome some of these limitations.

---

> ### Author Rebuttal · Authors · 2023-08-09
>
> We thank the reviewer for the detailed review of the paper and the valuable feedback. Below, we address the reviewer's questions in a point-by-point manner.
>
> **Response to Question 1:** We appreciate this question raised by the reviewer. This question indeed helps us find a typo in the term $P(s_t=s|s_0\sim\eta)$, which should be corrected as $P^\pi(s_t=s|s_0\sim\eta)$.
>
> Under any fixed policy $\pi(a|s)$ and transition function $P(s^\prime | s,a)$, the term $P^\pi(s_t=s|s_0\sim\eta)$ denotes the probability that $s_t = s$ at time step t when $s_0$ is sampled from the initial state distribution $\eta(\cdot)$ and the actions $\{a_0 , a_1, …, a_{t-1}\}$ in the Markov decision process are sampled from the policy $\pi$.
>
> To avoid potential confusion, under any policy $\pi$ and transition function $P$, let us correct the typo and rewrite the definition of the corresponding state-action visitation measure $d^\pi\_{P}(s,a)$ as below: $d^\pi\_{P}(s,a) := (1 - \gamma) \pi(a|s) \sum\_{t=0}^{\infty} \gamma^t P^\pi(s_t = s | s_0 \sim \eta)$.
>
> Hence, in Eq (5), the state-action visitation measure under the expert policy $\pi^E$ is defined as below:
> $d^E(s,a) := (1 - \gamma)  \pi(a|s) \sum\_{t=0}^{\infty} \gamma^t P^{\pi^E}(s_t = s | s_0 \sim \eta  )$.
>
> **Response to Question 2:** In (iii), we have shown that the objective function $L(\theta)$ can be expressed as $L(\theta)=\sum\_{t=0}^{\infty} \gamma^t \mathbb{E}_{(s_t,a_t)\sim(\eta,\pi^E,P)}[ r(s_t,a_t;\theta) + U(s_t,a_t) + \gamma \mathbb{E}\_{s\_{t+1} \sim \hat{P}(\cdot|s_t,a_t) }[V\_{\theta}(s\_{t+1})] ] - \sum\_{t=0}^{\infty}\gamma^t \mathbb{E}\_{s_t \sim (\eta,\pi^E,P)}[V\_{\theta}(s_t)].$
>
> Based on the equation in (iii), we can further write down the following equality: $L(\theta)=(\sum\_{t=0}^{\infty} \gamma^t \mathbb{E}_{(s_t,a_t)\sim(\eta,\pi^E,P)}[ r(s_t,a_t;\theta) + U(s_t,a_t) ]  + \sum\_{t=0}^{\infty} \gamma^{t+1} \mathbb{E}\_{(s_t,a_t)\sim(\eta,\pi^E,P),s\_{t+1} \sim \hat{P}(\cdot|s_t,a_t)}[V\_{\theta}(s\_{t+1})] ) - (\mathbb{E}\_{s_0 \sim \eta(\cdot)}[V\_{\theta}(s_0)] + \sum\_{t=0}^{\infty} \gamma^{t+1} \mathbb{E}\_{s\_{t+1} \sim (\eta,\pi^E,P)}[V\_{\theta}(s\_{t+1})] ).$
>
> Then it leads to the equality below the one labeled (iii) by interchanging terms.
>
> **Response to Question 3:**  In the first equation of eq. (30), we have obtained the following equation of the term T2:
> $ T_2 =  \sum\_{t=0}^{\infty} \gamma^{t+1} \mathbb{E}_{(s_t,a_t)\sim(\eta,\pi^E,P)}[\sum\_{s\_{t+1} \sim \mathcal{S}} V\_{\theta}(s\_{t+1}) (\hat{P}(s\_{t+1} | s_t, a_t) - P(s\_{t+1} | s_t, a_t))]$
>
> Here, recall that we have defined $P^{\pi^E}(s_t=s|s_0 \sim \eta)$ as the probability that $s_t=s$ when $s_0$ is sampled from the initial state distribution $\eta(\cdot)$ and the actions in the MDP are sampled from the expert policy $\pi^E$. Hence, we can obtain the second equality in (30): $T_2 = \gamma \sum\_{t=0}^{\infty} \sum\_{s \in \mathcal{S}, a\in \mathcal{A}} \gamma^t P^{\pi^E}(s_t = s|s_0\sim \eta) \pi^E(a_t = a|s_t = s)( \sum\_{s^\prime \in \mathcal{S}} V\_{\theta}(s^{\prime}) (\hat{P}(s^{\prime} | s_t=s, a_t=a) - P(s^{\prime} | s_t=s, a_t=a)) ).$
>
> Then we can further express the term $T_2$ as below:
> $T_2 = \gamma \sum\_{s \in \mathcal{S}, a\in \mathcal{A}} ( \pi^E(a|s) \sum\_{t=0}^{\infty} \gamma^t P^{\pi^E}(s_t = s|s_0 \sim \eta)) \cdot ( \sum\_{s^\prime \in \mathcal{S}} V\_{\theta}(s^{\prime}) (\hat{P}(s^{\prime} | s, a) - P(s^{\prime} | s, a)) )$.
>
> Given the expert policy $\pi^E$, recall that we have defined the corresponding state-action visitation measure as $d^E(s,a) :=(1-\gamma)\pi^E(a|s) \sum\_{t=0}^{\infty} \gamma^t P^{\pi^E}(s_t =s|s_0 \sim \eta).$
> Then we obtain the following expression of the term $T_2$ as below:
> $T_2 = \frac{\gamma}{1 - \gamma} \sum\_{s \in \mathcal{S}, a\in \mathcal{A}} d^E(s,a) \cdot ( \sum\_{s^\prime \in \mathcal{S}} V\_{\theta}(s^{\prime}) (\hat{P}(s^{\prime} | s, a) - P(s^{\prime} | s, a)) ).$
>
> Finally, we can show the last equality in eq (30):
> $T_2 = \frac{\gamma}{1 - \gamma} \mathbb{E}\_{(s,a)\sim d^E(\cdot, \cdot)}[ \sum\_{s^\prime \in \mathcal{S}} V\_{\theta}(s^{\prime}) (\hat{P}(s^{\prime} | s, a) - P(s^{\prime} | s, a))  ].$
>
> **Response to Question 4:** We thank the reviewer for pointing out this typo. We will correct it in our paper.
>
> **Response to Question 5:** We thank the reviewer for raising the question on the definition of the state-action visitation measure. The question indeed helps us find a typo in the definition of the state-action visitation measure. As we clarified in the response to Question 1, under any policy $\pi$ and transition function $P$, the corresponding state-action visitation measure $d^{\pi}\_{P}(s,a)$ is defined as below: $d^{\pi}\_{P}(s,a) := (1 - \gamma) \pi(a|s) \sum\_{t=0}^{\infty} P^\pi(s_t = s | s_0 \sim \eta)$.
> Here, $P^\pi(s_t=s|s_0\sim\eta)$ denotes the probability that $s_t = s$ when the actions in the MDP are sampled from the policy $\pi$ and the initial state $s_0$ is sampled from the initial state distribution $\eta(\cdot)$.
>
> We hope our correction of the typo and the corrected definition of the state-action visitation measure can address the reviewer’s question, and can improve the readability of our paper.

---

> > ### Comment · Reviewer_VBZK · 2023-08-19
> >
> > I thank the authors for responding to my comments and for providing these clarifications. Based on the explanation provided, I am happy to increase my score.

---

> > > ### Author Response · Authors · 2023-08-20
> > > **Many thanks for your comments and positive feedback**
> > >
> > > We truly appreciate your detailed comments and review! We will include these clarifications in our revised version.

---

### Official Review · Reviewer_5szs · 2023-07-07

**Soundness:** 4 excellent
**Presentation:** 4 excellent
**Contribution:** 4 excellent
**Rating:** 8
**Confidence:** 3

**Summary:**

The paper "Understanding Expertise through Demonstrations: A Maximum Likelihood Framework for Offline Inverse Reinforcement Learning" proposes an innovative approach of offline inverse reinforcement learning. After a deep theoretical analysis of the inter-dependence between errors arising from dynamics modeling from limited offline data and performance gaps of resulting policies, authors propose an efficient algorithm to practically exploit conclusions for reward/policy learning. A small experimental section validates the approach.

**Strengths:**

- Very well written with every notation well defined and every choice well justified
- Very interesting problem and strong theoretical analysis
- A practical algorithm that looks easy to reproduce
- Good results

**Weaknesses:**

- My main concern is that there is very few discussion about model uncertainty U in the paper, and particularly in section 3. I am surprised to not see it involved in the derivations and bounds, with no assumptions about it (except that it is bounded). No real meaning is given to it and it seems that it could be removed without changing anything in the theoretical conclusions. Is it true ? If yes, why introducing it in that part ? Also its impact is therefore not well understood from the theoretical analysis, which is a little be limitative to me (as it looks to have importance).
- Still on U, I feel that experimental results on the choice of U would have been very useful.

**Questions:**

- Could authors give more insights about U, both theoretically if possible, and by some experimental results that show its impact ?

**Limitations:**

.

---

> ### Author Rebuttal · Authors · 2023-08-09
>
> We thank Reviewer 5szs for your positive comments and recognizing the importance of this work. Below, we address the reviewer's comments in a point-by-point manner.
>
> **Our Response to Weakness 1:** We appreciate the reviewer for raising this insightful question. In our practical implementation, we utilize a set of ensemble models to construct the penalty function. It's important to note that the penalty function is based on heuristic design and relies solely on the estimated dynamics model. Consequently, once the construction of the estimated dynamics model is complete, the associated penalty $U(s,a)$ for each state-action pair $(s,a)$ becomes a fixed constant when using the proposed alternating algorithm for reward/policy update steps. It is essential to differentiate our approach from model-based offline reinforcement learning, where the reward function remains fixed and the penalty function plays a crucial role in alleviating distribution shifts. In contrast, in IRL, the parameterized reward function is kept adjusted and optimized to align with expert demonstrations. Thus, the penalty function $U(s,a)$, being a fixed regularization term added to the parameterized reward function, does not significantly impact the theoretical analysis of our offline IRL method. This characteristic makes model-based offline IRL less sensitive to the construction of the penalty function U, as compared to model-based offline reinforcement learning.
>
> Empirically, we've observed that the uncertainty-based penalty function effectively mitigates distribution shift effects during policy optimization in the estimated dynamics model, consistent with [R1]. In offline IRL settings, where a suitable initialization for the reward function may be lacking, the agent might suffer from distribution shifts in the initial training stages. Including the penalty function to regularize the agent's behavior and guide it to remain in the low-certainty region can facilitate the imitation of expert demonstrations. Furthermore, since the policy optimization subroutine in our proposed algorithm utilizes the practical implementations of model-based RL methods that explicitly incorporates the penalty function, it is natural to include the penalty function in our approach and theoretical analysis.
>
> [R1] Yu, Tianhe, et al. "Mopo: Model-based offline policy optimization." Advances in Neural Information Processing Systems 33 (2020): 14129-14142.
>
> **Our Response to Weakness 2:** We appreciate the reviewer for raising this insightful question. To address the reviewer's question, we provide a supplementary experiment on three different choices of U. Below, we will elaborate the experiment details.
>
> To estimate the model uncertainty by ensembles models, we have independently trained an ensemble of $N$ estimated dynamics models $\\{ P\_{\phi,\varphi}^i(s\_{t+1}|s_t,a_t)=\mathcal{N}(\mu^i\_{\phi}(s_t,a_t), \Sigma^i\_{\varphi}(s_t,a_t)) \\}\_{i=1}^N$ via likelihood maximization over transition samples. Here, each model estimates the location of the next state by gaussian distributions. Three common choices of the penalty function have been considered: 1) Max Aleatoric: $U(s,a)=-\max\_{i = 1,\cdots,N} || \Sigma^i\_{\varphi}(s,a) ||_F$, 2) Ensemble Variance: $U(s,a) = -(\frac{1}{N} \sum\_{i=1}^N (\Sigma^i\_{\varphi}(s,a))^2 + \frac{1}{N} \sum\_{i=1}^N (\mu^i\_{\phi}(s,a))^2 - (\bar{\mu}(s,a))^2)$ where $\bar{\mu}(s,a) = \frac{1}{N}\sum\_{i=1}^N \mu^i\_{\phi}(s,a)$, 3) Ensemble Standard Deviation: $U(s,a) = -\sqrt{\frac{1}{N} \sum\_{i=1}^N (\Sigma^i\_{\varphi}(s,a))^2 + \frac{1}{N} \sum\_{i=1}^N (\mu^i\_{\phi}(s,a))^2 - (\bar{\mu}(s,a))^2}.$ We evaluate the effect of the three penalty functions in our proposed algorithm on HalfCheetah. We provide 10 expert trajectories and utilize three transition datasets (medium-expert, medium, medium-play) from D4RL. We show the numerical results in the following table:
>
> | HalfCheetah | Ensemble Standard Deviation | Ensemble Variance | Max Aleatoric |
> |----------|----------|----------|----------|
> | Medium-Expert | $11137.21 \pm 872.47$ | $10941.80 \pm 878.90$ | $10752.50 \pm 655.96$ |
> | Medium-Replay | $8214.46 \pm 491.64$ | $8142.77 \pm 621.12$ | $8612.29 \pm 108.25$ |
> | Medium | $6324.95 \pm 556.42$ | $6454.43 \pm 664.95$ | $7973.86 \pm 108.86$ |
>
> According to our supplementary experiments, we observe that these three choices of penalty functions lead to similar final performance in our offline IRL method.
>
> **Our Response to Question 1:** We thank the reviewer for raising this insightful question. As we clarified in the response to Weakness 1, the penalty $U(s,a)$ is a fixed regularization term added on the reward function $r(s,a;\theta)$ and the reward function $r(s,a;\theta)$ will be updated every reward optimization step. Given that the reward function will be updated to align with expert demonstrations, the theoretical analysis will not heavily rely on the (fixed) penalty function.
>
> To better understand the effect of the penalty function, we provide a supplementary experiment on three different choices of U. We observe the final performance of the proposed offline IRL algorithm with different choices of penalty function will not deviate from each other by too much, since the penalty function represents a constant regularization term while the magnitude of the parameterized reward function will continuously update during the reward learning process..
>
> We also note that the uncertainty estimation is still an active research problem and the theoretical understanding is limited, especially in the context of neural networks. Hence, we utilize common heuristics design of the penalty function from the literature in model-based offline RL to develop our algorithm. We would like to thank the reviewer’s insightful comments again and we will consider developin theoretical understanding on the impact of penalty function in model-based offline IRL as one of the directions for future work.

---

> > ### Comment · Reviewer_5szs · 2023-08-11
> > **Thanks**
> >
> > Many thanks for these insightful answers, that well confortate the score I assigned to this submission.

---

> > > ### Author Response · Authors · 2023-08-11
> > > **Thank you for your positive evaluations**
> > >
> > > We sincerely appreciate you for taking time to review our paper and thank you for recognizing the contributions of this work.

---

### Author Rebuttal · Authors · 2023-08-10

We thank the detailed review and comments from all reviewers.

In the global response, we present a conclusion here and we will include it into our final version:

In this paper, we model the offline Inverse Reinforcement Learning (IRL) problem from a maximum likelihood estimation perspective. We develop a computationally-efficient algorithm that effectively recovers the underlying reward function and its associated optimal policy. We have also established statistical and computational guarantees for the performance of the recovered reward estimator. Through extensive experiments, we demonstrate that our algorithm outperforms existing benchmarks for offline IRL and Imitation Learning, especially on high-dimensional robotics control tasks. One limitation of our method is that we focus solely on aligning with expert demonstrations during the reward learning process. In an ideal scenario, reward learning should incorporate diverse metrics and data sources, such as expert demonstrations and preferences gathered through human feedback. One direction for future work is to broaden our algorithm framework and theoretical analysis for a wider scope in reward learning.

---

### Decision · Program_Chairs · 2023-09-21

**Decision:**

Accept (oral)

**Comment:**

The submitted paper was reviewed by 5 knowledgeable reviewers, all of whom recommended acceptance of the paper. Several initial concerns/questions were answered convincingly by the authors in their rebuttal and discussions with the reviewers. Hence, in line with the reviewers' recommendations, I am recommending acceptance of the paper. The authors are encouraged to further improve their paper by incorporating the feedback of the reviewers and relevant parts of the discussions with them in the final version of their paper.